

# Lieb-Schultz-Mattis, Luttinger, and 't Hooft – anomaly matching in lattice systems

**Meng Cheng[1] and Nathan Seiberg[2]**

**1** Department of Physics, Yale University, New Haven, CT 06511-8499, USA
**2** School of Natural Sciences, Institute for Advanced Study, Princeton, NJ 08540, USA

## Abstract

We analyze lattice Hamiltonian systems whose global symmetries have 't Hooft anomalies. As is common in the study of anomalies, they are probed by coupling the system to classical background gauge fields. For flat fields (vanishing field strength), the nonzero spatial components of the gauge fields can be thought of as twisted boundary conditions, or equivalently, as topological defects. The symmetries of the twisted Hilbert space and their representations capture the anomalies. We demonstrate this approach with a number of examples. In some of them, the anomalous symmetries are internal symmetries of the lattice system, but they do not act on-site. (We clarify the notion of "on-site action.") In other cases, the anomalous symmetries involve lattice translations. Using this approach we frame many known and new results in a unified fashion. In this work, we limit ourselves to 1+1d systems with a spatial lattice. In particular, we present a lattice system that flows to the $c = 1$ compact boson system with any radius (no BKT transition) with the full internal symmetry of the continuum theory, with its anomalies and its T-duality. As another application, we analyze various spin chain models and phrase their Lieb-Shultz-Mattis theorem as an 't Hooft anomaly matching condition. We also show in what sense filling constraints like Luttinger theorem can and cannot be viewed as reflecting an anomaly. As a by-product, our understanding allows us to use information from the continuum theory to derive some exact results in lattice model of interest, such as the lattice momenta of the low-energy states.



# 1   Introduction

A powerful theme in high energy physics and in condensed matter physics is constraining the behavior of complicated systems using their symmetries and their anomalies. The goal of this paper is to clarify some aspects of this theme in the context of systems of interest in condensed matter physics and to phrase them using the framework common in quantum field theory. In particular, we will discuss the Lieb-Schultz-Mattis theorem [1] and the Luttinger theorem [2,3] using this perspective.

    The original Lieb-Schultz-Mattis (LSM) theorem [1,4] stated that a translation-invariant, one-dimensional, spin-$\frac{1}{2}$ chain with SO(3)-invariant local interactions cannot be gapped with a unique ground state. For a review, see e.g., [5,6]. The key ingredient of the proof is to construct an excited state whose energy expectation value vanishes in the thermodynamic limit. Later, the construction was re-interpreted by Oshikawa [7] as the adiabatic insertion of a $2\pi$ U(1) flux,[1] and was further generalized to higher-dimensional spin systems. A rigorous proof of the higher-dimensional LSM theorem was given by Hastings [8].

---

[1]This is flux through a two-dimensional space bound by the one-dimensional space on which the degrees of freedom reside. Using a more mathematical language, it is continuously introducing a U(1) holonomy around the one-dimensional space. Below we will discuss it in more detail.

The original LSM theorem and its higher-dimensional generalizations rely on having a continuous internal symmetry. Further generalizations to finite internal symmetry in 1D spin chains have been formulated [9–12]. The general LSM theorem states that a translation-invariant spin chain cannot have a unique gapped (i.e., short-range correlated) ground state if each unit cell transforms as a nontrivial projective representation of the internal symmetry group. Various other generalizations of LSM theorems are discussed in e.g., [13, 14]. We will refer to these theorems collectively as LSM-type theorems.

Recent progress in the classification of symmetry-protected topological phases and their relation 't Hooft anomalies [15] has brought a new perspective on LSM-type theorems. LSM-type theorems forbid the existence of a short-range entangled state that preserves both translation and internal symmetries. It was recognized that this statement should be understood as a consequence of a mixed 't Hooft anomaly between the spatial translation symmetry and the internal symmetry [16–20]. Just as the familiar 't Hooft anomaly for internal symmetry, the mixed anomaly here leads to constraints on the low-energy theory. In [16], the anomaly constraints on gapped phases in 2+1d were derived by introducing defects of the internal symmetry into the theory. It was also shown that the same kind of constraints can be obtained by formally treating lattice translations as internal symmetries in the low-energy effective field theory [17–19, 21, 22].

Closely related to the mixed anomaly, lattice systems that satisfy LSM-type theorems can also be viewed as the boundary of a higher-dimensional crystalline SPT bulk [16,23]. A general classification of crystalline SPT phases was proposed in [24]. A central result from [24] is that there is an isomorphism between the group of crystalline SPT phases and that of SPT phases with internal symmetry, as long as the abstract symmetry group structures are the same. Another popular idea is that one can introduce defects, or "background gauge fields" of spatial symmetries on the lattice [16, 17, 19, 24–28], for example dislocations for translations or disinclinations for rotations, and it is conjectured that they can be described by the same mathematical framework as the internal symmetry defects.

As we said, our goal here is to phrase these advances in the framework of anomalies, as is common in quantum field theory. For the purpose of this paper, we will ignore gravitational anomalies, parity and time reversal anomalies, and conformal anomalies. In the continuum, this leaves only anomalies in internal global symmetries. The modern view of these anomalies involves coupling the system to classical background gauge fields for these symmetries, placing the system on a closed Euclidean spacetime, e.g., a torus, and studying the partition function.[2] The anomaly is the statement that this partition function transforms with additional phase factors under gauge transformations of these background fields, and the phase factors cannot be removed by introducing local counterterms. As a result, the global symmetry cannot be consistently gauged. For instance, 't Hooft anomaly in 0+1d quantum system with a global symmetry $G$ is the statement that the system realizes $G$ projectively. We will discuss examples of 't Hooft anomalies in 1+1d systems in great details in section 3 and 4. The existence of such an anomaly leads to 't Hooft anomaly matching conditions [15]. This is the statement that the anomaly computed in the short-distance theory should match the anomaly computed in the long-distance theory. In particular, a nontrivial anomaly cannot be matched by a completely trivial long-distance theory.

---

[2]There are several known ways of thinking about anomalies. All of them are special cases of the approach taken here, which is based on studying the partition function of the system coupled to background fields. This approach is also the starting point of the more abstract powerful treatment of anomalies in quantum field theory and in mathematics.

This understanding of anomalies in continuum theories raises a number of questions when we try to apply it to lattice systems:

- This picture of anomalies assumes a continuous and Lorentz invariant spacetime. Here, we try to apply it to the case where the microscopic systems are not Lorentz invariant and space is replaced by a lattice. How should we think of anomalies in this case? Does the fact that on the lattice there is no smoothness or continuity affect the discussion? For internal symmetries, methods to directly determine the 't Hooft anomaly have been developed in several important classes of lattice systems in [29–31]. However, it is not clear how to generalize these methods to include spatial symmetries.

- Some lattice systems, e.g., the one in [29,32], include internal symmetries, which do not act "on-site." (We will discuss them in detail in section 4.) Consequently, they cannot be coupled to background gauge fields. How should we analyze their 't Hooft anomalies using the quantum field theory approach?

- Of particular interest, especially in the context of the LSM theorem, is the symmetry of lattice translation. At long distances it leads to an internal symmetry and a continuous space translation. In this case, the symmetry groups of the short-distance theory and the long-distance theory are not the same. (We will discuss their group theory in sections 5 and 6.) How should we analyze an anomaly associated with such lattice translations? In this context, it was suggested to couple the system to "background gauge fields for the translation symmetry." What does this mean?

These questions were discussed by various authors including [16–19,23–33]. In addition, [34] suggested that Luttinger theorem should also be viewed as associated with an 't Hooft anomaly. We will elaborate on these discussions and state them in a unified framework.

In section 1.1, we will review the appearance of anomalies in quantum mechanics. Here, an anomaly means that the Hilbert space is in a projective representation of the internal symmetry group. A finite lattice system has a finite number of degrees of freedom and can be viewed as a particular quantum mechanical system. However, if we view it simply as a quantum mechanical system, we will miss the higher-dimensional anomalies.

The existence of higher-dimensional anomalies depends on locality of the lattice system. We take the lattice to be large, but finite. Then, the Hamiltonian is a finite sum of terms, each acting on near-by degrees of freedom. This local structure of the Hamiltonian allows us to introduce defects. They correspond to localized changes in the Hamiltonian. Since the defects are localized, for large (but finite) lattice, the dynamics far from the defects is not affected by their presence. As we will discuss, the properties of these localized defects capture the higher-dimensional anomalies.

For simplicity, in most of this paper, we will discuss systems in one spatial dimension. The generalization to higher dimensions introduces more elements, which we will not explore here.

When the global internal symmetry acts simply, i.e., on-site (see section 4), it is straightforward to turn on background gauge fields, as is common in lattice gauge theories. It is easy to see that in that case there cannot be an anomaly.

A more subtle situation, discussed in section 4, involves global internal symmetries that do not act on-site. In that case, we cannot couple the lattice system to general background gauge fields. However, as we will see, we can do it for background gauge fields with vanishing field strength. This is achieved by introducing twisted boundary conditions. Equivalently, we can think of the system as having topological defects.

Next, we explore how the system with defects transforms under its various global symmetries. These global symmetries include the internal symmetries that are left unbroken by the defects and also lattice translation. As we will see, these two symmetries (the remaining

internal symmetry and the translation symmetry) mix in a nontrivial way. These symmetries can be probed by computing the trace of the evolution operator with insertions of the symmetry operators. In other words, now we view the system as a quantum mechanical system (as opposed to a 1+1-dimensional system) in which all these symmetries, including the underlying translation, are ordinary global symmetries. With this interpretation, the trace with the symmetry operator insertion can be viewed as turning on temporal components of (flat) background gauge fields.

Anomalies arise when this trace is not gauge invariant – it transforms by a phase under gauge transformations of the various background fields. More precisely, we have to make sure that we cannot redefine the various operators such as to remove these phases.

It is convenient to characterize this anomaly by viewing the system as the boundary of a higher dimensional bulk and extend the background fields to the bulk. Then, we can make the partition function gauge invariant, but it depends on the extension to the bulk. This anomaly-inflow mechanism [35] that connects the system to a higher-dimensional bulk has been widely used in the literature of SPT phases. We emphasize though, that in our formulation adding such a bulk is merely a mathematical convenience. In some cases, including the study of SPT phases, such a bulk is physical. But the original theory and its anomaly can be studied even if no such bulk is physically present.

## 1.1 't Hooft anomaly in quantum mechanics

We start with a review of the simplest kind of 't Hooft anomaly that arises in 0+1d quantum mechanical systems.

Consider a quantum mechanical system with a global symmetry $G$. For simplicity let us assume that $G$ is unitary. This group acts faithfully on the operators in the theory.

Each $h \in G$ is implemented by a unitary transformation $\rho(h)$.[3] An operator $O$ transforms to ${}^h O = \rho(h) O \rho(h)^{-1}$ under the symmetry. The group structure implies that ${}^{h_1}({}^{h_2}O) = {}^{h_1 h_2}O$, and on states we must have

$$\rho(h_1)\rho(h_2) = e^{i\varphi(h_1, h_2)}\rho(h_1 h_2), \tag{1}$$

with phase factors $e^{i\varphi(h_1, h_2)} \in U(1)$ known as the Schur multipliers. Operators transform by conjugation and therefore these phases do not affect them. However, these phases are important in the action on the states. This means that the Hilbert space can be in a projective representation of $G$.

The phases $e^{i\varphi(h_1, h_2)}$ define a group 2-cocycle and further a cohomology class $[e^{i\varphi}]$ in $\mathcal{H}^2[G, U(1)]$. When the cohomology class $[e^{i\varphi}]$ is nontrivial, the symmetry has a 't Hooft anomaly. Equivalently, the Hilbert space is in a nontrivial projective representation of $G$.

To see the anomaly, consider the partition function of the system

$$Z[\beta] = \operatorname{Tr} e^{-\beta H}. \tag{2}$$

It can be thought of as a path integral over the system in Euclidean space where the Euclidean time is periodic, i.e., it is $S^1$. The background gauge field is classified by the holonomy around $S^1$. A general bundle can be created by inserting multiple operators in the trace:

$$Z[\beta, h_1, h_2, \ldots, h_n] = \operatorname{Tr}[e^{-\beta H}\rho(h_1)\rho(h_2)\cdots\rho(h_n)]. \tag{3}$$

---

[3]Below, we will discuss the action of $h$ in the presence of a twist in space by a group element $g$. We will denote it as $h(g)$. Then, the symmetry transformation in the untwisted problem, corresponding to $g = 1$ (or $g = 0$ in additive notation) will be denoted as $h(1)$ (or $h(0)$). In that case, there will be no reason to write $\rho(h(1))$.

The total holonomy is then $h_1 h_2 \cdots h_n$. Such a background with multiple insertions is gauge equivalent to one insertion of $h_1 h_2 \cdots h_n$. However, the partition function is generally not invariant under the gauge transformation. For example:

$$Z[\beta, h_1, h_2] = e^{i\varphi(h_1, h_2)} Z[\beta, h_1 h_2], \tag{4}$$

according to Eq. (1). We can attach arbitrary phase factors for each insertion, which amounts to redefining $\rho(h) \to u(h)\rho(h)$ where $u(h) \in U(1)$, and the phase $e^{i\varphi(h_1, h_2)} \to e^{i\varphi(h_1, h_2)} \frac{u(h_1)u(h_2)}{u(h_1 h_2)}$. Therefore the anomaly is labeled by the cohomology class.

Another kind of anomaly in quantum mechanics, which we will encounter below, is an anomaly in the space of coupling constants [36, 37]. The simplest example of it is a particle on a ring (a rotor) with a $2\pi$-periodic potential $V(\Phi)$. It is described by the Lagrangian, Hamiltonian, and commutation relations

$$\mathcal{L}(\theta) = \frac{1}{2}\dot{\Phi}^2 + \frac{\theta}{2\pi}\dot{\Phi} - V(\Phi), \qquad \Phi \sim \Phi + 2\pi,$$

$$H(\theta) = \frac{1}{2}\left(p - \frac{\theta}{2\pi}\right)^2 + V(\Phi), \tag{5}$$

$$[\Phi, p] = i.$$

Since $\Phi$ is periodic, the eigenvalues of $p$ are quantized and the theory with $\theta$ is the same as the theory with $\theta + 2\pi$. However these two systems are not exactly the same, they are related by a unitary transformation

$$H(\theta + 2\pi) = e^{i\Phi} H(\theta) e^{-i\Phi}. \tag{6}$$

If the potential $V(\Phi)$ is such that the system has a global symmetry, e.g., $\Phi \to \Phi + \frac{2\pi}{M}$ for some integer $M > 1$, then the unitary transformation in Eq. (6) transforms under this symmetry. This leads to a mixed anomaly between the $\theta$-periodicity and that global symmetry [36, 37]. The same kind of anomaly will figure below as an ordinary anomaly in higher dimensions.

## 1.2 Lightning review of some lattice models that flow to the $c = 1$ compact boson theory

In preparation for the discussion below, we would like to review two commonly used lattice models. Below, we will study them in more detail and will change them. Then we will couple them to gauge fields.

*The rotor model*

The Hamiltonian of the classical 2d XY model [38] can be thought of as an action for a 1+1d system in a discretized Euclidean time. Then, we can make time continuous and rotate to Lorentzian signature to find a Hamiltonian for a quantum 1+1d system. This Hamiltonian is known as the rotor model and it appears in various applications including a system of coupled Josephson junctions [39]

$$H = \sum_{j=1}^{L}\left(\frac{U}{2}\pi_j^2 - J\cos(\Phi_{j+1} - \Phi_j)\right), \qquad \Phi_j \sim \Phi_j + 2\pi,$$

$$[\Phi_j, \pi_{j'}] = i\delta_{j, j'}. \tag{7}$$

The coordinates $\Phi_j$ are circle valued, $\Phi_j \sim \Phi_j + 2\pi$, and correspondingly, their conjugate momenta $\pi_j$ have integer eigenvalues. The internal global symmetry of this system is

$O(2) = U(1) \rtimes \mathbb{Z}_2^{\mathcal{R}}$. Its $U(1) \subset O(2)$ subgroup is generated by $\sum_j \pi_j$ and the $\mathbb{Z}_2^{\mathcal{R}}$ generator $\mathcal{R}$ flips the signs of $\Phi_j$ and $\pi_j$.

For $J \gg U$, this lattice model flows to the a continuum conformal field theory. It is the $c = 1$ boson with radius[4] $R^4 = (2\pi)^2 \frac{J}{U}\left(1 + \mathcal{O}\left(\frac{U}{J}\right)\right)$. It is well known that this line of fixed points ends with a BKT transition at some value of $\frac{J}{U}$, which corresponds to $R = 2$. (Numerically, it is at $\frac{J}{U} \approx 0.8$ [40].) At smaller values of $\frac{J}{U}$, the model is gapped. We will return to this model in section 4.1.

This model is close to a distinct lattice model, which is also being referred to as the XY model. We will discuss it momentarily.

### The XXZ chain

Another system, that we will use is the spin-$\frac{1}{2}$ XXZ chain with the Hamiltonian

$$H = 2\sum_{j=1}^{L}\left(S_j^x S_{j+1}^x + S_j^y S_{j+1}^y + \lambda_z S_j^z S_{j+1}^z\right) = \sum_j\left(\tau_j^+ \tau_{j+1}^- + \tau_j^- \tau_{j+1}^+ + \frac{\lambda_z}{2}\tau_j^z \tau_{j+1}^z\right). \quad (8)$$

Here $S^\mu = \frac{1}{2}\tau^\mu$ for $\mu = x, y, z$ are spin-1/2 operators where $\tau^\mu$ are Pauli operators and

$$\tau^\pm = \frac{1}{2}(\tau^x \pm i\tau^y). \quad (9)$$

Note that below we will use $S^\pm = S^x \pm iS^y$ without the factor of 1/2.

Its global symmetry is $O(2) = U(1) \rtimes \mathbb{Z}_2^{\mathcal{R}}$. The $U(1)$ subgroup is generated by the charge $\sum_j S_j^z = \frac{1}{2}\sum_j \tau_j^z$, and the $\mathbb{Z}_2^{\mathcal{R}}$ is generated by $\mathcal{R} = \prod_j \tau_j^x$. It is enhanced to $SO(3)$ at $\lambda_z = 1$. We will discuss this $SO(3)$ invariant theory in section 5.

For $-1 < \lambda_z \le 1$, the continuum limit of this model is described by the $c = 1$ compact boson theory (see section 3) with radius [41, 42]

$$R_{\text{XXZ}}^2 = \frac{1}{1 - \frac{1}{\pi}\arccos(\lambda_z)}. \quad (10)$$

Again, our conventions are such that the self-dual radius is $R_{\text{XXZ}} = 1$ and it corresponds to $\lambda_z = 1$. At that point, the spin chain flows to the $SU(2)_1$ WZW model [43], which we will discuss in section 3.4.

For generic $-1 < \lambda_z \le 1$, the lattice $O(2)$ symmetry is enhanced in the continuum limit to $\left(U(1)_m \times U(1)_w\right) \rtimes \mathbb{Z}_2^{\mathcal{R}}$.[5] Denote the charges of the two $U(1)$ factors by $Q_m$ and $Q_w$. Then the lattice $U(1) \subset O(2)$ transformation is represented by $e^{i\xi Q_m}$ and the lattice $\mathbb{Z}_2^{\mathcal{R}} \subset O(2)$ flips the signs of $Q_m$ and $Q_w$. An important $\mathbb{Z}_2^{\mathcal{C}} \subset \left(U(1)_m \times U(1)_w\right) \rtimes \mathbb{Z}_2^{\mathcal{R}}$ symmetry of the continuum theory is generated by

$$C = e^{i\pi(Q_m + Q_w)}. \quad (11)$$

It arises from lattice translation and will be discussed in sections 1.3, 5, and 6.

At the special point with $\lambda_z = 0$, the Hamiltonian is $H = 2\sum_j\left(S_{j+1}^x S_j^x + S_{j+1}^y S_j^y\right)$. Using (10), this model flows to $R_{\text{XXZ}} = \sqrt{2}$, which corresponds to a free Dirac fermion. This model is also known as the XY model [1].

To conclude, the two systems (7) and (8) are distinct lattice models. They both have a global $O(2)$ symmetry and they both flow to the $c = 1$ compact boson. However, the range of radii of that boson is different in the two cases.

---

[4]Our conventions for the radius are that in the corresponding continuum conformal field theory, which we will discuss in section 3, T-duality maps $R \leftrightarrow \frac{1}{R}$ and hence the self-dual radius is $R = 1$.

[5]Following the CFT terminology, the labels $m$ and $w$ stand for momentum and winding.

### 1.3 Emanant symmetry

The XXZ lattice model of Eq. (8) leads to an interesting lesson about its IR symmetries.

Usually, emergent symmetries are not exact. They are violated by irrelevant operators. However, the $\mathbb{Z}_2^C$ symmetry of this model, which arises from lattice translation is an exact symmetry of the low-energy theory and is not violated even by irrelevant operators. (Related points have been mentioned by various authors in different contexts, e.g., [19].) This might appear surprising because this $\mathbb{Z}_2^C$ symmetry is not an exact symmetry of the finite $L$ lattice systems.

To see that, note that for large but finite $L$, every deformation of the low-energy theory must be invariant under the underlying $\mathbb{Z}_L$ lattice translation symmetry. This means that it is also invariant under this $\mathbb{Z}_2^C$ symmetry. More precisely, every operator in the low-energy theory that is invariant under the lattice $\mathbb{Z}_L$ translation symmetry, but violates the $\mathbb{Z}_2^C$ symmetry must have large momentum (of order $L$) under the translation symmetry of the low-energy theory. Therefore, it does not act in the low-energy theory. In the continuum limit with $L \to \infty$ such an operator has infinite spatial momentum and therefore it completely decouples.

For this reason, we do not refer to such a symmetry as an emergent symmetry, but as an emanant symmetry.[6] This will be discussed further and demonstrated in various examples in sections 5, 6, and 7.

Let us see the consequences of this understanding for the XXZ model. The standard BKT transition at $R_{\text{XXZ}} = 2$ (corresponding to $\lambda_z = -\frac{1}{\sqrt{2}}$) would be triggered by a $Q_w = \pm 1$ operator. Since this operator is $\mathbb{Z}_2^C$ odd, it cannot appear in the our low-energy action and therefore it does not trigger the transition. This is to be contrasted with the point $R_{\text{XXZ}} = 1$ (corresponding to $\lambda_z = 1$) where an operator with $Q_w = \pm 2$ becomes marginal. Since it is $\mathbb{Z}_2^C$ even, it can appear in the action and it triggers a BKT transition at this point. As a result, for $\lambda_z > 1$ the model is gapped, rather than leading to the conformal field theory with $R_{\text{XXZ}} < 1$. Related to that, for these values of $\lambda_z$ and $R_{\text{XXZ}}$ the relation (10) is not valid.

Emanant symmetries can also have anomalies. They will be discussed in sections 5, 6, and 7. There can be mixed anomalies between them and exact internal symmetries of the lattice theory. There can also be anomalies involving only emanant symmetries (e.g. $\mathbb{Z}_2^C$ in the XXZ model). These are closely related to a peculiar effect in a finite-size lattice system, namely the dependence of the ground-state lattice momentum on the system size. In particular, in sections 5 and 6, we will show that this dependence on the system size is a lattice precursor of the anomaly of the emanant symmetry in the continuum theory.

Before we end this discussion of emanant symmetries that arise from lattice translation, we would like to comment on a similar but distinct phenomenon. It is often the case that the low-energy theory of a system exhibits a generalized global symmetry [44], which is not present in the short-distance system. A typical example is QED – a U(1) gauge theory coupled to a massive electron. It has a magnetic one-form symmetry, but no electric one-form symmetry. At energies below the mass of the electron, the effective theory is a pure U(1) gauge theory without matter and it has an electric one-form symmetry. As for the emanant symmetries discussed above, there are no local operators (not even irrelevant ones) in the low-energy theory that violate it. However, unlike the discussion above, the generalized symmetry of the low-energy theory does not arise from microscopic translation. Other such examples are various gapped phases with topological order. The topological order is described by a topological field theory and exhibits various generalized symmetries that are not violated by any local operator.

---

[6]We thank T. Banks for suggesting this terminology.

### 1.4 Outline and summary

In section 2, we will present our general setup. We will start with the untwisted theory, which does not have any defects. Then, we will couple it to spatial background fields, i.e., introduce spatial twists, or equivalently, topological defects. Importantly, the global symmetry of the twisted theory is not the same as that of the untwisted theory. First, the internal symmetry is partially broken by the twist. More interestingly, the symmetry of lattice translation mixes in a nontrivial way with the internal symmetry. Finally, this whole new symmetry group might be realized projectively.

Section 3 will review the symmetries and the anomalies of the continuum theory of the $c = 1$ compact boson. This is the continuum limit of the lattice models reviewed in section 1.2. It is also the continuum limit of lattice models in later sections.

The new results in this paper will start in section 4. The expert reader, who is familiar with the background material can start reading there and consult the earlier sections for details. Here, we will discuss lattice models with internal symmetries with 't Hooft anomalies. We will present a new lattice model that flows to the $c = 1$ compact boson for every radius. (This models does not have a BKT transition). It has the full symmetry group $\left(\mathrm{U}(1)_m \times \mathrm{U}(1)_w\right) \rtimes \mathbb{Z}_2^{\mathcal{R}}$ of the continuum theory and it also has its anomaly. We will then discuss a known lattice model, the one first introduced in [32] and will analyze it using our approach.

In sections 5 and 6, we will discuss mixed anomalies between internal symmetries and lattice symmetries. This goes slightly outside the treatment of anomalies involving internal symmetries, as we cannot introduce traditional background gauge fields for them. (We will comment about this in more detail below.) In the continuum limit, lattice translation leads to a new internal symmetry. Using the terminology in section 1.3, this is an emanant symmetry. And then these anomalies become ordinary anomalies of internal symmetries and fit the standard framework of anomalies.

This discussion will be applied in section 5 to the SO(3) spin chain. Here we will reproduce the LSM result and some of its known generalizations using our approach. Among other things, our understanding will lead to a number of exact results about the spectrum of the finite size lattice and will clarify some peculiar facts that were found numerically.

In section 6, we will generalize the results in section 5 to other systems. In particular, we will discuss systems where lattice translation leads in the continuum limit to an internal emanant $\mathbb{Z}_n$ global symmetry. We will also see that the internal symmetries of the lattice system can mix with this emanant discrete internal symmetry.

Finally, in section 7, we will apply our understanding to Luttinger's theorem and other filling constraints. Phrasing these in our framework shows that these are not associated with standard 't Hooft anomalies. However, we will show that when the system is formulated with fixed U(1) charge (or fixed chemical potential) it has an interesting anomaly that leads to constraints similar to 't Hooft anomalies on the long-distance behavior of the system. We will demonstrate in a particular example that the low-energy theory has an emanant internal symmetry that depends on the filling fraction. In this example, this emanant symmetry has a mixed anomaly with the U(1) charge symmetry.

## 2 General setup

### 2.1 The untwisted problem

Let us setup the problem. We study a one-dimensional lattice with periodic boundary conditions. The sites are labeled by $j = 1, 2, \cdots, L$ and we will often use a notation where $j$ is an arbitrary integer subject to the identification $j \sim j + L$. This system has a $\mathbb{Z}_L$ translation

symmetry generated by a shift by one lattice site $T$ satisfying

$$T^L = 1. \tag{12}$$

We choose the convention that $T$ acts on a local operator $\mathcal{O}_j$ on a site $j$ as

$$T\mathcal{O}_j T^{-1} = \mathcal{O}_{j+1}. \tag{13}$$

There can also be parity symmetry, extending it to $\mathbb{D}_L$. In addition, there is an internal symmetry group $G$. Even though we can contemplate more complicated situations, we will assume that the full symmetry group $\mathcal{G}$ factorizes as $\mathcal{G} = G \times \mathbb{D}_L$.

We take the time direction to be Euclidean and compact and parameterize it by $\tau \sim \tau + \beta$. Then, the partition function is given in terms of the Hamiltonian $H$ as

$$\mathcal{Z}(\beta, L) = \mathrm{Tr}[e^{-\beta H}]. \tag{14}$$

In the coming subsections we will briefly review how we couple the system to background gauge fields. The discussion will be quite general and we refer the reader to the more detailed discussion below, where we will also present some examples.

## 2.2 Twisting in Euclidean time

First, we turn on background gauge fields for the internal symmetry group $G$ with components along the Euclidean time direction. For every element $h \in G$, there is an operator acting on the Hilbert space, $\rho(h)$. This allows us to generalize the partition function (14) to

$$\mathcal{Z}(\beta, L, h) = \mathrm{Tr}[e^{-\beta H}\rho(h)]. \tag{15}$$

The simplest anomaly arises when $\rho(h)$ realizes the symmetry group $G$ projectively. See section 1.1 for a review.

We can further generalize the partition function (15) by inserting a power of the translation operator $T$

$$\mathcal{Z}(\beta, L, h, n) = \mathrm{Tr}[e^{-\beta H}\rho(h)T^n]. \tag{16}$$

In terms of our Euclidean spacetime, this means that as we go around the Euclidean circle, we glue the spatial lattice after a shift by $n$ sites. Using Eq. (12), we have

$$\mathcal{Z}(\beta, h, n+L) = \mathcal{Z}(\beta, h, n). \tag{17}$$

If we think of our lattice system as a generic quantum mechanical system without paying attention to locality in space, the $\mathbb{Z}_L$ translation symmetry is simply another internal symmetry. This suggests that we can interpret the insertion of $T^n$ in Eq. (16) as a result of a background "time component of a translation gauge field." However, this interpretation will not be valid when we further generalize Eq. (16) below. See in particular the comments in footnotes 10 and 14.

## 2.3 Twisting in space

Some readers might find our general description below too abstract. They can skip it and go directly to the examples below, especially those in section 4. Hopefully, after going through these examples, this general discussion will be easier to follow.

The next crucial element is to introduce background gauge fields of the internal symmetry $G$ along the spatial direction. We limit ourselves to flat gauge fields. Then the choice of background gauge field is a choice of holonomy around the circle. It is labeled by a conjugacy

class $[g]$ of $G$, where $g$ is a representative element of the conjugacy class, and we will label the system by $[g]$.

One way to introduce this holonomy is to extend the system, let the site index $j$ be an arbitrary integer, and impose twisted boundary conditions on all the fields and operators

$$\mathcal{O}_{j+L} = g\mathcal{O}_j g^{-1}. \tag{18}$$

(This means that the fields are not functions on our lattice, but sections of a $G$ bundle.) Therefore, we refer to the system as twisted by $g$. We can further conjugate the entire system by an arbitrary group element $g'$ and write this equation as $g'\mathcal{O}_{j+L}g'^{-1} = g'g\mathcal{O}_j g^{-1}g'^{-1}$, thus showing that a twist by $g \in G$ is related to a twist by $g'gg'^{-1}$, i.e., it depends only on the conjugacy class of $g$.

A crucial point is that using (13), we learn that the relation $T^L = 1$ of (12) becomes

$$T^L = g. \tag{19}$$

Below we will discuss this relation in more detail and will see that it can be realized projectively. For that purpose we will have to define the various objects in this equation more carefully. This relation means that in the twisted theory, the translation symmetry and the internal symmetry mix in a nontrivial way. We will elaborate on this point in section 2.3.2.

### 2.3.1 The twist as topological defects

We will find it convenient to use another presentation of these background gauge fields. We use periodic boundary conditions, i.e., $\mathcal{O}_{j+L} = \mathcal{O}_j$ and represent the flat background gauge field as a change in the Hamiltonian. (Unlike (18), now the fields are functions rather than sections on our lattice.) To do that, we first trivialize the gauge field. This means that we cover our space with patches with transition functions between them.[7] Then, we absorb the transition functions in changes in the Hamiltonian. After doing it, the information in the background gauge field is captured by some position dependent coupling constants in the Hamiltonian. Since the transition functions are localized in space, the change in the Hamiltonian is also localized in space and we can interpret the background gauge field as adding a defect to the system.

Of course, the choice of the trivialization can be changed. This is done by performing gauge transformations that can change the transition functions and their locations. In the presentation with periodic boundary conditions and position dependent coupling constants, such changes in the trivialization amount to changing these coupling constants. Since these changes reflect the same holonomy of the background gauge fields, these changes of the Hamiltonian can be implemented by unitary transformations. In terms of defects, this means that the defects are topological. They can be moved using unitary transformations.

In fact, as we will see, in some cases, the local change in the Hamiltonian is not merely a function of the group element $g$, but can depend also on a certain lift of it. Again, different lifts are related by unitary transformations.

At this point, it is good to keep in mind the simple quantum mechanical example discussed around (5). There, we also saw a Hamiltonian that depends on a coupling constant $\theta$. The

---

[7]In special cases, where there is a group element $g' \in G$ such that $(g')^L = g$, we can "spread the twist homogeneously." We choose $L$ patches with $L$ equal transition functions between them. Starting with the original Hamiltonian with twisted boundary conditions (18), we redefine the fields $\phi_j' = (g')^{-j}\phi_j(g')^j$ and change the Hamiltonian accordingly. The new fields are subject to periodic boundary conditions $\phi_{j+L}' = \phi_j'$ and the problem is manifestly translational invariant. However, we prefer not to do it for the following reasons. First, the condition $(g')^L = g$ might not have a solution, and even if it does, it depends on $L$. More importantly, the Hamiltonian changes at every point on the lattice, hiding the fact that this modification of the problem should not affect the local dynamics.

system with $\theta$ was the same as the system with $\theta + 2\pi$, but the Hamiltonian was not the same. Instead, the two Hamiltonians were related by a unitary transformation (6).

We summarize that the twisted problem depends on the conjugacy class $[g]$ and some additional parameters. These additional parameters can be changed by unitary transformations. In order not to clutter the expressions, in the discussion below, we will mostly suppress these additional parameters, except when they are important.

### 2.3.2 The symmetry operators of the twisted problem

One advantage of using this defect picture of the twisted problem, where the Hamiltonian $H(g)$ depends on $g$, is that unlike the presentation (18), we can keep the other operators as they were in the untwisted theory. This is true for most of them, but if the twisted system has symmetries, we would like to identify the corresponding symmetry operators of the twisted theory. We will denote them as $\mathcal{O}(g)$, suppressing the additional parameters (which can be changed by unitary transformations), except when they are important.

Consider first symmetry operators associated with the internal symmetry $G$ and denote them by $h \in G$. Typically, they are not the same as in the untwisted problem

First, the spatial holonomy $[g]$ can reduce the global internal symmetry. One way to see it is that even though $h$ commutes with the original Hamiltonian, it might not commute with $H(g)$. The condition for them to commute is

$$gh = hg. \tag{20}$$

So the internal symmetry group is reduced from $G$ to the centralizer

$$C_g = \{h \in G \,|\, hg = gh\}. \tag{21}$$

Also, if we conjugate $g$ to find another twisted problem labeled by the same conjugacy class $[g]$, the group element $h$ should also be conjugated. Therefore, it depends on the other parameters, which we neglect.

We will denote the $h$ symmetry transformation in the $g$-twisted system as $h(g)$. Using this notation, $h(1)$ denotes the action of $h$ in the untwisted problem, which we earlier denoted by $\rho(h)$.[8]

We mentioned above that the internal symmetry group $G$ could act projectively in the original problem. Regardless of whether this is the case, the centralizer $C_g$ could act projectively in the twisted problem. In particular, we can have

$$h_1(g)h_2(g) = e^{i\varphi_g(h_1,h_2)}(h_1 h_2)(g), \qquad \text{for} \qquad h_1, h_2 \in C_g, \tag{22}$$

with a nontrivial phase $e^{i\varphi_g(h_1,h_2)}$ that depends on the twist parameter $g$. Such a phase is a manifestation of the 't Hooft anomaly of the $G$ symmetry.

Since $g \in C_g$, the group relation (20) itself could be realized projectively:

$$h(g)g(g) = e^{i\varphi_g(h,g)}(hg)(g) = e^{i(\varphi_g(h,g)-\varphi_g(g,h))}g(g)h(g), \qquad \text{for} \qquad h \in C_g. \tag{23}$$

This means that if $e^{i(\varphi_g(h,g)-\varphi_g(g,h))} \neq 1$, the operator corresponding to the temporal twist $h$ does not commute with the operator corresponding to the spatial twist $g$. Below, we will see examples of this phenomenon.

Next we turn to the translation $T$. Since defects are topological, the twisted system still has translation invariance. (This is obvious in the presentation (18).) However, the naive $\mathbb{Z}_L$

---

[8]In some cases below, where the group $G$ is Abelian, we will use an additive, rather than a multiplicative notation for its elements. Then, we will write $h(0)$ for the operators in the untwisted problem. We hope that it will be clear from the context whether we use a multiplicative or an additive notation.

translation symmetry generated by $T$ satisfying (12) is no longer present. Clearly the simple translation generator $T$ does not commute with $H(g)$. It shifts a defect at $I$ to $I+1$. However, we said above that the defects can be shifted using unitary transformations and therefore they are topological. This shift is associated with the action of $g$ around $I$. As a result, there is a new translation operator $T(g)$ satisfying

$$[T(g), H(g)] = 0 \,. \tag{24}$$

Therefore, $T(g)$ generates a new translation symmetry.

Interestingly, $T(g)^L$ is nontrivial. To see that, consider the system with a single defect at $I$. Recall that $T(g)$ includes an action of $g$ around $I$. Repeating this process $L$ times, the lattice returns to itself, but the defect is dragged all around the lattice. (See Fig. 1.) This leads to the key equation[9]

$$T(g)^L = e^{i\varphi_T(g)} g(g) \,, \tag{25}$$

with a projective phase $e^{i\varphi_T(g)}$. Recall our notation in which $T(g)$ and $g(g)$ are the operators in the problem twisted by $g$. It is easy to see that the relation (25) holds when there are multiple defects. Comparing with the related expression $T^L = g$ of (19), the latter is a group relation. The relation (25) takes into account the precise form of the operator (through the dependence on $g$) and its possible phase factor when it acts on states.

The key relation (19), and its realization as (25), show that in the $g$-twisted problem, the $\mathbb{Z}_L$ translation symmetry and the internal symmetry $C_g$ mix in a nontrivial way. If the total symmetry group of the untwisted theory (ignoring parity) is $\mathcal{G} = G \times \mathbb{Z}_L$, then the symmetry group of the $g$-twisted problem, $\mathcal{G}_g$ satisfies

$$\mathcal{G}_g / C_g = \mathbb{Z}_L \,. \tag{26}$$

For example, if $G = \mathbb{Z}_2$ and the symmetry group (ignoring parity) in the untwisted theory is $\mathcal{G} = \mathbb{Z}_2 \times \mathbb{Z}_L$, because of (19), in the twisted theory, it is $\mathcal{G}_g = \mathbb{Z}_{2L}$.[10] Below we will see interesting consequences of this equation.

We emphasize that the relation (19) is not an anomaly. It is simply the statement that the translation and internal symmetries mix in the twisted theory. When this relation is realized projectively, it could signal an anomaly.

In general, the twisted problem can realize the symmetry group $\mathcal{G}_g$, including both internal and translation symmetries, projectively. As we mentioned around (22), it is possible that $C_g$ is represented projectively. The phase in (25) represents another projective action. Another such phase can appear in[11]

$$h(g)T(g) = e^{i\varphi_g(h,T) - i\varphi_g(T,h)} T(g)h(g) \,, \qquad \text{for} \qquad h \in C_g \,. \tag{27}$$

When $e^{i\varphi_g(h,T)} \neq e^{i\varphi_g(T,h)}$, there is a mixed anomaly between the translation and the internal symmetry. We will discuss this type of anomaly in more details in sections 4 and 5.

---

[9]Similar relation between complete translation of space and the spatial twist operator is standard in theories with twisted boundary conditions. See e.g., the recent papers [45, 46].

[10]This mixing between translation and the internal symmetry shows that the symmetry group in the Hilbert space twisted by $g$ is not merely a subgroup of the symmetry group of the untwisted problem. Nor is it an extension of it. This is one reason we cannot view of the action by the operator $T(g)$ as representing a background value of a "translation gauge field."

[11]The phases can be further constrained in the following way. Associativity of $T(g)h_1(g)h_2(g)$ leads to $e^{i[\phi_g(h_1) + \phi_g(h_2)]} = e^{i\phi_g(h_1 h_2)}$, where $\phi_g(h) = \varphi_g(h,T) - \varphi_g(T,h)$. Therefore, these phases give a homomorphism from $C_g$ to U(1).



No defect and no operator

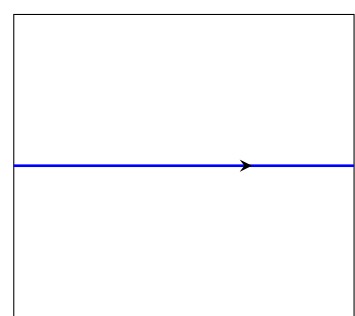

No defect. A $g$-operator is inserted at a fixed time. The arrow distinguishes it from $g^{-1}$.



A $g$-defect. The arrow distinguishes it from $g^{-1}$.

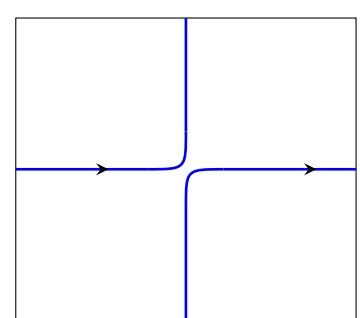

The action of $T^L$ drags the defect around space and hence $T^L = g$.

Figure 1: Rotating space in the presence of a defect. Space runs horizontally and time runs vertically. We see the effect of the symmetry operator $g$, the $g$-defect, and a pictorial derivation of Eq. (19) or its realization (25).

### 2.3.3 The partition function in the twisted system

We are now ready to generalize the partition function (16) to the twisted system. We write

$$\mathcal{Z}(\beta, L, g, h, n) = \text{Tr}[e^{-\beta H(g)} h(g) T(g)^n], \qquad \text{for} \qquad h \in C_g. \qquad (28)$$

It can be interpreted as a partition function of the Euclidean problem when space is twisted by $g$, Euclidean time is twisted by the internal symmetry group element $h$ and by the translation element $T^n$.

We stated above that the holonomies in space or time alone depend on the conjugacy class of $g$. When we consider both space and time (i.e., nontrivial $g$ and nontrivial $h$), we should identify the problems with

$$(g, h) \sim (\tilde{g} g \tilde{g}^{-1}, \tilde{g} h \tilde{g}^{-1}). \qquad (29)$$

Taking into account the condition (20),[12] we see that the parameters are in

$$\mathcal{M}_G = \{g, h \in G | gh = hg\} / \sim . \qquad (30)$$

They label a flat $G$ gauge field on a 2-torus. However, as we will see, $\mathcal{Z}$ might not be a good function on $\mathcal{M}_G$. It could be a section of a line bundle, thus signaling an anomaly.

---

[12]Recall that as in (23), the relation (20) could be realized projectively.

The full parameter space $\mathcal{M}$ that $\mathcal{Z}$ depends on includes the parameters labeling the flat gauge fields $\mathcal{M}_G$ of (30) and the integer $n$. And they are subject to the identifications following from Eq. (30) and the group relations $T^L = g$ and $hT = Th$.

An anomaly arises when $\mathcal{Z}$ is not a function on $\mathcal{M}$, but reminiscent of a section of a line bundle over $\mathcal{M}$.[13] This means that the partition function changes by a phase, as we compare its value at two points that should be identified.

For such phases as a function of the temporal twists $h$ and $T^n$, this is the same as an anomaly in quantum mechanics, i.e., the Hilbert space is in a projective representation of the symmetry group $\mathcal{G}_g$. The important point here is that these are the symmetries of the twisted problem, which are different than the symmetries of the untwisted problem. Since the twist parameters $g$ appear as coupling constants in the Hamiltonian, phases associated with them are similar to the anomaly in the space of coupling constants in [36, 37] (see the discussion around equation (5).)

Let us see some special cases of such anomalies. First, using Eq. (25), we have

$$\mathcal{Z}(\beta, L, g, h, n + L) = e^{i\varphi(g,h,n)} \mathcal{Z}(\beta, L, g, hg, n). \tag{31}$$

The phase $e^{i\varphi(g,h,n)}$ arises when $h(g)T(g)^{L+n}$ differs from $(hg)(g)T(g)^n$ by a phase. Sometimes, this phase can be removed by redefining $h(g)$ and $T(g)$. But then, such a redefinition might introduce phases elsewhere. In the following sections we will discuss specific examples of such phases.

As another special case, assume that the phase in (27) is nontrivial. Then, we can insert $T(g)T(g)^{-1}$ in the trace and cycling $T(g)$ around we find that

$$\mathcal{Z}(\beta, L, g, h, n) = 0. \tag{32}$$

We emphasize that this is not an inconsistency of the model or of the observable. It is simply the statement that states cancel each other in the trace (28), which reflects the 't Hooft anomaly.

Before we end this section, we would like to make a few comments.

First, the twist by the spatial translation generator $T$ or its twisted version $T(g)$ is introduced only along the Euclidean time direction. We did not include it as a twist in the spatial directions. Below we will see that in some cases, a change in $L$ behaves like a spatial twist in some global symmetry. While this seems to make sense in the continuum limit of $L \to \infty$, it is less clear for finite $L$.[14]

Second, it is interesting to extend our discussion to higher dimensions. A simple extension involves repeating this discussion for the various lattice symmetries including translations and rotation. But we can also introduce twists associated with lattice symmetries in space and in time. See [24–27] for interesting works in this direction.

Third, our discussion of the twisted problem is reminiscent of the standard analysis of orbifolds. In orbifolds, as here, we couple the system to background gauge fields and then we

---

[13]We used the phrase "reminiscent of a line bundle" because $\mathcal{M}$ includes disconnected parts labeled by an integer $n$. Below we will be imprecise and refer to it as a section.

[14]One might think of a change in $L$ as follows. As in the discussion around (18), start with an infinite chain and define the finite chain with $L$ sites with periodic boundary conditions by imposing the relation $T^L = 1$. Similarly, we can define the chain with $L$ sites with twisted boundary conditions by starting with the infinite chain and imposing $T^L = g$. This is equivalent to the standard treatment, which we also follow in this paper.

Now, consider imposing $T^L = gT^k$. This leads to a closed chain with $|L-k|$ sites and twisted boundary conditions. However, we could also try to interpret it as a chain with $L$ sites, whose boundary conditions are twisted by $gT^k$. This picture is particularly useful for $L \gg |k|$ and is consistent with the intuitive picture that a change in $L$ can be thought of as a twist by $T$. However, it is easy to see that this way of thinking about changing $L$ needs some clarifications for finite $L$. In particular, the Hilbert spaces of the systems with different values of $L$ have different sizes. Therefore, we cannot think of $T^L = gT^k$ as twisted boundary conditions for the system with $L$ sites. Also, a twist by an internal symmetry $g \in G$ breaks $G$ to the centralizer $C_g \subset G$. However, a change of $L$ does not break the $\mathbb{Z}_L$ translation symmetry to a subgroup. Instead, it turns it into $\mathbb{Z}_{L-k}$.

make them dynamical, i.e., we sum over them. This is possible only when $\mathcal{Z}$ is a good function on $\mathcal{M}$, i.e., there is no anomaly. See [47] for a discussion of the consistency of this gauging.

## 3 Example in the continuum: The $c = 1$ compact boson

In this section, we will review known properties of the $c = 1$ free compact boson and its anomalies from the perspective that we will use later in the discussion of lattice models.

The theory is characterized by the free Lagrangian

$$\mathcal{L} = \frac{R^2}{4\pi} \partial_\mu \Phi \partial^\mu \Phi, \qquad \Phi \sim \Phi + 2\pi. \tag{33}$$

In the condensed matter literature, such a theory is often referred to as a $c = 1$ Luttinger liquid, described by the following Lagrangian:

$$\mathcal{L} = \frac{1}{2\pi} \partial_t \Phi \partial_x \Theta - \frac{v}{4\pi} \left[ K^{-1} (\partial_x \Theta)^2 + K (\partial_x \Phi)^2 \right], \quad \Theta \sim \Theta + 2\pi, \quad \Phi \sim \Phi + 2\pi. \tag{34}$$

Here $v$ is the velocity. Integrating out $\Theta$, we obtain

$$\mathcal{L} = \frac{K}{4\pi v} \left[ (\partial_t \Phi)^2 - v^2 (\partial_x \Phi)^2 \right], \tag{35}$$

which can be identified with (33) if we set the speed of light $v$ to $v = 1$ and $K = R^2$. Alternatively, we can integrate out $\Phi$ and find the T-dual Lagrangian

$$\mathcal{L} = \frac{1}{4\pi R^2} \partial_\mu \Theta \partial^\mu \Theta, \qquad \Theta \sim \Theta + 2\pi. \tag{36}$$

For a generic value of $R$, the global internal symmetry of this system is

$$G_{c=1} = \left( U(1)_m \times U(1)_w \right) \rtimes \mathbb{Z}_2^{\mathcal{R}}. \tag{37}$$

The two U(1) factors are generated by charges $Q_m$ and $Q_w$ whose eigenvalues are integers. (Following the CFT terminology, the labels stand for momentum and winding.) We denote the corresponding group elements as

$$
\begin{aligned}
h_m(\xi_m) &= e^{i\xi_m Q_m}, \\
h_w(\xi_w) &= e^{i\xi_w Q_w}.
\end{aligned}
\tag{38}
$$

We will soon see that the symmetry transformations $h_m$ and $h_w$ can act projectively in the twisted Hilbert space. Then, a more careful notation will be needed.

The $R$-dependent contribution to the left and right dimensions, or equivalently the energy and momentum is

$$
\begin{aligned}
L_0 &= \frac{1}{4} \left( \frac{Q_m}{R} + R Q_w \right)^2 + \cdots, \\
\bar{L}_0 &= \frac{1}{4} \left( \frac{Q_m}{R} - R Q_w \right)^2 + \cdots, \\
E &= L_0 + \bar{L}_0 = \frac{Q_m^2}{2R^2} + \frac{R^2 Q_w^2}{2} + \cdots, \\
P &= L_0 - \bar{L}_0 = Q_m Q_w + \cdots.
\end{aligned}
\tag{39}
$$

Here and below, the ellipses in these quantities stand for contributions of the nonzero modes and the zero-point energy. They are independent of the charges $Q_m$ and $Q_w$ and the radius $R$. In this convention, T-duality maps $R \leftrightarrow \frac{1}{R}$, i.e., the selfdual radius, for which the theory has an SU(2) global symmetry is $R = 1$.

Sometimes it is also convenient to use a chiral basis for the charges

$$
\begin{aligned}
Q &= \frac{1}{2}\left(\frac{Q_m}{R} + RQ_w\right), \\
\overline{Q} &= \frac{1}{2}\left(\frac{Q_m}{R} - RQ_w\right), \\
\xi &= R\xi_m + \frac{\xi_w}{R}, \\
\overline{\xi} &= R\xi_m - \frac{\xi_w}{R},
\end{aligned}
\tag{40}
$$

where $Q$ is left-moving and $\overline{Q}$ is right-moving. But for most of our discussion we will use the non-chiral basis.

The $\mathbb{Z}_2^{\mathcal{R}}$ part of the symmetry (37) acts on $\Phi$ in (33) and the charges as

$$
\begin{aligned}
\mathcal{R} &: \Phi \to -\Phi, \\
\mathcal{R} &: (Q_m, Q_w) \to (-Q_m, -Q_w).
\end{aligned}
\tag{41}
$$

The $U(1)_m \times U(1)_w$ symmetry has a mixed 't Hooft anomaly. A simple way to characterize the anomaly is to couple the system to background U(1) gauge fields $A^{(m)}$ and $A^{(w)}$. Then, the anomaly states that the partition function $\mathcal{Z}$ as a function of these background fields is not gauge invariant. It can be made gauge invariant by coupling the system to a 2+1d bulk, extending the gauge field to the bulk and adding a Chern-Simons term

$$
\frac{1}{2\pi} \int A^{(m)} dA^{(w)}.
\tag{42}
$$

The $\mathbb{Z}_2^{\mathcal{R}}$ symmetry generated by $\mathcal{R}$ can be included in the description (42) by enlarging the gauge group of $G_{c=1}$.

## 3.1 Generic twisting

We place the theory on a Euclidean torus and turn on background gauge fields for $G_{c=1}$. We will focus on the special case of flat gauge fields. Starting with the spatial direction, a background field corresponds to twisted boundary condition by a $G_{c=1}$ transformation. This group has two kinds of conjugacy classes. First, we can have a $\mathbb{Z}_2^{\mathcal{R}}$ twist. We will be mostly interested in twists in the other conjugacy classes corresponding to $U(1)_m \times U(1)_w$. We twist the spatial boundary conditions by[15]

$$
g(\eta_m, \eta_w) = e^{i\eta_m Q_m} e^{i\eta_w Q_w}.
\tag{43}
$$

This can also be written in the chiral basis (40) as

$$
\begin{aligned}
g(\eta_m, \eta_w) &= e^{i\eta Q} e^{i\overline{\eta}\,\overline{Q}}, \\
\eta &= R\eta_m + \frac{\eta_w}{R}, \\
\overline{\eta} &= R\eta_m - \frac{\eta_w}{R}.
\end{aligned}
\tag{44}
$$

---

[15]Recall that we denote the group elements associated with the spatial twists by $g$ and the group elements acting in that Hilbert space by $h$. We denote their representation in the twisted Hilbert space by $h(g)$.

Since the charges $Q_m$ and $Q_w$ are quantized and we have the $\mathbb{Z}_2^{\mathcal{R}}$ symmetry (41), then the parameters $(\eta_m, \eta_w)$ are subject to the identifications

$$(\eta_m, \eta_w) \sim (\eta_m + 2\pi, \eta_w) \sim (\eta_m, \eta_w + 2\pi) \sim (-\eta_m, -\eta_w) \,. \tag{45}$$

Let us study the theory quantized on $S^1$ twisted by $g(\eta_m, \eta_w)$. Using the known spectral flow transformations (see e.g., [48]), it is easy to track how the two charges, the energy, and the momentum change with the twist parameters. In the chiral basis, we have

$$
\begin{aligned}
Q(\eta) &= Q(0) + \frac{\eta}{4\pi} \,, \\
\overline{Q}(\overline{\eta}) &= \overline{Q}(0) - \frac{\overline{\eta}}{4\pi} \,, \\
L_0(\eta) &= L_0(0) + \frac{\eta}{2\pi} Q(0) + \left(\frac{\eta}{4\pi}\right)^2 \,, \\
\overline{L}_0(\overline{\eta}) &= \overline{L}_0(0) - \frac{\overline{\eta}}{2\pi} \overline{Q}(0) + \left(\frac{\overline{\eta}}{4\pi}\right)^2 \,,
\end{aligned}
\tag{46}
$$

and in the non-chiral basis

$$
\begin{aligned}
Q_m(\eta_w) &= R(Q(\eta) + \overline{Q}(\overline{\eta})) = Q_m(0) + \frac{\eta_w}{2\pi} \,, \\
Q_w(\eta_m) &= \frac{1}{R}(Q(\eta) - \overline{Q}(\overline{\eta})) = Q_w(0) + \frac{\eta_m}{2\pi} \,, \\
E(\eta_m, \eta_w) &= L_0(\eta) + \overline{L}_0(\overline{\eta}) \\
&= E(0,0) + \frac{1}{2\pi}\left(\frac{\eta_w Q_m(0)}{R^2} + R^2 \eta_m Q_w(0)\right) + \frac{1}{8\pi^2}\left(\frac{\eta_w^2}{R^2} + R^2 \eta_m^2\right) \,, \\
P(\eta_m, \eta_w) &= L_0(\eta) - \overline{L}_0(\overline{\eta}) \\
&= P(0,0) + \frac{1}{2\pi}\left(\eta_m Q_m(0) + \eta_w Q_w(0)\right) + \frac{\eta_m \eta_w}{4\pi^2} \,.
\end{aligned}
\tag{47}
$$

As is clear from these expressions, they are not invariant under the identifications (45). These identifications relate the Hilbert space as a whole but as we track the various energy eigenstates while $(\eta_m, \eta_w)$ vary, the states are not mapped to themselves. The Hamiltonian $H(\eta_m, \eta_w)$ is mapped to itself (or to an operator related to it by a unitary transformation), but the energy eigenvalues $E(\eta_m, \eta_w)$ are not. For this reason we distinguish the notation $H(\eta_m, \eta_w)$ from $E(\eta_m, \eta_w)$.

Similarly, $P(\eta_m, \eta_w)$ are the eigenvalues of the momentum operators. (In the CFT literature, it is common to refer to $P$ as the spin of the state.) And $Q_m(\eta_w)$ and $Q_w(\eta_m)$ are the eigenvalues of the charge operators. They are also not invariant under the identifications (45).

To summarize, twists by different $(\eta_m, \eta_w)$ subject to the identifications lead to isomorphic Hilbert spaces. However, it is natural to have the various operators, including the charges, depend on $(\eta_m, \eta_w)$ without the identifications. This will have consequences soon when we discuss how the group elements are realized in the twisted Hilbert space.

Next, we introduce the temporal background gauge fields. Since we limit ourselves to flat gauge fields, these can be thought of as twisted boundary conditions around the Euclidean time direction, or equivalently, as insertions of charge operators. This means that we study the partition function

$$\mathcal{Z}(\beta, \eta_m, \eta_w, \xi_m, \xi_w, \vartheta) = \mathrm{Tr}\left[e^{-\beta H(\eta_m, \eta_w)} e^{i\xi_m Q_m(\eta_w) + i\xi_w Q_w(\eta_m)} e^{i\vartheta P(\eta_m, \eta_w)}\right]. \tag{48}$$

The charges in these expressions are the charges in the twisted Hilbert space $Q_m(\eta_w)$ and $Q_w(\eta_m)$. The representations of the corresponding group elements (38) are

$$
\begin{aligned}
h_m(\xi_m; \eta_w) &= e^{i\xi_m Q_m(\eta_w)} \,, \\
h_w(\xi_w; \eta_m) &= e^{i\xi_w Q_w(\eta_m)} \,.
\end{aligned}
\tag{49}
$$

Let us review our notation. $h_m(\xi_m)$ and $h_w(\xi_w)$ are group elements. They are realized in the twisted Hilbert space by $h_m(\xi_m; \eta_w)$ and $h_w(\xi_w; \eta_m)$ respectively.[16] It is important that while the group elements $h_m(\xi_m)$ and $h_w(\xi_w)$ satisfy the group relations linearly, their representations $h_m(\xi_m; \eta_w)$ and $h_w(\xi_w; \eta_m)$ in general only realize them projectively. Furthermore, as we will soon discuss, these representations do not respect the identifications of $(\eta_m, \eta_w)$ and $(\xi_m, \xi_w)$ (45).

Let us discuss the parameters that the partition function $\mathcal{Z}$ depends on. Naively, the spatial twist parameters $\eta_m$ and $\eta_w$ are subject to the identifications (45). The temporal twist parameters $\xi_m$ and $\xi_w$ are subject to similar identifications. Also, the spatial shift parameter $\vartheta$ is identified with $\vartheta + 2\pi$. But this identification involves a shift of $\xi_m$ and $\xi_w$, which is related to the relation (25). We will discuss it in detail soon. In addition, the identifications of the circle-valued parameters may require an additional unitary transformation, which can affect the partition function. They are examples of the discussions in section 2.3.

An anomaly is the statement that the partition function $\mathcal{Z}$ is not a function on this parameter space, but a section of a line bundle. Explicitly, in the untwisted theory, $P(0,0), Q_m(0), Q_w(0) \in \mathbb{Z}$. Therefore,

$$
\begin{aligned}
\mathcal{Z}(\beta, \eta_m, \eta_w, \xi_m, \xi_w, \vartheta + 2\pi) &= e^{-\frac{i}{2\pi}\eta_m\eta_w} \mathcal{Z}(\beta, \eta_m, \eta_w, \xi_m + \eta_m, \xi_w + \eta_w, \vartheta), \\
\mathcal{Z}(\beta, \eta_m, \eta_w, \xi_m + 2\pi, \xi_w, \vartheta) &= e^{i\eta_w} \mathcal{Z}(\beta, \eta_m, \eta_w, \xi_m, \xi_w, \vartheta), \\
\mathcal{Z}(\beta, \eta_m, \eta_w, \xi_m, \xi_w + 2\pi, \vartheta) &= e^{i\eta_m} \mathcal{Z}(\beta, \eta_m, \eta_w, \xi_m, \xi_w, \vartheta).
\end{aligned}
\tag{50}
$$

The phases in these expressions reflect a mixed anomaly between $U(1)_m$ and $U(1)_w$. Indeed, when either $\eta_m = \xi_m = 0$ or $\eta_w = \xi_w = 0$, there is no phase. Also, we can redefine $\mathcal{Z}$ by a phase and move the anomalous phase around, but we cannot remove it completely. For example, if we multiply $\mathcal{Z}$ by $e^{-i\frac{\eta_w\xi_m}{2\pi}}$, then the phase in the second equation is absent, but then $\mathcal{Z}$ would not be invariant under $\eta_w \to \eta_w + 2\pi$.

Particularly interesting is the first equation in (50). It is related to the operator equation associated with complete translation of space

$$
\begin{aligned}
U(\eta_m, \eta_w) &= e^{2\pi i P(\eta_m, \eta_w)} = e^{-\frac{i}{2\pi}\eta_m\eta_w} g(\eta_m, \eta_w), \\
g(\eta_m, \eta_w) &= h_m(\eta_m; \eta_w) h_w(\eta_w; \eta_m) = e^{i\eta_m Q_m(\eta_w)} e^{i\eta_w Q_w(\eta_m)}.
\end{aligned}
\tag{51}
$$

The anomalous phase $e^{-\frac{i}{2\pi}\eta_m\eta_w}$ is an example of the phase in (31) in the continuum theory.

As always with anomalies, we can move the phases around. Specifically, we can remove it from (51) by redefining $g(\eta_m, \eta_w)$ by a phase. This means that the group element $g(\eta_m, \eta_w)$ is realized in the twisted Hilbert space by

$$
g'(\eta_m, \eta_w) = e^{-\frac{i}{2\pi}\eta_m\eta_w} e^{i\eta_m Q_m(\eta_w)} e^{i\eta_w Q_w(\eta_m)},
\tag{52}
$$

so that

$$
U(\eta_m, \eta_w) = g'(\eta_m, \eta_w)
\tag{53}
$$

is not projective. This redefinition amounts to changing the representations of the symmetry operators in the twisted Hilbert space (49). For example, we can keep $h_m(\xi_m, \eta_w) = e^{i\xi_m Q_m(\eta_w)}$ unchanged and redefine $h_w(\xi_w, \eta_m)$ to $h'_w(\xi_w, \eta_m) = e^{i\xi_w Q_w(0)}$ (which is independent of $\eta_m$).

$$
\begin{aligned}
g'(\eta_m, \eta_w) &= h_m(\eta_m, \eta_w) h'_w(\eta_w, \eta_m), \\
h_m(\eta_m, \eta_w) &= e^{i\eta_m Q_m(\eta_w)}, \\
h'_w(\eta_w, \eta_m) &= e^{i\eta_w Q_w(0)}.
\end{aligned}
\tag{54}
$$

---

[16]This is a special case of the generic $h(g)$ discussed above.

The advantage of this redefinition is that (53) does not have a phase. However, this redefinition treats the momentum and the winding symmetries differently and therefore it is not natural in this system. Below, we will see study lattice models that realize the full momentum symmetry, but only a discrete subgroup of the winding symmetry. Then, such a redefinition will be quite natural.

## 3.2  Pure $U(1)_m$ twist

In the special case $(\eta_m, \eta_w) = (\sigma, 0)$, the situation simplifies because $Q_m$ does not flow. Then, it is easy to work out the spectrum using Eq. (47)

$$
\begin{aligned}
E(\sigma,0) &= E(0,0) + R^2 \frac{\sigma}{2\pi} Q_w(0) + R^2 \frac{\sigma^2}{8\pi^2}\,, \\
P(\sigma,0) &= P(0,0) + \frac{\sigma}{2\pi} Q_m(0)\,.
\end{aligned}
\tag{55}
$$

Also, the phases in the first and second equations in (50) vanish. Consequently, there is no phase in the key expression (51)

$$
U(\sigma,0) = e^{2\pi i P(\sigma,0)} = g(\sigma,0) = e^{i\sigma Q_m(0)}\,.
\tag{56}
$$

The only anomaly is in the phase in the third equation in (50), which shows that in the twisted theory, the $U(1)_w$ charges are not integers. Related to that, the $U(1)_w$ transformation in the twisted Hilbert space (49)

$$
h_w(\xi;\sigma) = e^{i\xi Q_w(\sigma)}\,,
\tag{57}
$$

is not periodic in $\xi$ or $\sigma$

$$
\begin{aligned}
h_w(\xi + 2\pi;\sigma) &= e^{i\sigma} h_w(\xi;\sigma)\,, \\
h_w(\xi;\sigma + 2\pi) &= e^{i\xi} h_w(\xi;\sigma)\,.
\end{aligned}
\tag{58}
$$

This means that for nontrivial $\sigma$, $U(1)_w$ is realized projectively.

For generic $\sigma$, the discrete $\mathbb{Z}_2^{\mathcal{R}}$ symmetry of Eq. (41) is broken. It is preserved only at $\sigma \in \pi\mathbb{Z}$. However, there is still something nontrivial about it. Consider the $U(1)_w$ symmetry transformation (57). For $\sigma = 0$, it satisfies $h_w(\xi;0)\mathcal{R} = \mathcal{R}h_w(-\xi,0)$ and in particular, $h_w(\pi,0)\mathcal{R} = \mathcal{R}h_w(\pi;0)$. However, for $\sigma = \pi$, where $\mathcal{R}$ is also a symmetry, the eigenvalues of $Q_w(\pi)$ are half-integer,

$$
h_w(\xi;\pi) = e^{i\xi Q_w(\pi)} = e^{i\xi(Q_w(0)+\frac{1}{2})}\,,
\tag{59}
$$

and therefore

$$
h_w(\xi;\pi)\mathcal{R} = e^{i\xi}\mathcal{R}h_w(-\xi;\pi)\,,
\tag{60}
$$

and in particular

$$
h_w(\pi;\pi)\mathcal{R} = -\mathcal{R}h_w(\pi;\pi)\,.
\tag{61}
$$

We see that while the full symmetry group $G_{c=1}$ is preserved at $\sigma \in \pi\mathbb{Z}$ and it is realized linearly at $\sigma \in 2\pi\mathbb{Z}$, it is realized projectively at $\sigma \in \pi + 2\pi\mathbb{Z}$.

The theory also has a (spatial) parity $\mathcal{P}$ transformation. It acts as

$$
\begin{aligned}
\mathcal{P}E(\sigma,0)\mathcal{P}^{-1} &= E(-\sigma,0)\,, \\
\mathcal{P}P(\sigma,0)\mathcal{P}^{-1} &= -P(-\sigma,0)\,,
\end{aligned}
\tag{62}
$$

and therefore it is a good symmetry only for $\sigma \in \pi\mathbb{Z}$.

Combining $\mathcal{P}$ and $\mathcal{R}$, we find another parity-like transformation $\mathcal{P}' = \mathcal{P}\mathcal{R}$, which commutes with $E(\sigma, 0)$ for all values of $\sigma$.

These transformations $\mathcal{P}'$ and $\mathcal{R}$ allow more twists. First, for any $\sigma$, we can add $\mathcal{P}'$ in the trace (48). Second, for $\sigma \in \pi\mathbb{Z}$, we also have the symmetry $\mathcal{R}$ and then we can add $\mathcal{R}$ in the trace. (Recall however, that it can be realized projectively.) Finally, we can also twist in the space direction by $\mathcal{R}$. We will discuss some of these twists below.

### 3.3 Twist in $U(1)_m \times \mathbb{Z}_2^X \subset U(1)_m \times U(1)_w$

We now extend the previous discussion of a twist in $U(1)_m$ to a twist in $U(1)_m \times \mathbb{Z}_2^X$.

First, we parameterize the group elements as $X h_m(\xi_m)$ with

$$X = e^{i\pi Q_w} \tag{63}$$

the generator of $\mathbb{Z}_2^X \subset U(1)_w$. This means that we consider a twist by

$$(\eta_m, \eta_w) = (\sigma, \pi N), \qquad N \in \mathbb{Z}. \tag{64}$$

The group element $X$ depends only on $N$ mod 2 and it is trivial for even $N$. However, as we mentioned in section 2.3, the relation between different values of $N$ with the same $X$ might need a unitary transformation. We will see examples of this below. For this reason, we keep the expression with generic $N$.

One consequence of the mixed anomaly between the continuous $U(1)_m$ symmetry and this $\mathbb{Z}_2^X$ is that for odd $N$, the $U(1)_m$ charges $Q_m(\pi N) = \frac{N}{2} + \mathbb{Z}$ are half-integer.

Another consequence of this anomaly is that for nonzero $\sigma$, the winding charges $Q_w(\sigma) \in \frac{\sigma}{2\pi} + \mathbb{Z}$ are not integers. This affects the $\mathbb{Z}_2^X$ charge $X$ in the twisted Hilbert space. Naively, it is $h_w(\pi; \sigma) = e^{i\pi Q_w(\sigma)}$ (see (49)), which does not square to one. Soon, we will argue that it is more natural to define this charge as

$$X(\sigma, \pi N) = e^{i\pi Q_w(0)}, \tag{65}$$

such that it does squares to one for every $\sigma$.

Again, the spectrum can be worked out using Eq. (47)

$$
\begin{aligned}
E(\sigma, \pi N) &= E(0,0) + \frac{N Q_m(0)}{2R^2} + \frac{R^2\sigma}{2\pi} Q_w(0) + \frac{N^2}{8R^2} + \frac{R^2\sigma^2}{8\pi^2}, \\
P(\sigma, \pi N) &= P(0,0) + \frac{\sigma}{2\pi} Q_m(0) + \frac{N}{2} Q_w(0) + \frac{\sigma N}{4\pi}.
\end{aligned}
\tag{66}
$$

We see here an example of the phenomenon mentioned above. The group element $X$ depends only on $N$ mod 2, but the expressions for the energy and the momentum depend on $N$.

Now, all the anomalies in Eq. (50) can be nontrivial. In particular, the phase in (51) is nonzero. We can remove it by redefining $g$, as in Eq. (52)

$$g'(\sigma, N\pi) = e^{-\frac{i}{2\pi}\sigma\eta_w} e^{i\sigma Q_m(\eta_w)} e^{i\eta_w Q_w(\sigma)} = e^{i\pi N Q_w(0)} e^{i\sigma Q_m(\pi N)}, \tag{67}$$

and interpret it, as in (54). The second factor $h_m(\sigma; \pi N) = e^{i\sigma Q_m(\pi N)}$ represents the spatial twist in $U(1)_m$ using the twisted charge $Q_m(\pi N)$. The first factor is not the naive expression $h_w(\pi N; \sigma) = e^{i\pi N Q_w(\sigma)}$, but $h'_w(\sigma; \pi N) = e^{i\pi N Q_w(0)}$. We think of it as

$$X(\sigma, \pi N) = h'_w(\pi; \sigma) = e^{i\pi Q_w(0)}, \tag{68}$$

as defined in (65). This $X(\sigma, \pi N)$ satisfies (51) without an anomalous phase.[17]

---

[17]Recall our notation that $X$ is a group element, while $X(\sigma, \pi N)$ is its representation in the twisted Hilbert space.

Next, consider the $\mathbb{Z}_2^C \subset \mathrm{U}(1)_m \times \mathbb{Z}_2^X$ subgroup generated by the element

$$C = e^{i\pi Q_m + i\pi Q_w} = X h_m(\pi). \tag{69}$$

Group theoretically, we can consider the $\mathbb{Z}_2$ factor in $\mathrm{U}(1)_m \times \mathbb{Z}_2^X$ either as $X$ (as above), or as $C$. But the anomaly is different.

Let us demonstrate it by twisting by

$$g = C^N h_m(\sigma) = X^N h_m(\sigma + \pi N), \tag{70}$$

i.e., we study the spatial twist[18]

$$(\eta_m, \eta_w) = (\sigma + \pi N, \pi N) \qquad \text{with} \qquad N \in \mathbb{Z}. \tag{72}$$

It is straightforward to determine how $C$ acts in the twisted Hilbert space. Using the previous results

$$\begin{aligned} h_m(\sigma; \pi N) &= e^{i\sigma Q_m(\pi N)}, \\ X(\sigma + \pi N, \pi N) &= e^{i\pi Q_w(0)}, \end{aligned} \tag{73}$$

we have

$$\begin{aligned} C(\sigma + \pi N, \pi N) &= X(\sigma + \pi N, \pi N) h_m(\pi, \pi N) = e^{i\pi Q_w(0)} e^{i\pi Q_m(\pi N)} \\ &= i^N e^{i\pi (Q_w(0) + Q_m(0))}. \end{aligned} \tag{74}$$

This expression for the action of $C(\sigma + \pi N, \pi N)$ has a number of interesting properties. First, it does not vary with $\sigma$ and therefore it generates a finite group. Second, it satisfies

$$U(\sigma + \pi N, \pi N) = C(\sigma + \pi N, \pi N)^N e^{i\sigma Q_m(\pi N)}, \tag{75}$$

without an additional phase.

Interestingly,

$$C(\sigma + \pi N, \pi N)^2 = (-1)^N. \tag{76}$$

We see a nontrivial phase even in the simple case of $\sigma = 0$

$$U(\pi N, \pi N)^2 = (-1)^N. \tag{77}$$

It follows that for odd $N$, the momenta (spin) of the states are quantized to $\frac{\mathbb{Z}}{2} + \frac{1}{4}$. Therefore, this $\mathbb{Z}_2^C$ symmetry is realized projectively in the twisted Hilbert spaces with odd $N$. This is related to the fact that this $\mathbb{Z}_2^C$ symmetry has a pure anomaly associated with the nontrivial element of $H^3(\mathbb{Z}_2, \mathrm{U}(1)) = \mathbb{Z}_2$.

Another aspect of this anomaly is the phase $i^N$ in (74). It means that the behavior as a function of $N$ depends on $N$ mod 4 rather than on $N$ mod 2. A twist by $N = 2$ is not completely trivial and a twist by $N = -1$ is not the same as with $N = +1$. We will discuss this modulo 4 periodicity further in section 3.4.

---

[18]Since as group elements $C^2 = X^2 = 1$, we could consider

$$(\eta_m, \eta_w) = (\sigma + \pi N, \pi N + 2\pi K) \qquad \text{with} \qquad N, K \in \mathbb{Z}, \tag{71}$$

instead of (72). Repeating the calculation below, instead of (74) we end up with $C(\sigma + \pi N, \pi N + 2\pi K) = (-1)^K C(\sigma + \pi N, \pi N)$. This $K$ dependent sign does not affect our conclusions.

### 3.4 $R = 1$

In this section we will specialize to the case of $R = 1$, where the theory becomes the $SU(2)_1$ WZW CFT.

For this value of $R$, the global $G_{c=1} = (U(1)_m \times U(1)_w) \rtimes \mathbb{Z}_2^{\mathcal{R}}$ symmetry of the generic $R$ theory is enhanced to

$$SO(4) = (SU(2) \times \overline{SU(2)})/\mathbb{Z}_2. \tag{78}$$

This larger symmetry includes an element that exchanges $Q_m \leftrightarrow Q_w$ in $G_{c=1}$ and implements the self-duality of the model. (It is easy to see that this is an order 4, rather than an order 2, element.)

Recall that the Hilbert space of the $SU(2)_1$ WZW theory decomposes as

$$(\mathcal{V}_0 \otimes \overline{\mathcal{V}}_0) \oplus (\mathcal{V}_{\frac{1}{2}} \otimes \overline{\mathcal{V}}_{\frac{1}{2}}). \tag{79}$$

Here $\mathcal{V}_j \otimes \overline{\mathcal{V}}_{\bar{j}}$ refers to the chiral and anti-chiral Kac–Moody representation labeled by spins $(j, \bar{j})$.

Below, we will be interested in the $O(3) = SO(3) \times \mathbb{Z}_2^C \subset SO(4)$ symmetry. The momentum symmetry $U(1)_m \subset G_{c=1}$ is included in the $SO(3)$ part, which acts on $|j, \bar{j}\rangle$ as diagonal spin rotation. The $\mathbb{Z}_2^C$ generator commutes with $SO(3)$ and in fact, it commutes with the whole chiral algebra of the model. It is also included in $G_{c=1}$ as in Eq. (69), $C = (-1)^{Q_m + Q_w}$. (The group element $X$ in Eq. (65) does not commute with $SO(3)$.) $C$ acts as

$$C|j, \bar{j}\rangle = (-1)^{2j}|j, \bar{j}\rangle. \tag{80}$$

The $SO(4)$ symmetry has 't Hooft anomaly and it can be described, as in (42), by a 2+1d Chern-Simons term for $(SU(2)_1 \times SU(2)_{-1})/\mathbb{Z}_2$.

Below, we will focus on the 't Hooft anomaly of the $O(3) = SO(3) \times \mathbb{Z}_2^C$ subgroup of $SO(4)$. Here $\mathbb{Z}_2^C$ is the center element of $SO(4)$. In terms of background $O(3) \subset SO(4)$ gauge fields, the 2+1d Chern-Simons theory becomes

$$S = \frac{1}{2}\left(\int A \cup w_2(B) + \int A \cup A \cup A\right), \tag{81}$$

where $A$ is the background gauge field for the center $\mathbb{Z}_2^C$, and $B$ is the background field for $SO(3)$. Here we adopt the convention that the partition function is $e^{2\pi i S}$, and $A$ takes values in $\mathbb{Z}/2\mathbb{Z}$ (so is $w_2$). We will refer to the first term as a mixed $SO(3)/\mathbb{Z}_2^C$ anomaly and to the second term as a pure $\mathbb{Z}_2^C$ anomaly. The latter has been discussed in the context of spin chain in [19].

#### 3.4.1 Generic $SO(4)$ twist

We are going to explore the anomaly by applying twists in the spatial and temporal directions.

Even though we now have a larger global symmetry, $SO(4)$ rather than $G_{c=1} = (U(1)_m \times U(1)_w) \rtimes \mathbb{Z}_2^{\mathcal{R}}$, the possible twists are almost the same. A generic $SO(4)$ spatial twists can be conjugated to the generic $G_{c=1}$ twist $g(\eta_m, \eta_w) = e^{i\eta_m Q_m} e^{i\eta_w Q_w}$ (43), with only one difference. Conjugating by an $SO(4)$ element that exchanges $Q_m \leftrightarrow Q_w$ leads to another identification in (45)

$$(\eta_m, \eta_w) \sim (\eta_m + 2\pi, \eta_w) \sim (\eta_m, \eta_w + 2\pi) \sim (-\eta_m, -\eta_w) \sim (\eta_w, \eta_m), \tag{82}$$

or using chiral notation

$$(\eta, \overline{\eta}) \sim (\eta + 2\pi, \overline{\eta} + 2\pi) \sim (\eta + 4\pi, \overline{\eta}) \sim (\eta, \overline{\eta} + 4\pi) \sim (-\eta, \overline{\eta}) \sim (\eta, -\overline{\eta}), \tag{83}$$

where we added a redundant identification to make it look symmetric. The other difference following from the larger symmetry is that a twist by $\mathcal{R}$ of (41) should not be considered separately because it is conjugate to $g(\pi, 0)$.

The discussion of the anomalies is almost identical to that in section 3.1 and we will not repeat it. Instead, we will follow sections 3.2 and 3.3 and comment on special twists.

### 3.4.2  SO(3) twist

Here we specialize to twist in $SO(3) \subset SO(4)$. This corresponds to $(\eta_m, \eta_w) = (\sigma, 0)$ (or in chiral notation, $\eta = \bar{\eta} = \sigma$) with $\sigma \sim \sigma + 2\pi \sim -\sigma$. This situation is almost identical to the one in section 3.2.

When $\sigma$ is changed from 0 to $2\pi$, the two sectors of the Hilbert space in Eq. (79) $\mathcal{V}_0 \otimes \overline{\mathcal{V}}_0$ and $\mathcal{V}_{\frac{1}{2}} \otimes \overline{\mathcal{V}}_{\frac{1}{2}}$ are swapped. The points $\sigma \in \pi\mathbb{Z}$ are special. At $\sigma \in 2\pi\mathbb{Z}$, the full SO(3) symmetry is unbroken. And at $\sigma \in \pi(2\mathbb{Z}+1)$ there is an unbroken $O(2) = U(1) \rtimes \mathbb{Z}_2^{\mathcal{R}}$, where the additional $\mathbb{Z}_2^{\mathcal{R}}$ is generated by the SO(3) transformation $\mathcal{R}$ corresponding to $\pi$ rotation around another axis. As explained in section 3.2, this O(2) symmetry is realized projectively at that point. Both the $U(1) \subset O(2)$ is realize projectively and the $\mathbb{Z}_2^{\mathcal{R}} \subset O(2)$ acts on the U(1) elements projectively, as in Eq. (60).

For these values of the twist parameters, the anomaly phases in the first and second equations in Eq. (50) are absent. The anomaly phase in the third equation is also absent if we limit the temporal twist to be also in SO(3), i.e., $\xi_w = 0$. (This is the same as the situation with generic $R$ discussed around Eq. (56).) This is consistent with the vanishing of the 't Hooft anomaly of the SO(3) symmetry.

As in the discussion in section 3.2, here we can also study an insertion of the parity-like symmetry operator $\mathcal{P}' = \mathcal{R}\mathcal{P}$ in the trace. We will discuss this trace when we study this system on the lattice in section 5.

Also, for $\sigma = \pi$, we can also insert $\mathcal{R}$ in the trace. This object has a natural interpretation. Up to conjugation by SO(3), in terms of the three SO(3) generators $J_i$, the spatial twist is by the SO(3) group element $e^{i\pi J_3}$ and the temporal twist is by the SO(3) generator $\mathcal{R} = e^{i\pi J_1}$. This is the standard presentation of an SO(3) bundle with nontrivial second Stiefel–Whitney class. We will discuss this partition function when we study this system on the lattice in section 5.

### 3.4.3  $O(3) = SO(3) \times \mathbb{Z}_2^C$ twist

Unlike the previous case of pure SO(3) twist, here the $\mathbb{Z}_2^C$ is nontrivial because of the two anomalies in Eq. (81). Fortunately, we do not need to do more work here, because this problem is identical to the one in section 3.3. The only new elements follow from the enhanced non-Abelian symmetry at $R = 1$.

Let us first take a first look at a pure $\mathbb{Z}_2^C$ twist by $C$ (69). It is in the center of SO(4). It corresponds to $(\eta_m, \eta_w) = (\pi, \pi)$, or using chiral notation, $(\eta, \overline{\eta}) = (2\pi, 0)$. Since $Q(2\pi) = Q(0) + \frac{1}{2}, \overline{Q}(2\pi) = \overline{Q}(0)$ the Hilbert space in $(\mathcal{V}_0 \otimes \overline{\mathcal{V}}_0) \oplus (\mathcal{V}_{\frac{1}{2}} \otimes \overline{\mathcal{V}}_{\frac{1}{2}})$ of (79) is mapped to $(\mathcal{V}_{\frac{1}{2}} \otimes \overline{\mathcal{V}}_0) \oplus (\mathcal{V}_0 \otimes \overline{\mathcal{V}}_{\frac{1}{2}})$, which transforms projectively under the SO(3). This reflects the first term in the anomaly action (81), a mixed anomaly between $\mathbb{Z}_2^C$ and SO(3).

We can easily add a twist in SO(3) by using the results in section 3.3. The most significant result there was the action of $C$ in the twisted Hilbert space (74)

$$C(\sigma + \pi N, \pi N) = e^{i\pi(Q_w(0) + Q_m(\pi N))} = i^N e^{i\pi(Q_m(0) + Q_w(0))} = i^N e^{2\pi i Q(0)}, \tag{84}$$

and it has the same interesting properties we mentioned there.

First,

$$C(\sigma + \pi N, \pi N)^2 = (-1)^N, \tag{85}$$

and therefore in the twisted Hilbert spaces with odd $N$, $C$ realizes the $\mathbb{Z}_2^C$ symmetry projectively. Its square is $-1$ and its eigenvalues in the two sectors $(\mathcal{V}_0 \otimes \overline{\mathcal{V}}_{\frac{1}{2}})$ and $(\mathcal{V}_0 \otimes \overline{\mathcal{V}}_{\frac{1}{2}})$ are $\pm i$. This result has already appeared in [45, 46].

Second, naively, $C$ is a $\mathbb{Z}_2$ twist and therefore its eigenvalues should depend only on $N$ mod 2. Instead, they depend on $N$ mod 4. Using Eq. (84), we have

$$
\begin{aligned}
N = 0, && C\left(\mathcal{V}_0 \otimes \overline{\mathcal{V}}_0\right) &= +1, & C\left(\mathcal{V}_{\frac{1}{2}} \otimes \overline{\mathcal{V}}_{\frac{1}{2}}\right) &= -1, \\
N = 1, && C\left(\mathcal{V}_0 \otimes \overline{\mathcal{V}}_{\frac{1}{2}}\right) &= -i, & C\left(\mathcal{V}_{\frac{1}{2}} \otimes \overline{\mathcal{V}}_0\right) &= +i, \\
N = 2, && C\left(\mathcal{V}_0 \otimes \overline{\mathcal{V}}_0\right) &= -1, & C\left(\mathcal{V}_{\frac{1}{2}} \otimes \overline{\mathcal{V}}_{\frac{1}{2}}\right) &= +1, \\
N = 3, && C\left(\mathcal{V}_0 \otimes \overline{\mathcal{V}}_{\frac{1}{2}}\right) &= +i, & C\left(\mathcal{V}_{\frac{1}{2}} \otimes \overline{\mathcal{V}}_0\right) &= -i,
\end{aligned}
\tag{86}
$$

where the operator $C$ is $C(\sigma + \pi N, \pi N)$ and the sectors of the Hilbert space $\mathcal{V}_j \otimes \overline{\mathcal{V}}_{\bar{j}}$ are labeled by $(j, \bar{j})$ as natural after the twist by $(\pi N, \pi N)$. This is an example of the issue discussed in section 2.3.

Finally, it satisfies the relation between the momenta (spin) and the twist

$$
U(\sigma + \pi N, \pi N) = e^{2\pi i P(\sigma + \pi N, \pi N)} = C(\sigma + \pi N, \pi N)^N e^{i\sigma Q_m(\pi N)},
\tag{87}
$$

without a phase.

We end this section with Tables 1 and 2 listing the properties of the low-lying states in the twisted Hilbert spaces. They are obtained using

$$
\begin{aligned}
E(\sigma + \pi N, \pi N) &= E(\pi N, \pi N) + \frac{\sigma}{2\pi}\left[Q(2\pi N) - \overline{Q}(0)\right] + \frac{\sigma^2}{8\pi^2}, \\
P(\sigma + \pi N, \pi N) &= P(\pi N, \pi N) + \frac{\sigma}{2\pi}\left[Q(2\pi N) + \overline{Q}(0)\right],
\end{aligned}
\tag{88}
$$

or in non-chiral notation

$$
\begin{aligned}
E(\sigma + \pi N, \pi N) &= E(\pi N, \pi N) + \frac{\sigma}{2\pi}Q_w(\pi N) + \frac{\sigma^2}{8\pi^2}, \\
P(\sigma + \pi N, \pi N) &= P(\pi N, \pi N) + \frac{\sigma}{2\pi}Q_m(\pi N).
\end{aligned}
\tag{89}
$$

The information in the tables demonstrates the $\mathcal{P}'$ symmetry at every $\sigma$ and the enhanced $\mathcal{R}$ symmetry at $\sigma = \pi$.

# 4 Anomalies in lattice models

While 't Hooft anomalies are often discussed in continuum field theories, they also exist in lattice models. Suppose the symmetry group of the system under consideration is $G$, then for every element $g$, the corresponding symmetry transformation is implemented by a unitary operator $\rho(g)$. Since we are interested in local systems, we require that $\rho(g)$ maps local operators to local operators, i.e., they are locality-preserving unitary transformations.

It is natural to expect that the absence of 't Hooft anomaly should correspond to a very "local" form of symmetry transformation on the lattice. Indeed, for the so-called "on-site" symmetries (the precise meaning will be given below), the lattice model can be consistently coupled to gauge fields and there is no 't Hooft anomaly.

We now explain what it means for a locality-preserving unitary operator to be on-site, when the lattice system has a tensor product Hilbert space. If the unitary operator can be

Table 1: The low-lying states for even $N$. We started from the spectrum $(\mathcal{V}_0 \otimes \overline{\mathcal{V}}_0) \oplus (\mathcal{V}_{\frac{1}{2}} \otimes \overline{\mathcal{V}}_{\frac{1}{2}})$ and determined the two U(1) charges in the first column. The charges $Q$ and $\overline{Q}$ in that column are the left and right charges for $\sigma = 0$, i.e., $Q(2\pi N)$ and $\overline{Q}(0)$. Then, we used (84),(88),(89) to find the values in the other columns. States in rows that are not separated by a line remain degenerate for every $\sigma$, reflecting the $\mathcal{P}'$ parity symmetry of the model. Note that we did not include all the states that are degenerate with those in the table at $\sigma = 0$, but we did include the degenerate states at $\sigma = \pi$, where we can check the $\mathcal{R}$ symmetry at this point.

| $(Q, \overline{Q})$ | $C(\sigma + \pi N, \pi N)$ | $P(\sigma + \pi N, \pi N)$ | $E(\sigma + \pi N, \pi N)$ |
|---|---|---|---|
| $(0, 0)$ | $i^N$ | $0$ | $\frac{\sigma^2}{8\pi^2}$ |
| $(-\frac{1}{2}, \frac{1}{2})$ | $-i^N$ | $0$ | $\frac{1}{2} - \frac{\sigma}{2\pi} + \frac{\sigma^2}{8\pi^2}$ |
| $(\frac{1}{2}, \frac{1}{2})$ | $-i^N$ | $\frac{\sigma}{2\pi}$ | $\frac{1}{2} + \frac{\sigma^2}{8\pi^2}$ |
| $(-\frac{1}{2}, -\frac{1}{2})$ | $-i^N$ | $-\frac{\sigma}{2\pi}$ | $\frac{1}{2} + \frac{\sigma^2}{8\pi^2}$ |
| $(\frac{1}{2}, -\frac{1}{2})$ | $-i^N$ | $0$ | $\frac{1}{2} + \frac{\sigma}{2\pi} + \frac{\sigma^2}{8\pi^2}$ |
| $(-1, 0)$ | $i^N$ | $1 - \frac{\sigma}{2\pi}$ | $1 - \frac{\sigma}{2\pi} + \frac{\sigma^2}{8\pi^2}$ |
| $(0, 1)$ | $i^N$ | $-1 + \frac{\sigma}{2\pi}$ | $1 - \frac{\sigma}{2\pi} + \frac{\sigma^2}{8\pi^2}$ |

written as a tensor product of unitary operators, each of which acting on disjoint regions (i.e., local subsets of sites) of the system, then it is said to be an on-site unitary operator. It can also happen that a unitary operator becomes on-site after conjugation by a finite-depth local unitary circuit. In this case, we will also call the unitary operator on-site. For a symmetry group to be on-site, we require that each symmetry transformation in the group is on-site, and they form a linear (rather than projective) representation for any system size. Equivalently, it acts linearly (rather than projectively) on each factor in the tensor product Hilbert space.

The reason to consider the notion of on-site symmetry is that an on-site symmetry can always be gauged. In other words, there is no 't Hooft anomaly. The converse, i.e., a non-anomalous symmetry is always on-site, is also sometimes stated. However, we will discuss an example where a non-on-site symmetry is still non-anomalous.

Here the restriction to tensor product Hilbert space is important. In the examples below, we will study gauge theories in a Hamiltonian formalism. The system has a large Hilbert space $\mathcal{H}$, which is a tensor product of local Hilbert spaces. Then, the physical Hilbert space $\mathcal{H}_{\text{physical}}$ is the subspace of $\mathcal{H}$ satisfying Gauss's law. When the Gauss's law constraint acts on several overlapping factors in $\mathcal{H}$, the invariant subspace $\mathcal{H}_{\text{physical}}$ is not a tensor product Hilbert space. In the examples below, we will see anomalous symmetries whose transformations can be taken to act on-site in $\mathcal{H}$. But the action on $\mathcal{H}_{\text{physical}}$ is more complicated, partly because $\mathcal{H}_{\text{physical}}$ is not a tensor product Hilbert space. In other words, the anomaly disappears if the Gauss's law constraints are dropped.

Table 2: The low-lying states for odd $N$. We started from the spectrum $(\mathcal{V}_0 \otimes \overline{\mathcal{V}}_{\frac{1}{2}}) \oplus (\mathcal{V}_{\frac{1}{2}} \otimes \overline{\mathcal{V}}_0)$ and determined the two U(1) charges in the first column. The charges $Q$ and $\overline{Q}$ in that column are the left and right charges for $\sigma = 0$, i.e., $Q(2\pi N)$ and $\overline{Q}(0)$. Then, we used (84),(88),(89) to find the values in the other columns. States in rows that are not separated by a line remain degenerate for every $\sigma$, reflecting the $\mathcal{P}'$ parity symmetry of the model. Note that we did not include all the states that are degenerate with those in the table at $\sigma = 0$, but we did include the degenerate states at $\sigma = \pi$, where we can check the $\mathcal{R}$ symmetry at this point.

| $(Q,\overline{Q})$ | $C(\sigma + \pi N, \pi N)$ | $P(\sigma + \pi N, \pi N)$ | $E(\sigma + \pi N, \pi N)$ |
|---|---|---|---|
| $(-\frac{1}{2}, 0)$ | $i^N$ | $\frac{1}{4} - \frac{\sigma}{4\pi}$ | $\frac{1}{4} - \frac{\sigma}{4\pi} + \frac{\sigma^2}{8\pi^2}$ |
| $(0, \frac{1}{2})$ | $-i^N$ | $-\frac{1}{4} + \frac{\sigma}{4\pi}$ | $\frac{1}{4} - \frac{\sigma}{4\pi} + \frac{\sigma^2}{8\pi^2}$ |
| $(\frac{1}{2}, 0)$ | $i^N$ | $\frac{1}{4} + \frac{\sigma}{4\pi}$ | $\frac{1}{4} + \frac{\sigma}{4\pi} + \frac{\sigma^2}{8\pi^2}$ |
| $(0, -\frac{1}{2})$ | $-i^N$ | $-\frac{1}{4} - \frac{\sigma}{4\pi}$ | $\frac{1}{4} + \frac{\sigma}{4\pi} + \frac{\sigma^2}{8\pi^2}$ |
| $(-\frac{1}{2}, 1)$ | $i^N$ | $-\frac{3}{4} + \frac{\sigma}{4\pi}$ | $\frac{5}{4} - \frac{3\sigma}{4\pi} + \frac{\sigma^2}{8\pi^2}$ |
| $(-1, \frac{1}{2})$ | $-i^N$ | $\frac{3}{4} - \frac{\sigma}{4\pi}$ | $\frac{5}{4} - \frac{3\sigma}{4\pi} + \frac{\sigma^2}{8\pi^2}$ |

We note that [49] presented lattice models with anomalous symmetry acting on-site on a Euclidean spacetime lattice. The symmetry is still anomalous because the Lagrangian density is not invariant. We will discuss the Hamiltonian model of this system in section 4.1.

Just like in the continuum, 't Hooft anomaly of internal symmetry are diagnosed by studying symmetry defects and their properties under gauge transformations (i.e., local deformations), which can be implemented on the lattice. Such methods to compute 't Hooft anomaly for lattice systems with internal symmetry have been developed in [30] and [31].

Below, we will study two lattice systems. One of them, in section 4.1, has a $\left(\mathrm{U}(1)_m \times \mathrm{U}(1)_w\right) \rtimes \mathbb{Z}_2^{\mathcal{R}}$ global symmetry. And the other, in section 4.2, has a $\left(\mathrm{U}(1)_m \times \mathbb{Z}_2^X\right) \rtimes \mathbb{Z}_2^{\mathcal{R}}$ global symmetry. These symmetries are all or some of the symmetries $G_{c=1}$ of the $c = 1$ compact boson. We will compute the anomalies of the lattice systems and will compare with the continuum discussion in section 3. These examples will demonstrate how to apply our framework on the lattice, which will be further generalized to include lattice symmetries in sections 5 and 6. They will also demonstrate how the anomaly is related to lack of "on-site" action, following its definition above.

## 4.1 An example with anomalous $G = \left(\mathrm{U}(1)_m \times \mathrm{U}(1)_w\right) \rtimes \mathbb{Z}_2^{\mathcal{R}}$ internal symmetry

### 4.1.1 The system and its symmetries

The rotor model with the Hamiltonian (7) is the standard lattice construction of the $c = 1$ compact boson. The corresponding Euclidean spacetime lattice model is the famous XY model. The latter has a known Villain formulation, which uses a noncompact field $\phi$ at the sites and

a $\mathbb{Z}$ gauge field $n$ on the links. All these lattice models have only the $U(1)_m \rtimes \mathbb{Z}_2^{\mathcal{R}}$ symmetry. The $U(1)_w$ winding symmetry emerges only in the continuum limit.

The authors of [49–51] suggested a modification of the Euclidean spacetime lattice model by restricting the field strength of the $\mathbb{Z}$ gauge field to be zero. This lattice model shares many properties with its continuum limit. It has exact $U(1)_w$ symmetry and therefore it does not have a BKT transition. This $U(1)_w$ symmetry has a mixed anomaly with the $U(1)_m$ symmetry. And the model has exact T-duality exchanging these two symmetries.[19]

In this section, we will present a Hamiltonian formulation of this modified Villain model; i.e., space is a one-dimensional lattice and time is continuous. This lattice model flows to the $c = 1$ compact boson with any radius and without a BKT transition. It has the full $G_{c=1} = (U(1)_m \times U(1)_w) \rtimes \mathbb{Z}_2^{\mathcal{R}}$ symmetry with its anomaly and exact T-duality.

As in the Villain model, we start with a noncompact field at the sites $\phi_j$. Since we use a Hamiltonian formalism, we also have their conjugate momenta $p_j$ at the sites. The Hamiltonian is[20]

$$H_{\text{matter}} = \sum_j \left( \frac{U_0}{2} p_j^2 + \frac{J_0}{2} (\phi_{j+1} - \phi_j)^2 \right),$$

$$[\phi_j, p_{j'}] = i\delta_{j,j'}. \tag{90}$$

This Hamiltonian is similar to that of the rotor model Hamiltonian (7), except that $\phi_j$ is noncompact and the cosine potential was replaced by a harmonic potential. This model has an $\mathbb{R}$ global shift symmetry:

$$\phi_j \to \phi_j + \xi, \qquad \xi \in \mathbb{R}, \tag{91}$$

generated by $\sum_j p_j$. In addition there is a $\mathbb{Z}_2^{\mathcal{R}}$ symmetry that flips the signs of all $\phi_j$ and $p_j$.

In order to change the global symmetry group from $\mathbb{R}$ to $U(1)$, we gauge the $\mathbb{Z}$ symmetry generated by the $\xi = 2\pi$ shift. To do this we need to couple the model (90) to a $\mathbb{Z}$ gauge theory. The gauge field is an integer-valued field on the links $n_{j,j+1}$ and its conjugate momentum is the electric field, which is a phase $e^{iE_{j,j+1}}$ on the links[21]

$$[n_{j,j+1}, E_{j',j'+1}] = i\delta_{j,j'}. \tag{92}$$

This leads to the Hamiltonian

$$H_{\text{Villain}} = \sum_j \left( \frac{U_0}{2} p_j^2 + \frac{J_0}{2} (\phi_{j+1} - \phi_j - 2\pi n_{j,j+1})^2 \right) - g^2 \sum_j \left( e^{iE_{j,j+1}} + \text{h.c.} \right), \tag{93}$$

where $g$ is the gauge coupling constant. We can also add other periodic terms involving the electric field $E_{j,j+1}$ of the form $\sum_j \left( e^{iME_{j,j+1}} + \text{h.c.} \right)$ with integer $M$. We will discuss such terms below.

In addition, we need to impose the Gauss's law constraint

$$G_j = e^{i(E_{j,j+1} - E_{j-1,j})} e^{-2\pi i p_j} = 1. \tag{94}$$

---

[19]After the completion of this work the papers [52,53] discussed Hamiltonian formulations of various modified Villain models.

[20]This model is usually introduced in solid state physics as a simple model for phonons, where $\phi_j$ is the displacement of an atom at site $j$ from the lattice position. The $\mathbb{R}$ shift symmetry is the translation symmetry.

[21]Instead of thinking of the variable $n_{j,j+1}$ as a coordinate and the variable $E_{j,j+1}$ as a momentum, these variables might appear more standard if we think of the pair as describing a rotor. (Compare with the discussion around (5).) $E_{j,j+1}$ is a circle-valued coordinate on the links and $-n_{j,j+1}$ is its conjugate momentum, whose eigenvalues are quantized. This interpretation will be more natural when we discuss the T-duality of the model in section 4.1.3. Also, $E_{j,j+1}$ can be identified with the Lagrange multiplier field in [49,51] and in the continuum, with the Luttinger field $\Theta$ in the Lagrangian (34).

After gauging the $\mathbb{Z}$ symmetry, a shift by $\xi \in 2\pi\mathbb{Z}$ is a gauge transformation, so the global symmetry becomes compact and we denote it by $U(1)_m$. It is still generated by the charge

$$Q_m = \sum_j q_j^m,$$
$$q_j^m = p_j. \tag{95}$$

Here $q_j^m$ is the momentum charge density i.e., the temporal component of the momentum current. Another way to see that the symmetry group is $U(1)$ is to notice that

$$e^{2\pi i Q_m} = \prod_j e^{2\pi i p_j} = 1, \tag{96}$$

where the last equality is due to the Gauss's law constraint (94) and the periodic boundary conditions. Thus $Q_m$ takes integer values.

This model is the Hamiltonian formulation of the standard Villain version of the XY-model. For large $\frac{J_0}{U_0}$, it is essentially the same as the rotor model Hamiltonian (7) with $U \approx U_0$ and $J \approx J_0$ and it flows to the $c = 1$ compact boson with radius $R^4 \approx (2\pi)^2 \frac{J_0}{U_0}$. For smaller values of $\frac{J_0}{U_0}$ this relation is modified and eventually, at some value of $\frac{J_0}{U_0}$ corresponding to $R = 2$, it undergoes a BKT transition to a gapped phase. One way to understand it is to note that both the rotor model (7) and the Villain model (93) lack the winding symmetry and therefore the low-energy theory should include winding operators. For $R > 2$, the winding operators are irrelevant and the model is gapless. For $R < 2$, the model is gapped. (For a detailed discussion of the renormalization group flow in the presence of winding operators around the point $R = 2$, see [54].)

In this formulation, the modification of [49–51] corresponds to setting the gauge coupling constant $g$ to zero. This leads to the modified Villain Hamiltonian

$$H_{\mathrm{mV}} = \sum_j \left( \frac{1}{2R^2} p_j^2 + \frac{1}{2} \left( \frac{R}{2\pi} \right)^2 (\phi_{j+1} - \phi_j - 2\pi n_{j,j+1})^2 \right). \tag{97}$$

Here we replaced the coupling constants $U_0$ and $J_0$ with the expressions depending on $R$, which is the radius of the boson in the low-energy theory.[22]

Now, the model has another global symmetry $U(1)_w$ generated by the $\mathbb{Z}$-valued Wilson line

$$Q_w = -\sum_j n_{j,j+1} = \sum_j q_{j,j+1}^w,$$
$$q_{j,j+1}^w = \frac{1}{2\pi} \left( \phi_{j+1} - \phi_j - 2\pi n_{j,j+1} \right). \tag{98}$$

We interpret it as the winding symmetry. Note that the naive winding charge density $n_{j,j+1}$ does not commute with Gauss's law. Instead, we use $q_{j,j+1}^w$, which does commute with it. In the continuum limit, this charge density $q_{j,j+1}^w$ flows to $\frac{1}{2\pi}\partial_x \Phi$.

We conclude that the full global symmetry of the lattice model is $\left( U(1)_m \times U(1)_w \right) \rtimes \mathbb{Z}_2^{\mathcal{R}}$, exactly as in the continuum limit.

As in [49], it is clear that the two symmetries $U(1)_m$ and $U(1)_w$ have a mixed anomaly. The anomaly arises because the momentum charge density $q_j^m$ and the winding charge density $q_{j,j+1}^w$ do not act in the same local Hilbert space. $q_j^m$ acts on the sites, while $q_{j,j+1}^w$ acts on the

---

[22]In the continuum limit, this theory flows to the $c = 1$ compact boson discussed in section 3. The fields $\phi_j$ and $E_{j,j+1}$ flow to the continuum circle-valued fields $\Phi$ and $\Theta$ in the Lagrangian (34).

links and the neighboring sites. Indeed, the standard signal of the anomaly is the nontrivial commutator between the two charge densities (currents)

$$[q^w_{j,j+1}, q^m_{j'}] = \frac{i}{2\pi}\left(\delta_{j+1,j'} - \delta_{j,j'}\right).$$ (99)

This is the lattice version of the Schwinger term in the continuum theory. Each charge density commutes with the other charge, but not the other charge density. The anomaly is present in the commutator between the two densities.[23]

Finally, let us relate some deformations of this model to other models discussed in this paper. First, if we turn on the winding one operator $\left(e^{iE_{j,j+1}} + \text{h.c.}\right)$, i.e., we go back to the Villain version of the XY-model (93), the system exhibits a BKT transition at $R = 2$. If we do not add this deformation, but instead, we add the winding-two operator $\left(e^{2iE_{j,j+1}} + \text{h.c.}\right)$, we find a BKT transition at $R = 1$. This is the same as the XXZ model (8). (See section 1.3.) If we do not add these operators, but add only the winding-four operator $\left(e^{4iE_{j,j+1}} + \text{h.c.}\right)$, the BKT transition is at $R = \frac{1}{2}$, as in the gauged XXZ model of section 4.2. (See footnote 31.)

### 4.1.2 $L = 1, 2$

An important aspect of these lattice model is that for large $L$ they have local interactions. This locality underlies the relation to the continuum theory. As we mentioned in the introduction, it also leads to some interesting consequences, including the existence of defects and anomalies. Yet, it is amusing to consider simple cases with low values of $L$, as they provide explicit examples. Specifically, we will analyze the almost trivial cases of $L = 1, 2$. $L = 1$ corresponds to a single site, labeled by $j = 1$, and a single link, labeled by $(j, j+1) = (1, 2) \sim (1, 1)$. The Hamiltonian and Gauss's law are

$$\begin{aligned} H_{\text{mV}} &= \frac{1}{2R^2}p_1^2 + \frac{R^2}{2}n_{1,2}^2, \\ e^{2\pi i p_1} &= 1. \end{aligned}$$ (100)

Gauss's law means that the eigenvalues of $p_1$ are quantized, or equivalently, the field $\phi_1$ is circle valued. We end up with two decoupled rotors $(p_1, \phi_1)$ and $(n_{1,2}, -E_{1,2})$. The spectrum is labeled by two integers, $Q_m = p_1$ and $Q_w = -n_{1,2}$, and the energy levels are $E = \frac{1}{2R^2}Q_m^2 + \frac{R^2}{2}Q_w^2$. This matches the energies of the zero modes of the $c = 1$ compact boson.

In this simple case of $L = 1$, the anomaly (99) vanishes because there is no difference between the charge and the charge density.

For $L = 2$, we can change variables to

$$\begin{aligned} \phi_1 &= \Phi + \hat{\phi} - \frac{\pi}{2}\hat{n}, & \phi_2 &= \Phi - \hat{\phi} + \frac{\pi}{2}\hat{n}, \\ p_1 &= \frac{1}{2}(P + \hat{p}), & p_2 &= \frac{1}{2}(P - \hat{p}), \\ n_{1,2} &= \frac{1}{2}(\mathcal{N} + \hat{n}), & n_{2,1} &= \frac{1}{2}(\mathcal{N} - \hat{n}), \\ E_{1,2} &= E + \hat{E} + \frac{\pi}{2}\hat{p}, & E_{2,1} &= E - \hat{E} - \frac{\pi}{2}\hat{p}. \end{aligned}$$ (101)

They satisfy standard commutation relations:

$$[\Phi, P] = [\hat{\phi}, \hat{p}] = [\mathcal{N}, E] = [\hat{n}, \hat{E}] = i,$$ (102)

---

[23]We see here an example of a phenomenon we mentioned in the introduction of this section. If we ignore Gauss's law, we can take the winding density to be $n_{j,j+1}$ and it acts on-site (more precisely, "on-link"). Then, the commutator (99) vanishes and we can say that there is no anomaly. However, restricting the theory to respect Gauss's law, we have to take $q^w_{j,j+1}$ as above, and then the anomaly is present.

and are subject to the identifications

$$(E, \hat{E}) \sim (E + 2\pi, \hat{E}) \sim (E, \hat{E} + 2\pi) \sim (E + \pi, \hat{E} + \pi). \tag{103}$$

In terms of these variables, the Hamiltonian and Gauss's law become

$$
\begin{aligned}
H_{\text{mV}} &= \frac{1}{4R^2} P^2 + \frac{R^2}{4} \mathcal{N}^2 + \frac{1}{4R^2} \hat{p}^2 + \left( \frac{R}{\pi} \right)^2 \hat{\phi}^2, \\
e^{2\pi i P} &= 1, \\
e^{2i\hat{E}} &= e^{\pi i P}.
\end{aligned}
\tag{104}
$$

This means that the eigenvalues of $P$ are quantized and they determine the eigenvalues of $\hat{E}$ up to a shift by $\pi$. Using (103), this shift can be absorbed in a shift of $E$ by $\pi$. Hence, $\hat{E}$ and $\hat{n}$ can be ignored.

We end up with three decoupled systems: two rotors $(P, \Phi)$ and $(\mathcal{N}, -E)$, and a harmonic oscillator $(\hat{p}, \hat{\phi})$.

The spectrum is determined by two integers $Q_m = P$ and $Q_w = -\mathcal{N}$ and the level of the harmonic oscillator. This is exactly as expected in the continuum theory, except that the continuum theory has an infinite number of oscillators. In comparing with the continuum results, note that the eigenvalues of $L_0 + \bar{L}_0$ are $L$ times the eigenvalues of the lattice Hamiltonian. This explains the factor of two difference between the coefficients of the zero-mode terms in (100) and (104).

Let us see the anomaly in this simple case. The momentum and winding densities (95) and (98) are

$$
\begin{aligned}
q_1^m &= \frac{1}{2}(P + \hat{p}), \qquad q_2^m = \frac{1}{2}(P - \hat{p}), \\
q_{1,2}^w &= -\frac{1}{\pi} \hat{\phi} - \frac{1}{2} \mathcal{N}, \qquad q_{2,1}^w = \frac{1}{\pi} \hat{\phi} - \frac{1}{2} \mathcal{N}.
\end{aligned}
\tag{105}
$$

These densities do not commute, thus reflecting the anomaly. The reason to study these particular linear combinations of the degrees of freedom might seem strange. However, these expressions are the ones that are local and gauge invariant for larger $L$.

This demonstrates an important point we mentioned in the introduction. The 1+1d anomaly depends on viewing the system as 1+1 dimensional and imposing locality in space. This is artificial in the trivial case of $L = 2$, but it is quite natural for larger values of $L$. Then, we have a particular form of the charges and the charge densities, those in (95) and (98), which become (105) for $L = 2$. Related to this locality is the existence of defects, which we will discuss it in section 4.1.4.

### 4.1.3 T-duality

This model has exact T-duality [49–51]. In this Hamiltonian formalism, we write the implicit change of variables

$$
\begin{aligned}
p_j &= \frac{1}{2\pi} \left( \tilde{\phi}_{j,j+1} - \tilde{\phi}_{j-1,j} - 2\pi \tilde{n}_j \right), \\
\tilde{p}_{j,j+1} &= \frac{1}{2\pi} \left( \phi_{j+1} - \phi_j - 2\pi n_{j,j+1} \right), \\
e^{iE_{j,j+1}} &= e^{i\tilde{\phi}_{j,j+1}}, \\
e^{i\tilde{E}_j} &= e^{i\phi_j}.
\end{aligned}
\tag{106}
$$

Here, $\tilde{\phi}_{j,j+1}$ and $\tilde{p}_{j,j+1}$ are real conjugate variables and $\tilde{E}_j \in S^1$ is conjugate to $\tilde{n}_j \in \mathbb{Z}$. With these variables, the Gauss's law constraint (94) is satisfied automatically. The reason for that is that the new variables (with tilde) are gauge invariant under the original $\mathbb{Z}$ gauge symmetry. Instead, they have their own $\mathbb{Z}$ gauge symmetry (under which the original variables are invariant) and therefore we need to impose a new Gauss's law

$$\tilde{G}_{j,j+1} = e^{i(\tilde{E}_{j+1}-\tilde{E}_j)}e^{-2\pi i \tilde{p}_{j,j+1}} = 1 \,. \tag{107}$$

With these variables, the Hamiltonian becomes

$$H_{\mathrm{mV}} = \sum_j \left( \frac{1}{2} \frac{1}{(2\pi R)^2} \left( \tilde{\phi}_{j,j+1} - \tilde{\phi}_{j-1,j} - 2\pi \tilde{n}_j \right)^2 + \frac{1}{2} R^2 \tilde{p}_j^2 \right) \,, \tag{108}$$

i.e., $R$ is mapped to $\frac{1}{R}$. Similarly, we have $Q_m = \tilde{Q}_w$ and $Q_w = \tilde{Q}_m$. Corresponding to that, the operator $e^{i\tilde{E}_j} = e^{i\phi_j}$ has $Q_m = 1$ and $Q_w = 0$ and the operator $e^{iE_{j,j+1}} = e^{i\tilde{\phi}_{j,j+1}}$ has $Q_w = 1$ and $Q_m = 0$.

Note that unlike the continuum theory, the lattice theory does not have an enhanced SU(2) global symmetry at the selfdual point $R = 1$. This can be seen clearly in the special cases of $L = 1, 2$, which we discussed in section 4.1.2.

### 4.1.4 The defects

Let us consider defects and study the operator algebra in their presence.

A winding twist by $\eta_w$ can be introduced by modifying Gauss's law at one site $J$ to be $G_J = e^{-i\eta_w}$. Following the computation in (96), this leads to $e^{2\pi i Q_m} = \prod_j e^{2\pi i p_j} = e^{i\eta_w}$, i.e., the momentum charge $Q_m$ (95) has a non-integer part $\frac{\eta_w}{2\pi}$. Alternatively, we can restore $G_J = 1$ by shifting at that site $p_J \to p_J + \frac{\eta_w}{2\pi}$. Below, we will follow this presentation. Then, the defect appears as a change in the Hamiltonian at the site $J$

$$H_{\mathrm{mV}}^{(J)}(\eta_w) = \sum_{j \neq J} \frac{1}{2R^2} p_j^2 + \frac{1}{2R^2} \left( p_J + \frac{\eta_w}{2\pi} \right)^2 + \sum_j \frac{1}{2} \left( \frac{R}{2\pi} \right)^2 (\phi_{j+1} - \phi_j - 2\pi n_{j,j+1})^2 \,. \tag{109}$$

The Hamiltonian $H_{\mathrm{mV}}^{(J)}(\eta_w)$ depends on a real, rather than on a circle-valued $\eta_w$. However, the shift $\eta_w \to \eta_w + 2\pi$ is implemented by a unitary transformation

$$H_{\mathrm{mV}}^{(J)}(\eta_w) = e^{i\phi_J} H_{\mathrm{mV}}^{(J)}(\eta_w + 2\pi) e^{-i\phi_J} \,, \tag{110}$$

highlighting the fact that the twisted theory depends only on $e^{i\eta_w}$.

It is nice to compare the defect with the discussion of the particle on a ring around (5). The shift of the momentum by the parameter $\frac{\eta_w}{2\pi}$ is analogous to the shift by $\frac{\theta}{2\pi}$ there. And the periodicity of the parameter is achieved by using a unitary transformation (6).

This redefinition of $p_J$ modifies the expression for the momentum charge density (95) to

$$Q_m(\eta_w) = \sum_j q_j^m(\eta_w) \,,$$

$$q_j^m(\eta_w) = p_j + \frac{\eta_w}{2\pi} \delta_{j,J} \,. \tag{111}$$

It is important that the differences between the charge densities $q_j^m(0) = p_j$ and $q_j^m(\eta_w)$ and similarly the total charges $Q_m(0)$ and $Q_m(\eta_w)$ are in additive c-numbers, which do not affect any commutation relation. It affects only the charge assignment of states in the Hilbert space. The quantized $Q_m(0)$ is still a conserved charge and we can continue to use it. However,

as we will see, there are advantages in using the spectral flowed charge $Q_m(\eta_w)$. For example, it is consistent with the unitary transformation (110) under shifting $\eta_w$ by $2\pi$.

This defect is topological. This can be seen by noting that a shift of the defect from $J$ to $J+1$ is implemented by conjugating this Hamiltonian by a unitary transformation[24]

$$H_{\mathrm{mV}}^{(J)}(\eta_w) = U_w(J, J+1) H_{\mathrm{mV}}^{(J+1)}(\eta_w) U_w(J, J+1)^{-1},$$
$$U_w(J, J+1) = \exp\left(i\eta_w q_{J,J+1}^w\right). \tag{112}$$

(Here we assumed for simplicity that there is no momentum defect in the interval $(J, J+1)$.)

Similarly, a momentum defect associated with a twist by $\eta_m$ at the link $(J, J+1)$ is introduced by changing the variable $n_{J,J+1}$ at that link to be $n_{J,J+1} \in -\frac{\eta_m}{2\pi} + \mathbb{Z}$. (This means that the wave function of $E_{J,J+1}$ is not single-valued under $E_{J,J+1} \to E_{J,J+1} + 2\pi$, but it is a section of a line bundle.) Alternatively, as for the winding defect, we can keep the variables unchanged, i.e., $n_{J,J+1} \in \mathbb{Z}$ and introduce the defect by changing one link term in the Hamiltonian

$$H_{\mathrm{mV}}^{(J,J+1)}(\eta_m) = \sum_j \frac{1}{2R^2} p_j^2 + \sum_{j \neq J} \frac{1}{2}\left(\frac{R}{2\pi}\right)^2 (\phi_{j+1} - \phi_j - 2\pi n_{j,j+1})^2$$
$$+ \frac{1}{2}\left(\frac{R}{2\pi}\right)^2 (\phi_{J+1} - \phi_J - 2\pi n_{J,J+1} + \eta_m)^2. \tag{113}$$

We will use this presentation with the modified Hamiltonian.

As for the winding defect, $H_{\mathrm{mV}}^{(J,J+1)}(\eta_m)$ and $H_{\mathrm{mV}}^{(J,J+1)}(\eta_m + 2\pi)$ are related by a unitary transformation

$$H_{\mathrm{mV}}^{(J,J+1)}(\eta_m) = e^{iE_{J,J+1}} H_{\mathrm{mV}}^{(J,J+1)}(\eta_m + 2\pi) e^{-iE_{J,J+1}}. \tag{114}$$

Again, it is nice to compare this with the discussion of the particle on a ring around (5).

Here, it is natural to redefine the winding charge (98) to

$$Q_w(\eta_m) = \sum_j q_{j,j+1}^w(\eta_m),$$
$$q_{j,j+1}^w(\eta_m) = \frac{1}{2\pi}\left(\phi_{j+1} - \phi_j - 2\pi n_{j,j+1}\right) + \frac{\eta_m}{2\pi}\delta_{j,J}. \tag{115}$$

As with our comment about the momentum charge in the presence of the winding defect (111), we could use the conserved charge $Q_w(0)$. The difference between them is only in the action on the states.

Again, this defect is topological. It is shifted from the link $(J-1, J)$ to the link $(J, J+1)$ using conjugation

$$H_{\mathrm{mV}}^{(J-1,J)}(\eta_m) = U_m(J) H_{\mathrm{mV}}^{(J,J+1)}(\eta_m) U_m(J)^{-1},$$
$$U_m(J) = \exp\left(i\eta_m q_J^m\right). \tag{116}$$

The definitions (112) and (116) could be modified by multiplying them by phases without affecting the result of the conjugation. For example, we can use $q_{j,j+1}^w(\eta_m)$ in (112) and $q_j^m(\eta_w)$ in (116). Soon, we will see the significance of this comment.

So far we considered the case of a single momentum defect or a single winding defect. If there are several momentum defects or several winding defects, they can be shifted and merged using repeated applications of (112) and (116). However, when both kinds of defects are present we can see a sign of the anomaly. Consider a winding defect with $\eta_w$ at $J$ and

---

[24]In the presentation before the shift of the momentum, we need to conjugate the system by $\exp\left(-i\eta_w n_{J,J+1}\right)$. This does not change the Hamiltonian, but accounts for the change in Gauss's law.

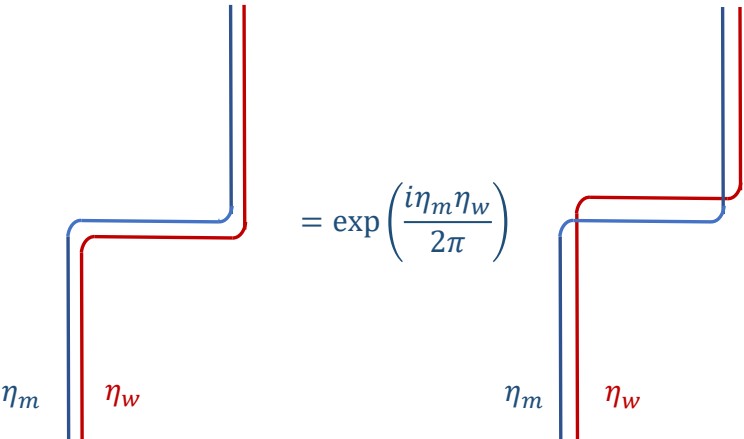

Figure 2: A momentum defect (in blue) and a winding defect (in red) are shifted to the right in two different orders. These two operations do not commute and differ by a phase, as in equation (117).

a momentum defect with $\eta_m$ at $(J', J' + 1)$, the shift operations (112) and (116) can still be used, but now they do not commute

$$U_m(J')U_w(J, J+1) = \exp\left(\frac{i\eta_m\eta_w}{2\pi}(\delta_{J+1,J'} - \delta_{J,J'})\right)U_w(J, J+1)U_m(J'),\tag{117}$$

where the phase follows from the anomalous commutator (99).[25] This lack of commutativity in shifting the defects is depicted in figure 2.

Next, we consider the translation symmetry. Without defects, we have a $\mathbb{Z}_L$ translation operator $T$, which shifts the lattice. It satisfies

$$\begin{aligned} T\mathcal{O}_j T^{-1} &= \mathcal{O}_{j+1}, \\ T^L &= 1, \end{aligned}\tag{118}$$

for any localized operator $\mathcal{O}_j$ around the site $j$.

$T$ does not commute with the Hamiltonian when defects are present. However, the system is still translational invariant. To see that, consider having a momentum defect at the link $(J, J + 1)$ and a winding defect at the site $J'$. Then

$$\begin{aligned} H_{\mathrm{mV}}^{(J,J+1),(J')} &= T(\eta_m, \eta_w) H_{\mathrm{mV}}^{(J,J+1),(J')} T(\eta_m, \eta_w)^{-1}, \\ T(\eta_m, \eta_w) &= U_m(J+1)U_w(J', J'+1)T = \exp\left(i\eta_m q_{J+1}^m\right)\exp\left(i\eta_w q_{J',J'+1}^w\right)T. \end{aligned}\tag{119}$$

Note that because of (117), for $J' = J$ or $J' = J + 1$, the operator $T(\eta_m, \eta_w)$ depends on the ordering of the two $U$'s. This leads to a c-number phase ambiguity. More generally, we can redefine $T(\eta_m, \eta_w)$ by an arbitrary c-number phase that depends on $\eta_m$ and $\eta_w$ without changing the action of $T(\eta_m, \eta_w)$ on $H_{\mathrm{mV}}^{(J,J+1),(J')}$. Below we will use this phase freedom.

---

[25]Here we used the moving operators as if the other defect is not present, i.e., $U_w(J, J+1) = \exp\left(i\eta_w q_{J,J+1}^w(0)\right)$ and $U_m(J') = \exp\left(i\eta_m q_{J'}^m(0)\right)$. If instead we wanted to use the shifted $q_{J,J+1}^w$ and $q_{J'}^m$, as in (115) and (111), this would not have been a commutator of two equivalent operators.

This translation operator satisfies

$$
\begin{aligned}
T(\eta_m, \eta_w)^L &= \prod_{j=1}^{L} \Big( \exp\big(i\eta_m q_{j+J}^m\big) \exp\big(i\eta_w q_{j+J'-1,j+J'}^w\big) \Big) \\
&= \exp\left(\frac{i\eta_m \eta_w}{2\pi}\right) \exp\left(i\eta_m \sum_j q_j^m\right) \exp\left(i\eta_w \sum_j q_{j,j+1}^w\right) \\
&= \exp\left(\frac{i\eta_m \eta_w}{2\pi}\right) \exp\left(i\eta_m Q_m(0)\right) \exp\left(i\eta_w Q_w(0)\right) \\
&= \exp\left(-\frac{i\eta_m \eta_w}{2\pi}\right) \exp\left(i\eta_m Q_m(\eta_w)\right) \exp\left(i\eta_w Q_w(\eta_m)\right) \\
&= \exp\left(i\eta_m Q_m(\eta_w)\right) \exp\left(i\eta_w Q_w(0)\right).
\end{aligned}
\tag{120}
$$

Here we used $T^L = 1$ and (99). The various expressions are similar to the discussion around (51)-(54). If we define the action of $\exp(i\eta_m Q_m)$ and $\exp(i\eta_w Q_w)$ as in the untwisted theory, $\eta_m = \eta_w = 0$, or as in the twisted theory, with the appropriate values of $\eta_m$ and $\eta_w$, we have an anomalous phase $\exp\left(\pm \frac{i\eta_m \eta_w}{2\pi}\right)$. This phase can be avoided by using $\exp(i\eta_m Q_m(\eta_w))$ and $\exp(i\eta_w Q_w(0))$. This choice is not symmetric, but is convenient in system where the $U(1)_m$ symmetry is present and only a discrete subgroup of $U(1)_w$ is present. (Compare with the discussion leading to (54).)

The rest of the analysis of the defects is as in section 3. In fact, since this model is the Hamiltonian version of the Lagrangian model in [49–51], this computation is conceptually the same as the discussion of the anomaly in [49].

Finally, it is amusing to consider these defects in the simple cases of small $L$ in section 4.1.2. For $L = 1$, there are no defects. But for $L = 2$ we can have momentum and winding defects. More precisely, in order to discuss defects, we should have a clear notion of locality. This is meaningful for large $L$. However, even in this simple case of $L = 2$, we can still discuss the defects of the larger $L$ problems and restrict them to $L = 2$.

Introducing a winding defect at the site 1 as in (109) and a momentum defect at the link $(1,2)$ as in (113) and changing variables as in (101) leads to the Hamiltonian

$$
\begin{aligned}
H_{\mathrm{mV}}^{(1,2),(1)} ={}& \frac{1}{4R^2} P^2 + \frac{R^2}{4} \mathcal{N}^2 + \frac{1}{4R^2}\left(\hat{p} + \frac{\eta_w}{2\pi}\right)^2 + \left(\frac{R}{\pi}\right)^2 \left(\hat{\phi} - \frac{\eta_m}{4}\right)^2 \\
&+ \frac{1}{4\pi}\left(\frac{\eta_w P}{R^2} - R^2 \eta_m \mathcal{N}\right) + \frac{1}{16\pi^2}\left(\left(\frac{\eta_w}{R}\right)^2 + (R\eta_m)^2\right).
\end{aligned}
\tag{121}
$$

The constant shifts of $\hat{\phi}$ and $\hat{p}$ do not affect the energy. This makes it clear that terms in the last line match (47) (recall that we have a factor of $L = 2$ in the energy, $Q_m = P$ and $Q_w = -\mathcal{N}$).

Let us discuss the translation symmetry of this system. The operator $T$ exchanges the values of the subscripts $1 \leftrightarrow 2$. (In this particular case of $L = 2$ it acts like space reflection.) Its action on the hatted variables is $\hat{\phi} \to -\hat{\phi}$ and $\hat{p} \to -\hat{p}$. (It also flips the sign of $\hat{n}$ and $\hat{E}$, but they have already been eliminated.) This is not a symmetry of the Hamiltonian (121). This fact can be corrected in various ways. Here we follow the discussion at larger $L$ and take, as in (119)

$$
T(\eta_m, \eta_w) = e^{i\eta_m p_2} e^{\frac{i\eta_w}{2\pi}(\phi_2 - \phi_1 - 2\pi n_{1,2})} T = e^{\frac{i\eta_m}{2}(P-\hat{p})} e^{-i\eta_w\left(\frac{\mathcal{N}}{2} + \frac{\hat{\phi}}{\pi}\right)} T.
\tag{122}
$$

One might want to modify it by combining it with a symmetry transformation, e.g., $e^{-\frac{i\eta_m}{2} P} e^{\frac{i\eta_w}{2} \mathcal{N}}$. However, this will be incompatible with locality of the larger $L$ problem: the action of $T(\eta_m, \eta_w)$ on operators in regions far from the defect should be identical to that of $T$. This demonstrate our comment after (105) about the importance of locality in the form of the operators.

## 4.2 An example with anomalous $G = (\mathrm{U}(1)_m \times \mathbb{Z}_2^X) \rtimes \mathbb{Z}_2^{\mathcal{R}}$ internal symmetry

In this section, we discuss another example, closely related to the system in section 4.1, but has a spin-$\frac{1}{2}$ degree of freedom on every site. This means that the local Hilbert space is finite dimensional rather than infinite dimensional as in section 4.1. Instead of the full $(\mathrm{U}(1)_m \times \mathrm{U}(1)_w) \rtimes \mathbb{Z}_2^{\mathcal{R}}$ symmetry, only a subgroup $(\mathrm{U}(1)_m \times \mathbb{Z}_2^X) \rtimes \mathbb{Z}_2^{\mathcal{R}}$ is realized exactly on the lattice. Here the $\mathbb{Z}_2^X$ factor corresponds to the subgroup generated by $\mathrm{U}(1)_w$ rotation by $\pi$ in the continuum. Similar models have been studied in [29, 32, 55–57].

### 4.2.1 The system and its symmetries

First, let us describe the model. Each site has a two-dimensional local Hilbert space (i.e., a qubit), and we denote the Pauli operators acting on site $j$ by $\sigma_j^a, a = x, y, z$. Throughout this section we will be only considering a chain with (possibly twisted) periodic boundary condition, and the number of sites $L$ of the chain will be a multiple of 4. We will discuss other values of $L$ in section 6.2.

The $\mathbb{Z}_2^X$ symmetry is generated by

$$X = \prod_j \sigma_j^x. \tag{123}$$

It is clearly an on-site symmetry.

The U(1) symmetry is generated by the following charge

$$\mathcal{Q} = \frac{1}{4} \sum_j \sigma_j^z \sigma_{j+1}^z. \tag{124}$$

The normalization with a factor of $\frac{1}{4}$ is such that $\mathcal{Q}$ takes integer values for $L = 0 \bmod 4$.[26]

It is important that $\mathcal{Q}$ is not an "on-site" symmetry, but the U(1) symmetry generated by $\mathcal{Q}$ has no 't Hooft anomaly. As we will see below, while the $\mathbb{Z}_2^X$ or U(1) are anomaly-free individually, there is a mixed anomaly between them. This also means that the $\mathbb{Z}_2^C$ subgroup generated by

$$C = X e^{i\pi \mathcal{Q}} \tag{125}$$

is anomalous, which was demonstrated explicitly in [30] and [31].

It will be extremely useful, especially for the study of symmetry defects, to introduce a dual presentation of the system, in terms of "domain wall" variables. Following [32], we define the map:[27]

$$\begin{aligned} \sigma_j^x &= \mu_{j-1,j}^z \tau_{j-1}^x \tau_j^x, \\ \sigma_j^y &= -\mu_{j-1,j}^y \tau_{j-1}^x \tau_j^x, \\ \sigma_j^z &= \mu_{j-1,j}^x. \end{aligned} \tag{126}$$

Here $\mu_{j,j+1}^z$ are $\mathbb{Z}_2$ gauge fields. The gauge symmetry is associated with the Gauss's law constraint:

$$\tau_j^z = \mu_{j-1,j}^x \mu_{j,j+1}^x, \tag{127}$$

which implies $\prod_j \tau_j^z = 1$.

---

[26] Our definition differs from the one in [32], where the U(1) charge is defined as $Q_{\mathrm{LG}} = \frac{1}{2} \sum_j \frac{1-\sigma_j^z \sigma_{j+1}^z}{2} = -\mathcal{Q} + \frac{L}{4}$. We choose this definition to make the connection with the continuum theory more straightforward.

[27] As common in duality, we can think of the new degrees of freedom as taking value in the dual lattice. The site variables $\sigma_j$ are mapped to link variables $\mu_{j-1,j}$.

In this representation, the symmetry generators become

$$X = \prod_j \mu^z_{j,j+1},$$
$$\mathcal{Q} = \frac{1}{4} \sum_j \tau^z_j. \tag{128}$$

We see that $X$ is the Wilson loop, measuring holonomy of the $\mathbb{Z}_2$ gauge field. $\mathcal{Q}$ satisfies the following operator identity:

$$e^{2\pi i \mathcal{Q}} = i^L \prod_j \tau^z_j = i^L = 1, \tag{129}$$

which re-affirms the fact that with periodic boundary conditions, $\mathcal{Q}$ is integer-valued. In addition, the expression of $\mathcal{Q}$ in terms of $\tau^z$ implies that the (gauge-non-invariant) $\tau^\pm_j$ operators carry charges $\pm\frac{1}{2}$ under the U(1) symmetry.

Naively, $X$ and $\mathcal{Q}$ are both "on-site" in the $\tau, \mu$ representation. However, the $\tau, \mu$ variables are redundant and the physical states are only obtained after enforcing the Gauss's law constraints. This fact is similar to an analogous fact discussed around equation (98).

In addition, it will be useful to define a $\mathbb{Z}_2^{\mathcal{R}}$ symmetry $\mathcal{R}$:

$$\mathcal{R} = \prod_{j=1}^{L/2} \sigma^x_{2j}. \tag{130}$$

This expression is meaningful only for even $L$. For odd $L$, the system does not have such a $\mathbb{Z}_2^{\mathcal{R}}$ symmetry. This will be discussed in detail in section 6.2. But in any event, now we limit ourselves to $L = 0 \mod 4$, so this definition makes sense. $\mathcal{R}$ satisfies

$$\mathcal{R}\mathcal{Q}\mathcal{R}^{-1} = -\mathcal{Q},$$
$$\mathcal{R}X = X\mathcal{R}. \tag{131}$$

As a concrete example, the Hamiltonian

$$H = \frac{1}{2} \sum_j (\sigma^x_j - \sigma^z_{j-1}\sigma^x_j\sigma^z_{j+1} + \lambda_z \sigma^z_j\sigma^z_{j+2}) \tag{132}$$

preserves the U(1) $\times \mathbb{Z}_2^X$ symmetry. The $\lambda_z = 0$ case was studied in [32]. In the dual representation, the model reads

$$H = \sum_j \left[ \mu^z_{j,j+1}(\tau^+_j\tau^-_{j+1} + \tau^-_j\tau^+_{j+1}) + \frac{\lambda_z}{2}\tau^z_j\tau^z_{j+1} \right]. \tag{133}$$

Here our $\tau^\pm$ are the same as in (9). This can be viewed as the XXZ model coupled to a $\mathbb{Z}_2$ gauge field.[28] The $\mathbb{Z}_2^X$ symmetry, which is generated by the Wilson line $X$ (128) is present because the Hamiltonian does not include terms that depend on $\mu^x$ or $\mu^y$.

Let us compare this construction to that in section 4.1. The matter Hamiltonian (90) is analogous to the XXZ Hamiltonian of $\tau^\pm, \tau^z$ (the Hamiltonian (133) with $\mu^z_{j,j+1} = 1$). In section 4.1, we gauged $\mathbb{Z} \subset \mathbb{R}$ to find (97). Here we gauge $\mathbb{Z}_2 \subset$ U(1) to find (133). And the Gauss's law constraint (94) is analogous to (127). The fact that the charge $Q_m$ (95) is

---

[28]Note that the gauge $\mathbb{Z}_2$ is a subgroup of the U(1) symmetry of the XXZ model, $\tau^\pm_j \to e^{\pm i\beta}\tau^\pm_j$ (with $\beta \sim \beta + 2\pi$), which is generated by $e^{\frac{i\beta}{2}\sum_j \tau^z_j}$. After the gauging, this global symmetry does not act faithfully, but $e^{\frac{i\xi}{4}\sum_j \tau^z_j}$ with $\xi \sim \xi + 2\pi$ (as in (128)) does act faithfully.

quantized using Gauss's law (96) is analogous to the quantization of the charge $\mathcal{Q}$ (128) using Gauss's law (129). The $\mathbb{Z}$ gauging led to a new $U(1)_w$ symmetry whose charge is the $\mathbb{Z}$ Wilson line (98) and the $\mathbb{Z}_2$ gauging leads to a new $\mathbb{Z}_2^X$ symmetry, whose charge is the $\mathbb{Z}_2$ Wilson line $X$ (128). This new symmetry was present in (97) because we set $g = 0$, and here it is present because (133) is independent of $\mu^x$ and $\mu^y$.

### 4.2.2 The defects

Now let us study defects of the $U(1)_m \times \mathbb{Z}_2^X$ symmetry. We will consider defects of $X, e^{i\sigma\mathcal{Q}}$ and $C = Xe^{i\pi\mathcal{Q}}$, and analyze how the symmetries are modified in the presence of the defects. For simplicity, when writing the twisted Hamiltonian we will set $\lambda_z = 0$.

The discussion of the defects will be similar to the analysis in section 4.1.4. In fact, the global $U(1) \times \mathbb{Z}_2^X$ symmetry here is a subgroup of the global $U(1)_m \times U(1)_w$ symmetry there.

First, the $\mathbb{Z}_2^X$ symmetry is generated by the $\mathbb{Z}_2$ Wilson loop and therefore, a $\mathbb{Z}_2^X$ defect can be created by modifying the Gauss's law constraint at a site $J$ to:

$$\tau_J^z = -\mu_{J-1,J}^x \mu_{J,J+1}^x. \tag{134}$$

Essentially, it means that the dual system is in the $\prod_j \tau_j^z = -1$ sector. This has the following consequence: using the identity Eq. (129) we now have $e^{2\pi i\mathcal{Q}} = -1$, so $\mathcal{Q} \in \mathbb{Z} + \frac{1}{2}$. In other words, a defect of the $X$ symmetry carries half-integer $U(1)$ charge. This is a hallmark of the mixed anomaly.

There is an alternative way to describe the defects, which is more straightforward to connect to the $\sigma_j^a$ spin representation. We first notice that by conjugating all operators with $\tau_J^x$, the Gauss's law constraint can be restored to the original form. The conjugation changes the Hamiltonian to

$$H^{(J)}(X) = \sum_{j \neq J-1,J} \mu_{j,j+1}^z (\tau_j^+ \tau_{j+1}^- + \text{h.c.}) + \sum_{j=J-1,J} \mu_{j,j+1}^z (\tau_j^+ \tau_{j+1}^+ + \text{h.c.}). \tag{135}$$

This Hamiltonian can then be translated back to the $\sigma_j^a$ presentation using the identities in (126):

$$H^{(J)}(X) = \sum_{j \neq J,J+1} (\sigma_j^x - \sigma_{j-1}^z \sigma_j^x \sigma_{j+1}^z) + \sum_{j=J,J+1} (\sigma_j^x + \sigma_{j-1}^z \sigma_j^x \sigma_{j+1}^z). \tag{136}$$

At the same time, the $U(1)$ charge is also modified by the conjugation to

$$\mathcal{Q}(X) = \frac{1}{4} \Big( \sum_{j \neq J} \tau_j^z - \tau_J^z \Big) = \sum_j q_j^m(X),$$

$$q_j^m(X) = q_j^m - \frac{\delta_{j,J}}{2} \tau_J^z, \tag{137}$$

where, as in (128), $q_j^m$ is the local charge density. Here we follow our conventions and denote the charge in the $X$-twisted theory by $\mathcal{Q}(X)$. Note that we have the relation $\mathcal{Q}(X) = \mathcal{Q} + \frac{\tau_J^z}{2} \in \mathbb{Z} + \frac{1}{2}$, so the modified $U(1)$ charge takes half-integer values, reflecting the mixed anomaly between the $U(1)$ and the $\mathbb{Z}_2^X$ symmetries. This is as in the continuum discussion in section 3 and as in the modified Villain theory in section 4.1.[29]

---

[29]As in the modified Villain theory, the local charge density $q_j^m$ is shifted at the location of the defect by $\pm\frac{1}{2}$. However, since in the modified Villain model, the eigenvalues of the local $U(1)$ charge density can be arbitrarily large and here they are $\pm\frac{1}{4}$, the change in the local charge in (137) is not merely an additive constant, but it depends on the operator $\tau_J^z$.

This defect is topological, as it can be shifted from $J$ to $J+1$ by the following conjugation:

$$
\begin{aligned}
H^{(J+1)}(X) &= U_X(J, J+1) H^{(J)}(X) U_X(J, J+1)^{-1}, \\
U_X(J, J+1) &= \tau_J^x \mu_{J,J+1}^z \tau_{J+1}^x = \sigma_{J+1}^x.
\end{aligned}
\tag{138}
$$

We can also introduce a defect corresponding to $e^{i\sigma \mathcal{Q}}$ (in the absence of $X$ defects):[30]

$$
H^{(J,J+1)}(\sigma) = \sum_{j \neq J} \mu_{j,j+1}^z (\tau_j^+ \tau_{j+1}^- + \text{h.c.}) + \mu_{J,J+1}^z (e^{-i\frac{\sigma}{2}} \tau_J^+ \tau_{J+1}^- + \text{h.c.}).
\tag{139}
$$

This Hamiltonian is obtained by conjugating all sites right to $J$ by $e^{i\frac{\sigma}{4}\tau^z}$. Clearly, this defect is located at the link $(J, J+1)$.

The appearance of $\frac{\sigma}{2}$ in the Hamiltonian (139) might seem surprising because it is not invariant under $\sigma \to \sigma + 2\pi$. However, the Hamiltonians at these two values of $\sigma$ are related by conjugation

$$
H^{(J,J+1)}(\sigma + 2\pi) = \mu_{J,J+1}^x H^{(J,J+1)}(\sigma) \mu_{J,J+1}^x.
\tag{140}
$$

This conjugation by $\mu_{J,J+1}^x = \sigma_{J+1}^z$ does not affect Gauss's law (127), but changes the eigenvalue of $X$. This is also an example of the general comments in section 2.3 about the Hamiltonian in the twisted theory.

In the $\sigma^a$ spin representation, the defect Hamiltonian reads

$$
\begin{aligned}
H^{(J,J+1)}(\sigma) = {} & \frac{1}{2} \sum_{j \neq J+1} (\sigma_j^x - \sigma_{j-1}^z \sigma_j^x \sigma_{j+1}^z) \\
& + \frac{1}{2} \cos \frac{\sigma}{2} (\sigma_{J+1}^x - \sigma_J^z \sigma_{J+1}^x \sigma_{J+2}^z) + \frac{1}{2} \sin \frac{\sigma}{2} (\sigma_J^z \sigma_{J+1}^y - \sigma_{J+1}^y \sigma_{J+2}^z).
\end{aligned}
\tag{141}
$$

This defect is also topological. It can be shifted from the link $(J-1, J)$ to $(J, J+1)$ by the following unitary conjugation:

$$
\begin{aligned}
H^{(J,J+1)}(\sigma) &= U_m(J) H^{(J-1,J)}(\sigma) U_m(J)^{-1}, \\
U_m(J) &= e^{-i\sigma q_J^m}.
\end{aligned}
\tag{142}
$$

Using such a shift, we can fuse two defects. Consider a defect with $\sigma$ at the link $(J-1, J)$ and the another defect with $\sigma'$ at the link $(J, J+1)$. Applying the moving operator to move the defect with $\sigma$ to the link $(J, J+1)$, there is now a single defect with $\sigma + \sigma'$. This is simple when $-\pi < \sigma, \sigma', \sigma + \sigma' < +\pi$. However, when $-\pi < \sigma, \sigma' \leq +\pi$, but $\sigma + \sigma'$ not in that range, we can conjugate by $\mu_{J,J+1}^x$, as in (140) to bring $\sigma + \sigma'$ back to this fundamental domain. As we said after (140), this has the effect of changing the eigenvalue of $X$. We can say that in this case, the fusing of $\sigma$ and $\sigma'$ defects creates charge of the $X$ symmetry. This phenomenon again can be viewed as a manifestation of the mixed anomaly.

This change of charge can be thought of as follows. When the $X$ charge is part of a continuous symmetry with charge $Q_w$ (as in the continuum discussion in section 3 and the lattice discussion in section 4.1 ), it is shifted by one unit as $\sigma \to \sigma + 2\pi$. In our case, where $X$ is a $\mathbb{Z}_2$ element, it does not flow. Instead, as $\sigma \to \sigma + 2\pi$, its eigenvalues change sign.

Note that when there is a $\sigma$ defect in the link $(J, J+1)$ and an $X$ defect at $J$ or $J+1$, the form of the Hamiltonian must be further modified. For example, when the site $J$ has an $X$ defect, we further conjugate $H^{(J,J+1)}(\sigma)$ (but not the Gauss's law constraint) by $\tau_J^x$, and arrive at the following Hamiltonian:

$$
H^{J;(J,J+1)}(X, \sigma) = \sum_{j \neq J-1, J} \mu_{j,j+1}^z (\tau_j^+ \tau_{j+1}^- + \text{h.c.}) + \left( \mu_{J-1,J}^z \tau_{J-1}^+ \tau_J^+ + e^{i\frac{\sigma}{2}} \mu_{J,J+1}^z \tau_J^+ \tau_{J+1}^+ + \text{h.c.} \right).
\tag{143}
$$

---

[30]We use $\sigma$ to denote the twist parameter and $\sigma_j^a$ to denote the variables in the $\sigma_j^a$ presentation. We hope that this will not cause confusion.

Similarly, if the $\sigma$ defect is on the link $(J-1,J)$, then the Hamiltonian is

$$H^{J;(J-1,J)}(X,\sigma) = \sum_{j \neq J-1,J} \mu^z_{j,j+1}(\tau^+_j \tau^-_{j+1} + \text{h.c.}) + \left(e^{-i\frac{\sigma}{2}} \mu^z_{J-1,J} \tau^+_{J-1} \tau^+_J + \mu^z_{J,J+1} \tau^+_J \tau^+_{J+1} + \text{h.c.}\right). \quad (144)$$

They are related by unitary conjugation that moves the $\sigma$ defect from the $(J-1,J)$ link to the $(J,J+1)$ link:

$$H^{J;(J,J+1)}(X,\sigma) = U_m(J) H^{J;(J-1,J)}(X,\sigma) U_m(J)^{-1}, \quad (145)$$
$$U_m(J) = e^{i\frac{\sigma}{4}\tau^z_J} = e^{-i\sigma q^m_J(X)}.$$

Lastly, we consider a special case of the defect (143) corresponding to $C = X e^{i\pi Q}$. We will define it as a composite of a $\sigma = \pi$ defect at the link $(J,J+1)$ and an $X$ defect at the site $J$. The Hamiltonian can be obtained from setting $\sigma = \pi$ in (143). While $C^2 = 1$, two $C$ defects are nontrivial. This can be easily seen by first applying the defect moving operators, and then fusing the two $\sigma = \pi$ defects. As we have discussed above, this has the effect of flipping the sign of $X$ charge, and hence flipping the sign of the $C$ charge.

### 4.2.3   The continuum limit

To compare with the continuum, it is convenient to start with the XXZ model (8)

$$H_{\text{XXZ}} = \sum_j \left(\tau^+_j \tau^-_{j+1} + \text{h.c.} + \frac{\lambda_z}{2} \tau^z_j \tau^z_{j+1}\right), \quad (146)$$

with $-1 < \lambda_z \leq 1$. It flows at long distances to the $c = 1$ compact boson. For reasons that will be clear soon, we denote the two charges of its global $U(1)_m \times U(1)_w$ symmetry by $\tilde{Q}_m$ and $\tilde{Q}_w$.

The Hamiltonian (133) is obtained by gauging $\mathbb{Z}_2 \subset U(1)$. This constrains $\tilde{Q}_m$ to be even and we get a new $\mathbb{Z}^X_2$ symmetry from the gauging, which is generated by the $\mathbb{Z}_2$ Wilson line $X$ (128).

In the continuum limit of this system, the global $U(1)$ is the momentum symmetry, and therefore, the gauging has the effect of halving the radius, leading to a compact boson CFT with radius [32]

$$R = \frac{R_{\text{XXZ}}}{2}. \quad (147)$$

This continuum theory also has a $\left(U(1)_m \times U(1)_w\right) \rtimes \mathbb{Z}^{\mathcal{R}}_2$ global symmetry. But its symmetry operators are different than before the gauging. The gauging excludes the states with odd $\tilde{Q}_m$ and adds from the twisted sector states with $\tilde{Q}_w \in \mathbb{Z} + \frac{1}{2}$. We can express the charges of the new symmetry as

$$Q_m = \frac{1}{2}\tilde{Q}_m, \quad (148)$$
$$Q_w = 2\tilde{Q}_w,$$
$$Q_m, Q_w \in \mathbb{Z}.$$

We readily identify the $U(1)$ of the lattice model as $U(1)_m$ generated by $e^{i\xi Q} = e^{i\xi Q_m}$ and $X = e^{i\pi Q_w}$. This means that the momentum $Q_m$ is a microscopically exact symmetry, but only $Q_w$ mod 2 is an exact symmetry in the lattice model. We should note that the identification is done for $L = 0$ mod 4. For other $L$, the matching between the lattice model and the continuum theory is more subtle and will be discussed in section 6.2.[31]

---

[31]As we will discuss in section 6.2, the lattice translation symmetry leads at large $L$ to another internal symmetry, which extends the $\mathbb{Z}_2$ of $(-1)^{Q_w}$ to $\mathbb{Z}_4$ of $e^{\frac{i\pi}{2}Q_w}$. Using the terminology in section 1.3, this $\mathbb{Z}_4$ symmetry is an emanant symmetry and hence it is exact in the continuum limit. Therefore, when we check for a possible BKT transition, we should focus on operators with $Q_w = \pm 4$. They are irrelevant for $R > \frac{1}{2}$. Hence, this model is gapless there and is gapped for $R < \frac{1}{2}$. This is consistent with the discussion before gauging (in section 1.3) using the relation (147) between $R$ and $R_{\text{XXZ}}$. In terms of the lattice model, this transition happens at $\lambda_z = 1$.

# 5 Anomalies involving lattice translation, an emanant $\mathbb{Z}_2$ symmetry, and LSM

## 5.1 General considerations

In the following, we consider systems with a tensor product Hilbert space structure, and an internal symmetry of the following type. For each $g \in G$, we have an operator $\rho_j(g)$ acting on the Hilbert space of the site labeled by $j$. And the symmetry operator of the group element $g$ is represented by

$$\rho(g) = \bigotimes_j \rho_j(g). \tag{149}$$

This representation may be projective. If it is linear, then according to our definition in section 4 the symmetry action is on-site and free of any kind of 't Hooft anomaly.

More generally, assume that the representation on a single site is projective:

$$\rho_j(g_1)\rho_j(g_2) = e^{i\varphi(g_1, g_2)} \rho_j(g_1 g_2). \tag{150}$$

Then, the $G$ symmetry itself has no 1+1d 't Hooft anomaly.[32]

In this case, a background gauge field for $G$ resides on the links. And therefore the $G$-defects are also located on links. (There is no problem doing this even when $G$ acts projectively on the local Hilbert space i.e., it does not act on-site according to our definition in section 4.) For a single $g$ defect on the link $(J, J+1)$, the modified translation operator $T(g)$ takes the following form:

$$T(g) = \rho_{J+1}(g)T. \tag{151}$$

Since $h \in C_g$, $\rho_j(g)$ and $\rho_j(h)$ commute when acting on operators and hence, $\rho(h) = \otimes_j \rho(h)$ still commutes with the Hamiltonian with the $g$ defect. Therefore, $h(g) = h(1)$ for $h \in C_g$. For simplicity, we will suppress the $g$ label.

In the examples below, $T$ and $g \in G$ and $h \in C_g$ commute as group elements. But the action of $G$ on one site is projective and is associated with the phases in (150). Therefore,

$$T(g)h = e^{i[\varphi(g, h) - \varphi(h, g)]} h T(g). \tag{152}$$

When $e^{i[\varphi(g, h) - \varphi(h, g)]} \neq 1$, the site Hilbert space forms a nontrivial projective representation. We see from (152) that in this case the whole system with a $g$ defect forms a nontrivial projective representation of $\mathcal{G}_g$, which is a manifestation of the mixed anomaly between the internal symmetry $G$ and the lattice translation $\mathbb{Z}_L$. As we will discuss in the next subsection, this anomaly is related to the Lieb-Schultz-Mattis theorem.

The anomaly associated with the Lieb-Schultz-Mattis theorem is a mixed anomaly between lattice translation and the internal symmetry $G$. One way it can be realized in the low-energy limit is through the appearance of an emanant internal symmetry with a mixed anomaly between it and the internal symmetry $G$. Also, this emanant symmetry can have its own anomaly. In the following sections, we will discuss this emanant symmetry and its anomalies in detail.

## 5.2 The SO(3) spin chain

We will consider a translation-invariant spin-1/2 chain with short-range, SO(3)-invariant interactions. The prototypical example is the antiferromagnetic Heisenberg chain:

$$H = 2\sum_{j=1}^{L} \mathbf{S}_j \cdot \mathbf{S}_{j+1}. \tag{153}$$

---

[32]For certain system size, $G$ may act projectively, which is a 0+1d 't Hooft anomaly as discussed in section 1.1, but this by itself does not lead to a 1+1d anomaly in $G$.

The theory has a $\mathbb{Z}_L$ translation symmetry generated by $T$

$$T\mathbf{S}_j T^{-1} = \mathbf{S}_{j+1}. \tag{154}$$

This condition leaves the phase of $T$ ambiguous. To fix it, we can choose a completely polarized state $|\uparrow \cdots \uparrow\rangle$, and require that it is invariant under $T$ without any phase. In the basis of diagonal $S_j^z$, $T$ acts as

$$T|m_1, m_2, \cdots m_L\rangle = |m_L, m_1, \cdots m_{L-1}\rangle. \tag{155}$$

Define the total spin operator

$$\mathbf{S} = \sum_{j=1}^{L} \mathbf{S}_j. \tag{156}$$

Since each site forms a spin-1/2 representation, the entire chain is in an integer/half-integer representation of SO(3) for even/odd $L$, so

$$e^{2\pi i S^x} = e^{2\pi i S^y} = e^{2\pi i S^z} = (-1)^L. \tag{157}$$

We emphasize that while we chose the simplest Heisenberg Hamiltonian, our discussion is more general and it is independent of the specific form of the Hamiltonian.

## 5.3 SO(3) defect

The model can be twisted by an element of SO(3). Up to conjugation, we can take this element to be $e^{i\sigma S_z}$ with $\sigma \sim \sigma + 2\pi \sim -\sigma$. The twisted Hamiltonian is:

$$H(\sigma) = 2\sum_{j=1}^{L-1} \mathbf{S}_j \cdot \mathbf{S}_{j+1} + \left(e^{-i\sigma} S_L^+ S_1^- + \text{h.c.} + 2S_L^z S_1^z\right). \tag{158}$$

Recall that we use the convention $S^{\pm} = S^x \pm i S^y$. Here we put the defect at the link $(L, 1)$. We can move the twist to the link $(1, 2)$ by conjugating $H(\sigma)$ with $e^{-i\sigma S_1^z}$, so the defect is a topological one.

It is useful to consider the operator

$$\mathcal{R} = e^{i\pi S^x} = 2i S^x. \tag{159}$$

It generates $\mathbb{Z}_2^{\mathcal{R}} \subset SO(3)$. For $\sigma \neq 0 \bmod \pi$, this $\mathbb{Z}_2^{\mathcal{R}}$ symmetry is violated. Instead, $\mathcal{R}$ implements the identification $\sigma \sim -\sigma$:

$$H(-\sigma) = \mathcal{R} H(\sigma) \mathcal{R}^{-1}. \tag{160}$$

The internal symmetry now depends on $\sigma$:

$$G(\sigma) = \begin{cases} SO(3), & \sigma = 0, \\ O(2), & \sigma = \pi, \\ SO(2), & \sigma \neq 0, \pi. \end{cases} \tag{161}$$

Here O(2) is generated by $e^{i\xi S^z}$ and $\mathcal{R}$. Below, we will also consider parity $\mathcal{P}$. There is also time-reversal $e^{i\pi S^y} K$, but we will ignore it.

The twisted Hamiltonian $H(\sigma)$ commutes with the twisted translation operator

$$T(\sigma) = e^{i\sigma S_1^z} T. \tag{162}$$

With this definition we find

$$\begin{aligned} T(\sigma)^L &= e^{i\sigma S^z}, \\ T(\sigma + 2\pi) &= -T(\sigma). \end{aligned} \tag{163}$$

We can redefine $T(\sigma)$ by a $\sigma$-dependent phase, e.g., multiply it by $e^{i\sigma/2}$ or $e^{-i\sigma/2}$, and remove the minus sign in the second equation in (163). Such a redefinition, introduces phases in other expressions involving $T(\sigma)$. We will return to this freedom in redefining $T(\sigma)$ below. It is important that this minus sign does not mean that the eigenvalues of $T(\sigma)$, $e^{i\alpha_n(\sigma)}$ satisfy $e^{i\alpha_n(\sigma+2\pi)} = -e^{i\alpha_n(\sigma)}$. Instead, it means that for every $n$, there is $m$ such that $e^{i\alpha_n(\sigma+2\pi)} = -e^{i\alpha_m(\sigma)}$. Namely, the entire spectrum of $T(\sigma)$ must satisfy the relation, but the eigenvalues can rearrange themselves as $\sigma$ varies from 0 to $2\pi$.

It will also be useful to consider the parity transformation $\mathcal{P}$ that maps site $j$ to site $L+1-j$ (i.e., reflection across the center of the link $(L, 1)$). $\mathcal{P}$ is a symmetry only when $\sigma = 0, \pi$. In fact

$$\mathcal{P}H(\sigma)\mathcal{P}^{-1} = H(-\sigma). \tag{164}$$

However, using Eq. (160), $\mathcal{P}' = \mathcal{R}\mathcal{P}$ commutes with the Hamiltonian. It is a good parity-like symmetry for any $\sigma$. It satisfies

$$\begin{aligned} \mathcal{P}'T(\sigma) &= T(\sigma)^{-1}\mathcal{P}', \\ \mathcal{P}'e^{i\xi S^z} &= e^{-i\xi S^z}\mathcal{P}', \\ (\mathcal{P}')^2 &= (-1)^L. \end{aligned} \tag{165}$$

(The phase in the last relation can be absorbed in a redefinition of $\mathcal{P}'$, but for simplicity, we will not do this.)

At $\sigma = 0$, the full symmetry group of the spin chain is $SO(3) \times \mathbb{D}_L$, where $\mathbb{D}_L = \mathbb{Z}_L \rtimes \mathbb{Z}_2^P$ is the dihedral group generated by $T$ and $P$. When $\sigma = \pi$, the symmetry group is $O(2) \times \mathbb{D}_L$. Interestingly, this group is realized projectively, i.e.,

$$T(\pi)\mathcal{R} = -\mathcal{R}T(\pi). \tag{166}$$

This fact is interpreted as a mixed anomaly between $SO(3)$ and $\mathbb{Z}_L \subset \mathbb{D}_L$.

## 5.4 Twisted partition functions

Define the twisted partition function:

$$\mathcal{Z}(\beta, L, \sigma, \xi, n) = \text{Tr}[e^{-\beta H(\sigma)} e^{i\xi S^z} T(\sigma)^n]. \tag{167}$$

Then following Eq. (163), we immediately have

$$\mathcal{Z}(\beta, L, \sigma + 2\pi, \xi, n) = (-1)^n \mathcal{Z}(\beta, L, \sigma, \xi, n), \tag{168}$$

and

$$\mathcal{Z}(\beta, L, \sigma, \xi, n + L) = \mathcal{Z}(\beta, L, \sigma, \xi + \sigma, n). \tag{169}$$

It then follows from $e^{2\pi i S^z} = (-1)^L$ that

$$\mathcal{Z}(\beta, L, \sigma, \xi + 2\pi, n) = (-1)^L \mathcal{Z}(\beta, L, \sigma, \xi, n). \tag{170}$$

Conjugating by $\mathcal{R}$, we find

$$\mathcal{Z}(\beta, L, \sigma, \xi, n) = \mathcal{Z}(\beta, L, -\sigma, -\xi, n). \tag{171}$$

Finally, conjugating by $\mathcal{P}'$ gives

$$\mathcal{Z}(\beta, L, \sigma, \xi, n) = \mathcal{Z}(\beta, L, \sigma, -\xi, -n), \tag{172}$$

reflecting the $\mathcal{P}'$ symmetry of the problem with generic $\sigma$.

At $\sigma = 0$, these relations lead to

$$\begin{aligned}
\mathcal{Z}(\beta, L, 0, \xi, n) &= \mathcal{Z}(\beta, L, 0, -\xi, n), \\
\mathcal{Z}(\beta, L, 0, \xi, n) &= \mathcal{Z}(\beta, L, 0, \xi, -n),
\end{aligned} \tag{173}$$

which are special cases of the enhanced SO(3) symmetry and hence $\mathcal{P}$ at this point. And at $\sigma = \pi$, they lead to

$$\begin{aligned}
\mathcal{Z}(\beta, L, \pi, \xi, n) &= (-1)^n \mathcal{Z}(\beta, L, \pi, -\xi, n), \\
\mathcal{Z}(\beta, L, \pi, \xi, n) &= (-1)^n \mathcal{Z}(\beta, L, \pi, \xi, -n),
\end{aligned} \tag{174}$$

reflecting the enhanced O(2) symmetry and hence $\mathcal{P}$ at this point. The factor of $(-1)^n$ is related to the projective relation (166).

Given that for every $\sigma$, the theory has the parity-like symmetry $\mathcal{P}'$, we can study another partition function

$$\mathcal{Z}^{\mathcal{P}'}(\beta, L, \sigma, \xi, n) = \text{Tr}[e^{-\beta H(\sigma)} e^{i\xi S^z} \mathcal{P}' T(\sigma)^n]. \tag{175}$$

Cycling $T(\sigma)$ around the trace using (165), shows that the trace depends only on $n$ mod 2. Similarly, replacing $e^{i\xi S^z}$ in the trace by $e^{i\frac{\xi}{2} S^z} e^{i\frac{\xi}{2} S^z}$ and cycling the first factor around the trace using (165), we transform it to $\xi = 0$, thus showing that $\mathcal{Z}^{\mathcal{P}'}$ is independence of $\xi$. Furthermore, combining this fact with $\mathcal{Z}^{\mathcal{P}'}(\beta, L, \sigma, \xi + 2\pi, n) = (-1)^L \mathcal{Z}^{\mathcal{P}'}(\beta, L, \sigma, \xi, n)$, we learn that for odd $L$, $\mathcal{Z}^{\mathcal{P}'}(\beta, L, \sigma, \xi, n) = 0$.[33] To summarize

$$\begin{aligned}
\mathcal{Z}^{\mathcal{P}'}(\beta, L, \sigma, \xi, n) &= \mathcal{Z}^{\mathcal{P}'}(\beta, L, \sigma, 0, n \bmod 2), \\
\mathcal{Z}^{\mathcal{P}'}(\beta, L, \sigma, \xi, n) &= 0, \quad \text{for} \quad L \in 2\mathbb{Z} + 1.
\end{aligned} \tag{176}$$

For $\sigma = 0, \pi$, the theory has a larger global symmetry and we can study additional partition functions. For $\sigma = 0$ an insertion of any SO(3) global symmetry operator can be conjugated to $e^{i\xi S^z}$ and therefore there is no new partition function to study. However, for $\sigma = \pi$, because the symmetry group becomes O(2) and the spatial twist $e^{i\pi S^z}$ is central, we can consider

$$\mathcal{Z}^{\mathcal{R}}(\beta, L, \pi, \xi, n) = \text{Tr}[e^{-\beta H(\pi)} \mathcal{R} e^{i\xi S^z} T(\pi)^n]. \tag{177}$$

Space is twisted by $g(g) = e^{i\pi S^z}$ and Euclidean time is twisted by $h(g) = \mathcal{R} e^{i\xi S^z}$. More mathematically, we created an SO(3) bundle with nontrivial second Stiefel–Whitney class on the torus.

It is important that for odd $L$, $g(g)$ and $h(g)$ realize the global SO(3) symmetry projectively. Still, we can study the system in a nontrivial SO(3) bundle. As in our general discussion, $h$ is in the centralizer of $g$ even though their representations in the twisted Hilbert space $g(g)$ and $h(g)$ do not commute.

---

[33]Equivalently, we can use a basis of the Hilbert space with diagonal $H(\sigma)$, $T(\sigma)$, and $S^z$. Then, the relations (165) show that $\mathcal{P}'$ maps these states to states with the inverse eigenvalues of $T(\sigma)$ and the opposite eigenvalues of $S^z$. Consequently, only states with $T(\sigma)$ eigenvalues $\pm 1$ and vanishing $S^z$ eigenvalues can be $\mathcal{P}'$ eigenstates and contribute to the trace $\mathcal{Z}^{\mathcal{P}'}$. For odd $L$, there are no such states and therefore $\mathcal{Z}^{\mathcal{P}'}$ vanishes. For even $L$, the partition function is independent of $\xi$ and depends only on $n$ mod 2.

For all $L$, $T(\pi)$ and $\mathcal{R}$ anti-commute (see Eq. (166)). Consequently, the trace (177) vanishes

$$\mathcal{Z}^{\mathcal{R}}(\beta, L, \pi, \xi, n) = \text{Tr}[e^{-\beta H(\pi)} \mathcal{R} e^{i\xi S^z} T(\pi)^n] = 0, \tag{178}$$

which can be shown by inserting $1 = T(\pi)T(\pi)^{-1}$ in the trace and cycling one of the factors around the trace. This is a special case of (32).

The relations (168),(170),(171) are associated with the internal symmetry. Here we can think of $\sigma$ and $\xi$ as background gauge fields for that symmetry and the relations characterize the distinct backgrounds. The other relations (169),(172) are associated with the lattice symmetries. Here, unlike the purely internal symmetry relations, they involve action also on the parameters of the internal symmetry. We emphasize that this action on the internal symmetry parameters is not an anomaly.

The anomaly is the statement that the relations (168) and (170) involve phases. Mathematically, the partition function is not a function on the parameter space, but it is a section on that space. (More precisely, they involve phases that cannot be absorbed in a redefinition of $\mathcal{Z}$.) Similarly, the vanishing of the partition function (178) reflects an anomaly.

The conditions on the partition function (168)-(172),(178) are independent of $\beta$. Therefore, they should be satisfied also in the low-energy limit obtained by taking $\beta$ large. The anomaly associated with the phases in (168) and (170) means that the partition function cannot be a constant in that limit. Consequently, the low-energy spectrum cannot be trivial. It should account for the same phases. Similarly, the vanishing (178), which follows from the anomaly (166) shows the system cannot have a unique ground state. These are the 't Hooft anomaly matching relations – these phases and this vanishing are independent of $\beta$ and therefore, they are the same in the UV and the IR. In this particular case of a spin chain, this is the LSM result.

## 5.5 Spectrum of the lattice model using the continuum analysis

In this section we will make some reasonable qualitative assumptions, which will allow us to derive *exact expressions* for some of the properties of the lattice spectrum. These expressions will also demonstrate some of the discussion above.

It is well-known that the continuum theory of the spin-$\frac{1}{2}$ Heisenberg chain is the SU(2)$_1$ theory [43, 58], where the SO(3) spin rotation becomes the diagonal SO(3) of the SO(4) symmetry, and the unit lattice translation becomes the central element $C$ of SO(4).

For large $L$, the results of the lattice model should agree with the continuum answers in section 3.4. More precisely, for finite $L$, the lattice Hilbert space is finite and only the low-lying states should match the continuum analysis. As $L$ gets larger, more and more states should match and the agreement in their properties should become more and more accurate.

On the lattice, we have only an SO(3) ⊂ SO(4) global symmetry and therefore, the states are labeled only by that symmetry. Furthermore, we can twist only by SO(3) and then the states are labeled by U(1) ⊂ SO(3). We identify this charge as

$$S^z = Q_m = Q + \overline{Q}. \tag{179}$$

In the continuum discussion in section 3.4.3, we introduced a $\mathbb{Z}_2^C$ twist labeled by $N$ mod 4. We identify it with the number of lattice sites

$$N = L \bmod 4. \tag{180}$$

We stress that we cannot prove this assertion, but simply assume it here. See the related discussion in section 6.1. Then, we can express Eq. (179) more precisely as

$$S^z = Q(2\pi N + \sigma) + \overline{Q}(\sigma) = Q_m(\pi N). \tag{181}$$

Note that this charge does not change as we vary $\sigma$. The chiral charges $Q(2\pi N)$ and $\overline{Q}(0)$ are defined separately in the continuum, but only their sum (181) is meaningful on the lattice.

The continuum discussion leading to (88) and (89) determines the lattice energy as (recall (180))

$$
\begin{aligned}
E(L,\sigma) &= A(L)\big(L_0(2\pi N + \sigma) + \overline{L}_0(\sigma)\big) + B(L,\sigma) + \mathcal{O}\!\left(\frac{1}{L}\right) \\
&= E(L,0) + A(L)\left(\frac{\sigma}{2\pi}\big(Q(2\pi N) - \overline{Q}(0)\big) + \frac{\sigma^2}{8\pi^2}\right) + \mathcal{O}\!\left(\frac{1}{L}\right) \\
&= E(L,0) + A(L)\left(\frac{\sigma}{2\pi}Q_w(\pi N) + \frac{\sigma^2}{8\pi^2}\right) + \mathcal{O}\!\left(\frac{1}{L}\right).
\end{aligned}
\tag{182}
$$

Here, $A(L)$, $B(L,\sigma)$, and the $\mathcal{O}\!\left(\frac{1}{L}\right)$ corrections are non-universal and depend on the details of the lattice Hamiltonian. Also, note that this expression depends separately on $Q(2\pi N)$ and $\overline{Q}(0)$, or equivalently, on $Q_w(\pi N)$, which are meaningful only in the continuum.

Most interesting is the match between the lattice $T(\sigma)$ and the continuum quantities. The lattice expression

$$
T(\sigma)^L = e^{i\sigma S^z}
\tag{183}
$$

should correspond to the continuum expressions (87),(88), and (89)

$$
\begin{aligned}
U &= e^{2\pi i P(\sigma + \pi N, \pi N)} = C(\sigma + \pi N, \pi N)^N e^{i\sigma Q_m(\pi N)}, \\
P(\sigma + \pi N, \pi N) &= P(\pi N, \pi N) + \frac{\sigma}{2\pi}Q_m(\pi N), \\
C(\sigma + \pi N, \pi N) &= e^{i\pi(Q_w(0) + Q_m(\pi N))} \\
&= i^N e^{i\pi(Q_m(0) + Q_w(0))} = i^N e^{2\pi i Q(0)} = (-i)^N e^{2\pi i Q(2\pi N)}.
\end{aligned}
\tag{184}
$$

We expect that as $L \to \infty$, the eigenvalues of $T(\sigma)$ of the low-lying states approach the continuum expression (84)[34]

$$
\lim_{L\to\infty} T(\sigma) = C(\sigma + \pi N, \pi N)^{-1} = i^N e^{-2\pi i Q(2\pi N)}.
\tag{185}
$$

The $\mathcal{O}\!\left(\frac{1}{L}\right)$ corrections to Eq. (185) are determined such that raising $T(\sigma)$ to a large power (of order $L$) matches with the continuum translation by $P(\sigma + \pi N, \pi N)$

$$
T(\sigma) = C(\sigma + \pi N, \pi N)^{-1} e^{\frac{2\pi i}{L}P(\sigma + \pi N, \pi N)} = i^N e^{-2\pi i Q(2\pi N)} e^{\frac{2\pi i}{L}P(\sigma + \pi N, \pi N)}.
\tag{186}
$$

The limit (185) and the expression (186) include a factor $C(\sigma + \pi N, \pi N)^{-1}$ rather than $C(\sigma + \pi N, \pi N)$ such that the lattice relation (183) is compatible with the continuum expression (184). This will be discussed further in section 6.

Adding $\mathcal{O}\!\left(\frac{1}{L^2}\right)$ corrections in the exponent, would not preserve the condition (183). Therefore, this expression is exact for all $L$. The dependence on finite $L$ enters only through the information about the charges of the states in the spectrum. The $T(\sigma)$ eigenvalues of these states are given exactly by Eq. (186)!

Using these expressions, and the information in Tables 1 and 2, it is easy to calculate the eigenvalues of $T(\sigma)$ and the energies $E(\sigma)$ of the low-lying states and they are presented in Tables 3 and 4.

As we emphasized above, our large $L$ expressions for the energy have higher order corrections in $\frac{1}{L}$. However, the expressions for the momenta are exact. Let us demonstrate it for

---

[34]The reader might find it surprising that we wrote $C(\sigma + \pi N, \pi N)^{-1}$ in this expression. As we discussed above, $C$ can act projectively in the twisted Hilbert space. Therefore, $C(\pi N, \pi N)^{-1}$ is not necessarily the same as $C(\pi N, \pi N)$. The fact that this expression should have $C^{-1}$ rather than $C$ will be clear soon.

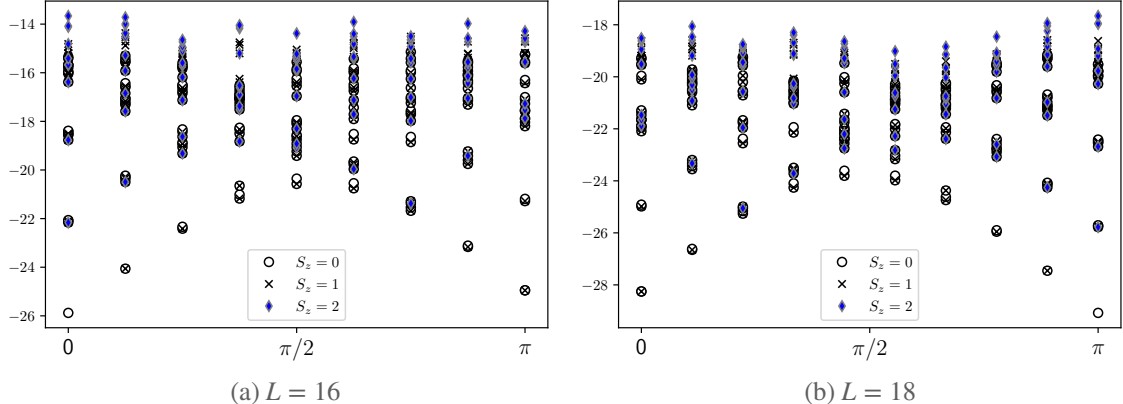

(a) $L = 16$            (b) $L = 18$

Figure 3: Spectra of the extended Heisenberg model with even length $L = 16$ and 18. Here the horizontal axis is the lattice momentum and the vertical axis is the energy. We only show states with momentum in $[0, \pi]$, as the $[-\pi, 0]$ part are related by parity symmetry. Note that the ground-state momentum is different in these two cases.

small values of $L$ and focus on the ground states. We will write wavefunctions in the $S_z = \uparrow, \downarrow$ basis.

For $L = 2$, the ground state of the Heisenberg Hamiltonian is an SU(2) singlet

$$\frac{1}{\sqrt{2}}\Big(|\downarrow\uparrow\rangle - |\uparrow\downarrow\rangle\Big). \tag{187}$$

Clearly, its $T$ eigenvalue is $-1$, as in Table 3.

For $L = 3$, the ground states are two SU(2) doublets. The states with $S_z = \frac{1}{2}$ are

$$\frac{1}{\sqrt{3}}\Big(|\downarrow\uparrow\uparrow\rangle + \omega|\uparrow\downarrow\uparrow\rangle + \omega^2|\uparrow\uparrow\downarrow\rangle\Big), \qquad \omega = e^{\pm\frac{2\pi i}{3}}, \tag{188}$$

with $T$ eigenvalue $\omega^{-1}$. Our formulas in Table 4 lead to the eigenvalues $i^3 e^{-\frac{i\pi}{6}} = e^{-\frac{2\pi i}{3}}$ and $-i^3 e^{\frac{i\pi}{6}} = e^{\frac{2\pi i}{3}}$.

For $L = 4$, there is a unique ground state. It is an SU(2) singlet and its wavefunction is

$$\frac{1}{2\sqrt{3}}\Big[\Big(|\uparrow\uparrow\downarrow\downarrow\rangle + |\downarrow\uparrow\uparrow\downarrow\rangle + |\downarrow\downarrow\uparrow\uparrow\rangle + |\uparrow\downarrow\downarrow\uparrow\rangle\Big) - 2\Big(|\uparrow\downarrow\uparrow\downarrow\rangle + |\downarrow\uparrow\downarrow\uparrow\rangle\Big)\Big]. \tag{189}$$

It is easy to see that this state indeed has $T = 1$, as in Table 3. Note in particular, that the ground state eigenvalue of $T$ differs from the value for $L = 2$ in (187), reflecting the modulo four (as opposed to the naive modulo two) periodicity.

For $L = 5$, the expressions for the ground state wavefunctions are too complicated to write down, but we still find the ground states in SU(2) doublets and $T$ eigenvalue $e^{\pm\frac{2\pi i}{5}}$, as predicted by the expressions in Table 4.

We have checked explicitly the formulas for larger $L$ both for the ground states and the low-lying excited states and included also the twist by $\sigma$. In all these cases, we found perfect agreement between this analysis and the explicit lattice calculations. In Figs. 3 and 4 we show representative plots of the low-energy spectra of an extended Heisenberg model (including next-nearest-neighbor Heisenberg interactions to minimize finite-size effect) for $L = 16, 17, 18, 19$.

One might question whether the exact expressions for small values of $L$ can change as we vary the Hamiltonian. The crucial point is that the eigenvalues of $T$ are quantized as $L$'th root

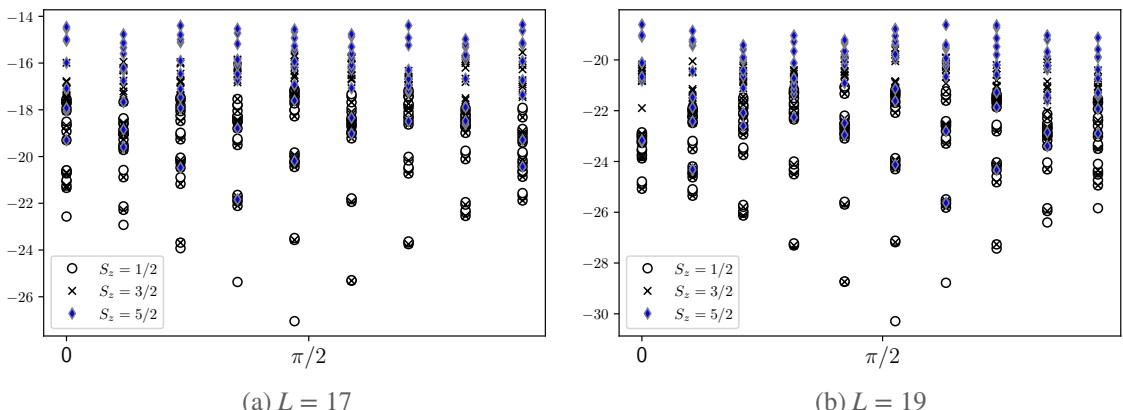

(a) $L = 17$                              (b) $L = 19$

Figure 4: Spectra of the extended Heisenberg model with odd length $L = 17$ and 19. Note that the ground-state momentum is different in these two cases.

of one. Hence, continuous changes in the Hamiltonian can change the energy eigenvalues, but they cannot change the eigenvalues of $T$. Having said that, a deformation of the Hamiltonian changes the energy eigenvalues and can affect their order. Low-energy states can become high-energy states and vice versa. Then, the match with the large $L$ limit is not obvious.

Let us summarize this discussion. The lattice $\mathbb{Z}_L$ translation symmetry leads in the continuum to an emanant $\mathbb{Z}_2^C$ symmetry. This symmetry is an exact symmetry rather than an approximate emergent symmetry. The lattice theory has a mixed anomaly between the $\mathbb{Z}_L$ translation and the internal SO(3) symmetries. It becomes the mixed anomaly between the emanant $\mathbb{Z}_2^C$ symmetry and the internal SO(3) of the continuum theory (the first term in the anomaly expression (81)). The pure $\mathbb{Z}_2^C$ anomaly of the continuum theory (the second term in the anomaly expression (81)) arises through the modulo four periodicity as a function of $L$ and is reflected in the factor of $i^N$ in the formula for $T$ (186).

## 5.6 Deformations of this model

### 5.6.1 Deforming, while preserving $G = \mathrm{O}(2) \subset \mathrm{O}(3)$ – the XXZ model

The SO(3) Hamiltonian (153)

$$H = 2 \sum_j \mathbf{S}_j \cdot \mathbf{S}_{j+1} \tag{190}$$

can be deformed to the XXZ Hamiltonian (146)

$$H_{\mathrm{XXZ}} = \sum_j \left( \tau_j^+ \tau_{j+1}^- + \mathrm{h.c.} \right) + \frac{\lambda_z}{2} \sum_j \tau_j^z \tau_{j+1}^z. \tag{191}$$

In the continuum, this model flows to the $c = 1$ compact boson theory (33) with radius (10). Its global symmetries were discussed in section 4.2.3.

Our discussion of the SO(3) spin chain model (190) can be repeated easily for the more general XXZ model (191). One difference is that the global SO(3) symmetry of (190) is reduced to $\mathrm{O}(2) \subset \mathrm{SO}(3)$, but this does not affect most of our conclusions.

First, as in the SO(3) case, we can add defects for the global U(1) symmetry, parameterized by $\sigma$. Second, for $\sigma = 0 \bmod \pi$ the O(2) symmetry is preserved and we can insert the $\mathbb{Z}_2^{\mathcal{R}}$ generator $\mathcal{R}$ in the traces, as in (177). Third, we can twist space by $\mathcal{R}$. Finally, the low-energy theory has an emanant $\mathbb{Z}_2^C$ symmetry, which arises from lattice translation $T$ (see section 1.3). Defects in this $\mathbb{Z}_2^C$ lead to dependence of the low-energy theory on $L \bmod 4$. As in the special

Table 3: The low-lying states of the lattice Hamiltonian (158) for even $N = L$ mod 4. In the expression for the energy, we suppressed the non-universal multiplicative and additive constants (including the term $\frac{\sigma^2}{8\pi^2}$), as well as the $\mathcal{O}\left(\frac{1}{L}\right)$ corrections. Here we used the information in Table 1.

| $\left(Q(2\pi N),\overline{Q}(0)\right)$ | $S^z = Q_m(\pi N)$ | $T(\sigma)$ | $E$ |
|---|---|---|---|
| $(0,0)$ | $0$ | $i^N$ | $0$ |
| $(-\frac{1}{2},\frac{1}{2})$ | $0$ | $-i^N$ | $\frac{1}{2} - \frac{\sigma}{2\pi}$ |
| $(\frac{1}{2},\frac{1}{2})$ | $1$ | $-i^N e^{\frac{i\sigma}{L}}$ | $\frac{1}{2}$ |
| $(-\frac{1}{2},-\frac{1}{2})$ | $-1$ | $-i^N e^{-\frac{i\sigma}{L}}$ | $\frac{1}{2}$ |
| $(\frac{1}{2},-\frac{1}{2})$ | $0$ | $-i^N$ | $\frac{1}{2} + \frac{\sigma}{2\pi}$ |
| $(-1,0)$ | $-1$ | $i^N e^{\frac{i}{L}(2\pi-\sigma)}$ | $1 - \frac{\sigma}{2\pi}$ |
| $(0,1)$ | $1$ | $i^N e^{-\frac{i}{L}(2\pi-\sigma)}$ | $1 - \frac{\sigma}{2\pi}$ |

case with SO(3) symmetry, for $L$ even the global O(2) symmetry is realized linearly, but it is realized projectively for odd $L$.

Examining the partition functions with defects as a function of the various twists, is essentially the same as in the SO(3) model and leads to LSM-type theorem for the XXZ chain, which have already appeared in [12].

Finally, in terms of the continuum limit, this theory with its defects is captured by the discussion in section 3 with generic $R \geq 1$ and becomes the discussion of the SO(3) system in the special case $R = 1$ of section 3.4.

### 5.6.2 Deforming, while preserving $G = \mathbb{Z}_2 \times \mathbb{Z}_2^{\mathcal{R}} \subset O(2) \subset SO(3)$ – the XYZ model

In this section, we deform the spin chain (191) further to preserve only $\mathbb{Z}_2 \times \mathbb{Z}_2^{\mathcal{R}} \subset O(2) \subset SO(3)$. The Hamiltonian is

$$H_{\text{XYZ}} = \frac{1}{2} \sum_j \left( \lambda_x \tau_j^x \tau_{j+1}^x + \lambda_y \tau_j^y \tau_{j+1}^y + \lambda_z \tau_j^z \tau_{j+1}^z \right). \tag{192}$$

This model has already been analyzed using defects in [12]. Here we will present it along the lines of our earlier discussion and will phrase it as an anomaly.

The system has a $\mathbb{Z}_2 \times \mathbb{Z}_2^{\mathcal{R}}$ symmetry generated by

$$\begin{aligned} Z &= i^{-L} e^{\frac{i\pi}{2} \sum_j \tau_j^z} = \prod_j \tau_j^z, \\ X &= i^{-L} \mathcal{R} = \prod_j \tau_j^x, \end{aligned} \tag{193}$$

Table 4: The low-lying states of the lattice Hamiltonian (158) for odd $N = L$ mod 4. In the expression for the energy, we suppressed the non-universal multiplicative and additive constants (including the term $\frac{\sigma^2}{8\pi^2}$), as well as the $\mathcal{O}\left(\frac{1}{L}\right)$ corrections. Here we used the information in Table 2.

| $\left(Q(2\pi N), \overline{Q}(0)\right)$ | $S^z = Q_m(\pi N)$ | $T(\sigma)$ | $E$ |
|:---:|:---:|:---:|:---:|
| $(-\frac{1}{2}, 0)$ | $-\frac{1}{2}$ | $-i^N e^{\frac{i}{2L}(\pi - \sigma)}$ | $\frac{1}{4} - \frac{\sigma}{4\pi}$ |
| $(0, \frac{1}{2})$ | $\frac{1}{2}$ | $i^N e^{-\frac{i}{2L}(\pi - \sigma)}$ | $\frac{1}{4} - \frac{\sigma}{4\pi}$ |
| $(\frac{1}{2}, 0)$ | $\frac{1}{2}$ | $-i^N e^{\frac{i}{2L}(\pi + \sigma)}$ | $\frac{1}{4} + \frac{\sigma}{4\pi}$ |
| $(0, -\frac{1}{2})$ | $-\frac{1}{2}$ | $i^N e^{-\frac{i}{2L}(\pi + \sigma)}$ | $\frac{1}{4} + \frac{\sigma}{4\pi}$ |
| $(-\frac{1}{2}, 1)$ | $\frac{1}{2}$ | $-i^N e^{-\frac{i}{2L}(3\pi - \sigma)}$ | $\frac{5}{4} - \frac{3\sigma}{4\pi}$ |
| $(-1, \frac{1}{2})$ | $-\frac{1}{2}$ | $i^N e^{\frac{i}{2L}(3\pi - \sigma)}$ | $\frac{5}{4} - \frac{3\sigma}{4\pi}$ |

where we introduced a convenient phases relative to the definitions in section 5.2. They satisfy

$$Z^2 = X^2 = 1,$$
$$ZX = (-1)^L XZ. \tag{194}$$

(Our sign choices in (193) are such that the relations in the first line have no phases for all $L$.) We see that for odd $L$, the $\mathbb{Z}_2 \times \mathbb{Z}_2^{\mathcal{R}}$ symmetry is realizes projectively.

Since we can relabel $x \to y \to z$ (with appropriate changes of the coupling constants in the Hamiltonian (192)), without loss of generality, we can limit the discussion to a spatial twist by $g = Z$, corresponding to $\sigma = \pi$. The Hamiltonian with a $Z$-defect on the link $(L, 1)$ is

$$H(Z) = \frac{1}{2} \sum_{j \neq L} \left(\lambda_x \tau_j^x \tau_{j+1}^x + \lambda_y \tau_j^y \tau_{j+1}^y + \lambda_z \tau_j^z \tau_{j+1}^z\right) + \frac{1}{2}\left(-\lambda_x \tau_L^x \tau_1^x - \lambda_y \tau_L^y \tau_1^y + \lambda_z \tau_L^z \tau_1^z\right). \tag{195}$$

It commutes with $Z$ and $X$ of (193) of the untwisted problem and therefore there is no need to write $Z(Z)$ or $X(Z)$.

Using a compact notation, we will label the system with twist $g = Z^\ell$ by the integer $\ell$ (where $\ell \sim \ell + 2$). Using (162) and recalling our sign convention in (193), the translation generator is

$$T(\ell) = (\tau_1^z)^\ell T. \tag{196}$$

Then, using the relations (194) (or the results in section 5.3 by restricting the twists to be in the $\mathbb{Z}_2 \times \mathbb{Z}_2^{\mathcal{R}}$ subgroup), we find

$$T(\ell)X = (-1)^\ell XT(\ell),$$
$$\mathcal{P}T(\ell) = T(\ell)^{-1}\mathcal{P}, \tag{197}$$
$$T(\ell)^L = Z^\ell.$$

These relations lead to constraints on the partition functions[35]

$$\mathcal{Z}(\beta, L, \ell, k, m, n) = \text{Tr}[e^{-\beta H(\ell)} Z^k X^m T(\ell)^n]. \tag{198}$$

Following the steps in section 5.4, we find

$$
\begin{aligned}
\mathcal{Z}(\beta, L, \ell, k, m, n+L) &= \mathcal{Z}(\beta, L, \ell, k+\ell, m, n), \\
\mathcal{Z}(\beta, L, \ell, k, m, -n) &= \mathcal{Z}(\beta, L, \ell, k, m, n), \\
\mathcal{Z}(\beta, L, \ell, k, m, n) &= (-1)^{n\ell+kL} \mathcal{Z}(\beta, L, \ell, k, m, n), \\
\mathcal{Z}(\beta, L, \ell, k, m, n) &= (-1)^{mL} \mathcal{Z}(\beta, L, \ell, k, m, n), \\
\mathcal{Z}(\beta, L, \ell, k, m, n) &= (-1)^{m\ell} \mathcal{Z}(\beta, L, \ell, k, m, n),
\end{aligned} \tag{199}
$$

and of course, $\mathcal{Z}$ is periodic in $\ell$, $k$, and $m$. The first relation uses the last equation in (197) and the others can be derived by inserting in the trace $1 = \mathcal{P}\mathcal{P}$, or $1 = XX$, or $1 = ZZ$, or $1 = T(\ell)T(\ell)^{-1}$ and circling one of the factors around the trace. Therefore,

$$
\begin{aligned}
\mathcal{Z}(\beta, L, \ell, k, 0, n) &= 0, && \text{for} && n\ell + kL \in 2\mathbb{Z}+1, \\
\mathcal{Z}(\beta, L, \ell, k, 1, n) &= 0, && \text{for} && L \in 2\mathbb{Z}+1 \quad \text{or} \quad \ell = 1.
\end{aligned} \tag{200}
$$

As above, the minus signs and the vanishing partition functions signal anomalies. They are related to the fact that states cancel each other in the trace. These results are independent of $\beta$ and the parameters in the Hamiltonian, as long as the internal $G = \mathbb{Z}_2 \times \mathbb{Z}_2^{\mathcal{R}}$ symmetry is preserved. Therefore, they should be true also in the low-energy theory, thus leading to constraints on the low-energy behavior. In particular, the ground state cannot be unique.

It is known that the XYZ model is generally gapped (see e.g., [59]), except when the symmetry is enhanced to O(2) or SO(3). Because of the 't Hooft anomaly, gapped phases must spontaneously break the global symmetry $\mathbb{Z}_2 \times \mathbb{Z}_2^{\mathcal{R}} \times \mathbb{Z}_L$. For example, when $\lambda_z$ is dominant, the ground state has anti-ferromagnetic order so $\langle \tau_j^z \rangle \sim (-1)^j n_z$ where $n_z \neq 0$. In this case, both $\mathbb{Z}_L$ and $\mathbb{Z}_2^{\mathcal{R}}$ are broken spontaneously.

To conclude, the internal symmetry of this model is $G = \mathbb{Z}_2 \times \mathbb{Z}_2^{\mathcal{R}}$. Adding the translation symmetry, the symmetry is $\mathcal{G} = \mathbb{Z}_2 \times \mathbb{Z}_2^{\mathcal{R}} \times \mathbb{Z}_L$. For even $L$ this symmetry is realized linearly and for odd $L$, it is realized projectively. For all $L$, the symmetry of the twisted Hilbert space is $\mathcal{G}_Z = \mathbb{Z}_2 \times \mathbb{Z}_{2L}$.[36] This change in the symmetry group is not an anomaly. However, the fact that this symmetry is realized projectively is an anomaly.

## 5.7 Another $G = \mathbb{Z}_2 \times \mathbb{Z}_2$ example

Recently, a model of a spin-1/2 chain with $\mathbb{Z}_2 \times \mathbb{Z}_2$ symmetry has been studied [60,61], whose phase diagram resembles the deconfined quantum criticality in 2+1 spin systems. Its Hamiltonian is

$$H = -\sum_j (\tau_j^x \tau_{j+1}^x + \lambda_z \tau_j^z \tau_{j+1}^z) + \frac{1}{2}\sum_j (\tau_j^x \tau_{j+2}^x + \tau_j^z \tau_{j+2}^z). \tag{201}$$

As one tunes $\lambda_z$, there is a continuous phase transition at $\lambda_z \approx 1.4645$ between a valence bond solid and a ferromagnetic phase.

---

[35]Comparing with the notation in section 5.4, some of the arguments should be multiplied by $\pi$ and the partition function with $m = 1$ is related to the partition function denoted as $\mathcal{Z}^{\mathcal{R}}$ in (177).

[36]For the untwisted problem with odd $L$, the symmetry can also be written as $\mathcal{G} = \mathbb{Z}_2 \times \mathbb{Z}_{2L}$, but this hides the role of the preferred generator $T$. In the twisted problem, $\mathbb{Z}_2 \times \mathbb{Z}_{2L}$ is not isomorphic to $\mathbb{Z}_2 \times \mathbb{Z}_2 \times \mathbb{Z}_L$ for even $L$.

This model has a $\mathbb{Z}_2^X \times \mathbb{Z}_2^S$ internal symmetry generated by

$$X = \prod_j \tau_j^x \,,$$
$$S = \prod_j \tau_j^z \,. \tag{202}$$

It is realized projectively for odd $L$:

$$X^2 = S^2 = 1 \,,$$
$$XS = (-1)^L SX \,. \tag{203}$$

Let us consider an $S$ defect at the link $(J, J+1)$. The Hamiltonian is

$$
\begin{aligned}
H^{(J,J+1)}(S) = & -\sum_{j \neq J} (\tau_j^x \tau_{j+1}^x + \lambda_z \tau_j^z \tau_{j+1}^z) - (-\tau_J^x \tau_{J+1}^x + \lambda_z \tau_J^z \tau_{J+1}^z) \\
& + \frac{1}{2} \sum_{j \neq J-1, J} (\tau_j^x \tau_{j+2}^x + \tau_j^z \tau_{j+2}^z) \\
& + \frac{1}{2}(-\tau_{J-1}^x \tau_{J+1}^x + \tau_{J-1}^z \tau_{J+1}^z) + \frac{1}{2}(-\tau_J^x \tau_{J+2}^x + \tau_J^z \tau_{J+2}^z) \,.
\end{aligned}
\tag{204}
$$

To see that it is topological, note that

$$H^{(J-1,J)}(S) = \tau_J^z H^{(J,J+1)}(S) \tau_J^z \,. \tag{205}$$

The translation symmetry in the presence of the defect is generated by $T(S) = \tau_{J+1}^z T$. It satisfies

$$T(S)^L = S \,,$$
$$T(S)S = ST(S) \,,$$
$$T(S)X = -XT(S) \,. \tag{206}$$

Here we did not write $S(S)$ and $X(S)$ because they are the same as in the untwisted theory. The minus sign in this last equation reflects an LSM anomaly between the lattice translation symmetry and the internal $\mathbb{Z}_2^X \times \mathbb{Z}_2^S$ symmetry and it leads to its usual consequences.

This example is similar to the one in section 5.6.2, but it differs from it in a crucial way. The easiest way to see that is to compare the system with its continuum limit. The critical theory is again a $c = 1$ boson [60, 61], where the symmetries of the lattice model are realized as

$$T = \mathcal{R} e^{\frac{2\pi i}{L} P} \,,$$
$$X = (-1)^{Q_w} \,,$$
$$S = (-1)^{Q_m} \mathcal{R} \,. \tag{207}$$

As the Heisenberg chain, we have an LSM anomaly between the internal $\mathbb{Z}_2^X \times \mathbb{Z}_2^S$ and the lattice translation symmetries. However, in this case the emanant $\mathbb{Z}_2$ symmetry that arises from lattice translation symmetry, denoted by $\mathcal{R}$, has no self-anomaly. Therefore, our understanding suggests that the spectrum of the theory at large $L$ varies with $L$ mod 2 rather than with $L$ mod 4 (see also section 6.1) and that the ground state of the system always has vanishing lattice momentum, i.e., its $T$ eigenvalue is equal to one.

These assertions can be verified using exact diagonalization. For example, for $L = 2$, the Hamiltonian (201) simplifies and for $\lambda_z > 0$, the ground state is

$$\frac{1}{\sqrt{2}}\big(|\uparrow\uparrow\rangle + |\downarrow\downarrow\rangle\big), \tag{208}$$

whose eigenvalues of $T$, $X$, and $S$ are all equal to one. Note that this is unlike the ground state of the $L = 2$ Heisenberg chain in (187).

We will see another example of this phenomenon in section 6.2.

# 6 Generalizations

## 6.1 An internal emanant $\mathbb{Z}_n^C$ from $\mathbb{Z}_L$ lattice translation and the $H^3(\mathbb{Z}_n, \mathrm{U}(1))$ anomaly

In the SO(3) spin chain and its various deformations in section 5, we started with a lattice system with a $\mathbb{Z}_L$ translation symmetry. In the continuum limit, $L \to \infty$, we ended up with a theory with an internal emanant $\mathbb{Z}_2^C$ symmetry and a continuous translation U(1) symmetry. This continuum limit depends on $N = L \bmod 4$.

The fact that there are 4 continuum limits rather than 2 continuum limits was associated with an anomaly of the internal $\mathbb{Z}_2^C$ symmetry of the continuum theory, which is given by the nontrivial element in $H^3(\mathbb{Z}_2, \mathrm{U}(1)) = \mathbb{Z}_2$. Here we would like to generalize this to a $\mathbb{Z}_n^C$ internal symmetry and phrase it in a more general way, such that it applies to many other models.

We do not claim that this is the most general way an internal symmetry can arise from lattice translation. Nor do we claim that additional consistency conditions cannot constrain our picture further.

### 6.1.1 $M$ different $L \to \infty$ limits

Before we start, we note that the $L \to \infty$ limit of the group $\mathbb{Z}_L$ is not unique. It depends on which representations we focus on. The representations of $\mathbb{Z}_L$ are labeled by an integer $k \bmod L$. The generator of $\mathbb{Z}_L$ is represented by $e^{\frac{2\pi i k}{L}}$. As we take $L \to \infty$, we can focus on representations with $k \sim L$ and then $\mathbb{Z}_L \to \mathbb{Z}$. Alternatively, we can focus on representations with fixed $k$ and then $\mathbb{Z}_L \to \mathrm{U}(1)$. In our case, this U(1) becomes the continuous spatial translation group. Below, we will focus on other representations and will find $\mathbb{Z}_L \to \mathbb{Z}_n^C \times \mathrm{U}(1)$.

We consider a general lattice system, focus on the low-lying energy eigenstates, and label them by $r$. Clearly, for different values of $L$, the index $r$ runs over different ranges. Then, we assume that there is a well-defined $L \to \infty$ limit through values of $L$ with fixed

$$N = L \bmod M, \tag{209}$$

for some finite integer $M$. (In the SO(3) example in section 5, we had $M = 4$.) This assumption means that the energies of the low-lying states $E_r$ with fixed $r$ have a good limit,[37] and the eigenvalues of $T$

$$T_r(N) = C_r(N)^{-1} e^{\frac{2\pi i P_r(N)}{L}} \tag{210}$$

are such that the momenta $P_r$ and the phases $C_r$ have good limits. As is clear from this expression, these eigenvalues can depend $N$, i.e., on how we take $L \to \infty$.

---

[37]This well-defined limit of the energies is up to $r$-independent but possibly $L$-dependent additive and multiplicative constants. These constants can be absorbed in renormalization.

In fact, we can conclude a stronger statement. Since

$$T_r(N)^L = 1,\tag{211}$$

the eigenvalues (210) must be exact even for finite $L$ and also

$$e^{2\pi i P_r(N)} = C_r(N)^L.\tag{212}$$

The assumption of a smooth limit with fixed $N$ leads to

$$e^{2\pi i P_r(N)} = C_r(N)^N,$$
$$C_r(N)^M = 1.\tag{213}$$

The equations (210) - (213) are exact. The values of $P_r(N)$ and $C_r(N)$ are independent of $L$. The only dependence on $L$ is through the range of $r$.

This can be interpreted as follows. The $\mathbb{Z}_L$ lattice symmetry leads at large $L$ and low energies to a global internal $\mathbb{Z}_M$ symmetry with generator $C$, whose eigenvalues are $C_r(N)$. The $M$ different ways of taking $L \to \infty$, which are labeled by $N$, correspond to $M$ different kinds of defects, associated with $M$ different twisted Hilbert spaces.[38]

Let us understand better the relevant group theory. This $\mathbb{Z}_M$ internal symmetry of the continuum theory is not a subgroup of the microscopic $\mathbb{Z}_L$. Even in the simple case of $N = 0$, i.e., $L = KM$ with integer $K$, where $\mathbb{Z}_L$ has a $\mathbb{Z}_M$ subgroup, generated by $T^K$, this is not the $\mathbb{Z}_M$ group that we are talking about.

Instead, our $\mathbb{Z}_M$ is some kind of a quotient. Let us clarify it. In the simple case of $L = KM$, consider the subgroup $\mathbb{Z}_K \subset \mathbb{Z}_L$. It is generated by $T^M$. It is an exact symmetry of the lattice model and every state has a well defined $\mathbb{Z}_K$ "charge," which is an integer modulo $K$. Since for large $L$ (and therefore also large $K$), the low-lying states have $\lim_{L \to \infty} T^M = 1$, the operators in this $\mathbb{Z}_K$ subgroup, $(T^M)^m$ with $m = 1, ..., K$, have a smooth $m \sim K \to \infty$ limit and lead to U(1). This U(1) symmetry is the translation symmetry of the continuum theory. In this limit, the integer modulo $K$ charge of the microscopic $\mathbb{Z}_K$ becomes the integer U(1) charge, which we interpret as the continuum momentum. (This is similar to the discussion at the beginning of this subsection, except that here $\mathbb{Z}_K \to$ U(1), rather than $\mathbb{Z}_L \to$ U(1).)

Now, we can identify our $\mathbb{Z}_M$ as the quotient

$$\mathbb{Z}_M = \frac{\mathbb{Z}_L}{\mathbb{Z}_K}, \qquad \text{for} \qquad L = KM.\tag{214}$$

It should be emphasized that this $\mathbb{Z}_M$ quotient is not a symmetry of the lattice model at finite $L$. In fact, we cannot even assign unambiguous $\mathbb{Z}_M$ transformation laws to all the states.

This situation is different for large $L$. Then, $K$ is also large and every low-energy state has a well-defined integer (rather than an integer modulo $K$) momentum charge. Hence, it has a well-defined transformation law under the quotient $\mathbb{Z}_M$. Consequently, this $\mathbb{Z}_M$ symmetry becomes meaningful for the low-lying states at large $L$, and then, as we said in section 1.3, it is an exact symmetry of the low-energy theory. Unlike a standard emergent symmetry, which can be violated by irrelevant operators, since it originates as a quotient of the exact underlying $\mathbb{Z}_L$, this $\mathbb{Z}_M$ symmetry is an emanant symmetry and is not violated by irrelevant operators.

When $N = L \bmod M \neq 0$, i.e., $L = N + KM$ with integer $K$, $T^L = 1$ leads to

$$T^{MK} = T^{-N}, \qquad \text{for} \qquad L = N + KM.\tag{215}$$

For large $L$, the left hand side becomes a complete translation of space and the right hand side can be though of as a twist by the element $T^{-N}$.

---

[38]Note that this interpretation is consistent with the picture that a change in $L$ is related to a twist in a global symmetry. As stated in footnote 14, this picture is meaningful only at large $L$.

### 6.1.2 Going from $\mathbb{Z}_M$ to $\mathbb{Z}_n^C$

Our experience in section 5 shows that this picture should be supplemented with a crucial element, as this new $\mathbb{Z}_M$ internal symmetry might not act faithfully on the operators in the theory or the states in the untwisted theory.

Consider the limit with $N = 0$, corresponding to the untwisted theory, i.e., the theory with no defect. It could happen that the collection of phases $\{C_r(0)\}$ satisfies a stronger relation

$$C_r(0)^n = 1, \qquad \text{for all} \qquad r. \tag{216}$$

(Recall that this entire discussion is about the low-energy spectrum at large $L$.) This means that the Hilbert space of the untwisted continuum theory realizes only the quotient

$$\mathbb{Z}_n^C = \frac{\mathbb{Z}_M}{\mathbb{Z}_{M/n}}. \tag{217}$$

If this is indeed the case, the global emanant symmetry that arises from the $\mathbb{Z}_L$ lattice translation is $\mathbb{Z}_n^C$ rather than $\mathbb{Z}_M$ and we can expect to have only $n$ twisted theories. How come we have $M$ rather than $n$ twisted Hilbert spaces?

Our understanding of the case with $M = 4$ and $n = 2$ discussed in section 5 suggests an answer to this question, which we will present now. As we said above, this picture might not be the most general way things could happen, nor do we claim that we imposed all possible consistency conditions on it.

Let us assume that the limits of the energies and the momenta $P_r(N)$ depend only on

$$I = L \bmod n = N \bmod n. \tag{218}$$

These correspond to the $n$ twisted problems associated with the group elements of the internal $\mathbb{Z}_n^C$ symmetry of the continuum theory. The difference between the $\frac{M}{n}$ limits with the same $I$ but different $N$ is only in the eigenvalues of $T$, which become the eigenvalues of $C$. Furthermore, we allow the $\mathbb{Z}_M$ phases $C$ that realize the $\mathbb{Z}_n^C$ quotient (217) to act projectively.

More explicitly, limits with different $N$, but with the same $I = N \bmod n$, lead to almost identical theories. The only difference between them is that the eigenvalues of $C$ are different

$$C_r(N + n) = \zeta(N)C_r(N). \tag{219}$$

The phases $\zeta(N)$ are independent of the state labeled by $r$ and are $n$th root of one because the global symmetry is $\mathbb{Z}_n$

$$\zeta(N)^n = 1. \tag{220}$$

Using this,

$$C_r(N + n)^n = C_r(N)^n. \tag{221}$$

Since the momenta $P_r(N)$ are the same as $P_r(N + n)$, the first equation in (213) leads to $C_r(N + n)^{N+n} = C_r(N)^N$ and hence

$$
\begin{aligned}
C_r(N)^n &= \zeta(N)^{-I}, \\
\zeta(N + n)^I &= \zeta(N)^I.
\end{aligned}
\tag{222}
$$

The relations (222) characterize the way the global $\mathbb{Z}_n^C$ symmetry is realized projectively[39] in Hilbert spaces with $N \neq 0 \bmod n$,

$$
\begin{aligned}
C_r(N)^n &= \omega(N) = \zeta(N)^{-I}, \\
\omega(N + n) &= \omega(N).
\end{aligned}
\tag{223}
$$

---

[39]We should clarify the following point. The continuum operator $C$ generate the group $\mathbb{Z}_n^C$. A representation with $C(N)^n = e^{i\varphi} \neq 1$ is not a standard projective representation because we can remove the phase by redefining $C(N) \to e^{-\frac{i\varphi}{n}} C(N)$. However, for us, the representation $C(N)$ has a preferred normalization, as the large $L$ limit of the lattice translation generator $T(N)$, and therefore such a redefinition is not possible.

We see that the "projectiveness" of $\mathbb{Z}_n^C$, which is given by $\omega(N) = \zeta(N)^{-I}$, depends only on $I = N$ mod $n$, which is consistent with the picture that the limits with $N$ and $N + n$ are almost the same.

The phases $\zeta(N)$ and $\omega(N)$ can be interpreted as follows. In the absence of any anomaly, the $\mathbb{Z}_n^C$ symmetry in the continuum limit would imply that the fusion of $n$ defects or $n$ operators is trivial. Instead, the phase $\zeta(N)$ signals an anomaly in the action of $C$, which is visible when fusing $n$ defects. And $\omega(N)$ is an anomaly in the $\mathbb{Z}_n^C$ action associated with fusing $n$ operators.

It is possible that a more detailed analysis can constrain these phases further. Instead of following such an analysis, let us assume that the resulting continuum low-energy theory is relativistic. Then, defects and operators are different manifestations of the same object and we can relate $\zeta(N)$ and $\omega(N)$. Specifically, consider the system with one defect, corresponding to $N = 1$, and act on it with the group element $C^{P+n}$ for some integer $P$. It acts as $C(1)^{P+n} = \omega(1)C(1)^P$. Performing a Euclidean spacetime rotation, the left hand side of this expression $C(1)^{P+n}$ turns into $C^{-1}$ acting on a system with $N = P + n$ defects, i.e., $C(P+n)^{-1} = \zeta(P)^{-1}C(P)^{-1}$. And the right hand side $\omega(1)C(1)^P$ turns into $\omega(1)C^{-1}$ acting on $N = P$ defects, i.e., $\omega(1)C(P)^{-1}$. We learn that relativistic invariance leads to $\zeta(P) = \omega(1)^{-1}$ and hence, it is independent of $P$.

We conclude that if the limit leads to a relativistically invariant system, it is labeled by a single $\mathbb{Z}_n$ phase $\zeta$ and then

$$
\begin{aligned}
C_r(N + n) &= \zeta C_r(N), \\
C_r(N)^n &= \zeta^{-N}, \\
\zeta^n &= 1.
\end{aligned}
\tag{224}
$$

Consequently, $M$ could to be as large as $n^2$.

We see that the internal global symmetry in the limit of large $L$ is $\mathbb{Z}_n^C$ and it is realized faithfully in the untwisted theory, labeled by $N = 0$. If $n = M$, there are $n$ twisted theories labeled by $N = 0, 1, ..., n-1$ and the symmetry is realized faithfully and linearly in all of them. If $n \neq M$, there are $M$ twisted theories labeled by $N = 0, 1, ..., M-1$. The symmetry is realized linearly when $N = 0$ mod $n$ (equivalently, $I = 0$) and is realized projectively otherwise. In that case, this is the lattice precursor of the pure $\mathbb{Z}_n^C$ anomaly of the continuum theory.

Let us state it more explicitly from the perspective of the continuum theory at infinite $L$. As we discussed in section 2.3, the twisted theory depends on the conjugacy class of the spatial twist $g \in G$. (Additional parameters labeling the spatial twist correspond to unitary transformations on that Hilbert space.) Here, we twist by $C^N$ and as a group element $C^n = 1$. Therefore, the twisted theory depends on $N$ mod $n$, rather than on $N$ mod $M$. However, the operators are mapped nontrivially under $N \to N + n$ and hence the larger periodicity.

This larger periodicity is related to the anomaly of the continuum theory. In the continuum, a system with a $\mathbb{Z}_n$ global symmetry can have an anomaly classified by $\mathcal{H}^3(\mathbb{Z}_n, \mathrm{U}(1)) = \mathbb{Z}_n$. Denoting the $\mathbb{Z}_n$ class by $p = 0, 1, \cdots, n-1$, the momenta of the states in the fundamental defect Hilbert space are in $P_r(1) = \frac{c_r}{n} + \frac{p}{n^2} + \mathbb{Z}$ with $c_r = 0, 1, \cdots, n-1$ and the corresponding $C$ eigenvalues are $C_r(1) = e^{\frac{2\pi i}{n}(c_r + \frac{p}{n})}$ [45,62]. This agrees with our general discussion above with $\zeta = e^{-\frac{2\pi i p}{n}}$ in (224).[40]

Let us summarize this discussion. We started with a lattice model with a $\mathbb{Z}_L$ translation symmetry generated by $T$. The continuum theory has an emanant $\mathbb{Z}_n^C$ internal symmetry, which is not present on the lattice. In the continuum limit, this $\mathbb{Z}_n^C$ can have an anomaly characterized

---

[40]Note that depending on $n$, we could choose different generators $C$. This would change the value of $p$ and mix the twisted Hilbert spaces. However, since the emanant $\mathbb{Z}_n^C$ symmetry arises from lattice translation, as in (210), the generator $C$ has a natural microscopic definition and cannot be redefined.

by $H^3(\mathbb{Z}_n^C, U(1)) = \mathbb{Z}_n$. Since the $\mathbb{Z}_n^C$ symmetry is not present on the lattice, it might appear challenging to find the lattice precursor of this anomaly.

Our analysis of the $L \to \infty$ limit addressed this challenge. We focused on the low-energy states for large but finite $L$ and examined their $T$ eigenvalues. They are given exactly by (210)

$$T_r(N) = C_r(N)^{-1} e^{\frac{2\pi i P_r(N)}{L}}, \qquad N = L \bmod M. \tag{225}$$

Hence, for infinite $L$, the lattice translation operator $T$ acts on the low-lying states as the internal symmetry generator $C(N)^{-1}$. For $N = 0 \bmod n$, the $\mathbb{Z}_n^C$ generated by $C(N)$ acts linearly. And for $N \neq 0 \bmod n$, it acts projectively. This means that for large but finite $L$, the eigenvalues of $T$ on the low-lying states (e.g., the ground state lattice momentum) characterize both the internal emanant $\mathbb{Z}_n^C$ symmetry and its anomaly.

This is another sign that we should not think about the $\mathbb{Z}_n^C$ symmetry as an emergent symmetry and its anomaly as an emergent anomaly. Instead, it has exact consequences on the lattice and leads to an exact symmetry and an exact anomaly in the continuum.

### 6.1.3 Example: SU($n$) spin chain

Let us demonstrate these ideas with the SU($n$) spin chain, where each site transforms as the fundamental representation of SU($n$). The Hamiltonian describes antiferromagnetic Heisenberg interactions of nearest-neighbor spins. The internal global symmetry of the lattice model is PSU($n$) = SU($n$)/$\mathbb{Z}_n$ and another emanant $\mathbb{Z}_n^C$ symmetry arises from lattice translation.

The continuum theory of this model is SU($n$)$_1$ WZW model [43, 58, 63], whose global symmetry is $G_{\text{SU}(n)} = \frac{\text{SU}(n) \times \text{SU}(n)}{\mathbb{Z}_n}$. The microscopic PSU($n$) symmetry is the diagonal subgroup of $G_{\text{SU}(n)}$ and the emanant $\mathbb{Z}_n^C$ symmetry is in the center of one of the SU($n$) factors.

Let us review some facts about the IR CFT. SU($n$)$_1$ Kac-Moody has $n$ representations labeled by $\ell = 0, 1, ..., n-1$, which we denote as $\mathcal{V}_\ell$. (When comparing with the discussion of $n = 2$ in section 3.4, substitute $\ell = 2j$.) The conformal dimension of the representation $\mathcal{V}_\ell$ is $h_n(\ell) = \frac{\ell(n-\ell)}{2n} \bmod 1$. In the expressions below, we identify $\ell \sim \ell + n$. Using this notation, the Hilbert space of the theory is given by

$$\bigoplus_{\ell=0}^{n-1} \mathcal{V}_\ell \otimes \overline{\mathcal{V}}_{n-\ell}. \tag{226}$$

A twist by the $\mathbb{Z}_n^C$ generator $N$ times maps $\mathcal{V}_\ell \otimes \overline{\mathcal{V}}_{n-\ell} \to \mathcal{V}_{\ell+N} \otimes \overline{\mathcal{V}}_{n-\ell}$. Therefore, the Hilbert space of the twisted theory transforms as

$$\bigoplus_{\ell=0}^{n-1} \mathcal{V}_{\ell+N} \otimes \overline{\mathcal{V}}_{n-\ell}. \tag{227}$$

The momenta of the states in $\mathcal{V}_{\ell+N} \otimes \overline{\mathcal{V}}_{n-\ell}$ are

$$P_\ell(N) = h_n(\ell + N) - h_n(n - \ell) = \frac{N(n - N - 2\ell)}{2n} \bmod 1. \tag{228}$$

An anomaly is the obstruction to satisfying the two relations

$$\begin{aligned} e^{2\pi i P(N)} &= C(N)^N, \\ C(N)^n &= 1. \end{aligned} \tag{229}$$

Clearly, the eigenvalue of $C(N)$ of the states in $\mathcal{V}_{\ell+N} \otimes \overline{\mathcal{V}}_{n-\ell}$ is given by $e^{-\frac{2\pi i \ell}{n}}$ times an $\ell$ independent phase. We can try to adjust this phase so that we can satisfy (229).

For $n$ odd, we can satisfy these conditions with

$$C_\ell(N) = e^{2\pi i \left( \frac{Nn - N - 2\ell}{2n} \right)}, \tag{230}$$

such that

$$
\begin{aligned}
e^{2\pi i P_\ell(N)} &= C_\ell(N)^N, \\
C_\ell(N)^n &= 1, \qquad n \in 2\mathbb{Z} + 1.
\end{aligned} \tag{231}
$$

On the other hand, for even $n$, there is no choice of the phase in $C(N)$ satisfying the two requirements (229). For example, with the same choice as for odd $n$, (230), we have

$$
\begin{aligned}
e^{2\pi i P_\ell(N)} &= C_\ell(N)^N, \\
C_\ell(N)^n &= (-1)^N, \qquad n \in 2\mathbb{Z}.
\end{aligned} \tag{232}
$$

We conclude that there is no anomaly for odd $n$, while for even $n$, there is $\mathbb{Z}_2$ rather than the more general $\mathcal{H}^3(\mathbb{Z}_n^C, U(1)) = \mathbb{Z}_n$ anomaly. This is consistent with the discussion in [64,65], where it was shown that center symmetry in any WZW CFT can have at most $\mathbb{Z}_2$ anomaly.

Let us relate this continuum analysis to the lattice discussion in sections 6.1.1 and 6.1.2. The lack of anomaly for odd $n$ means that we should set $M = n$ in Eq. (213) and find agreement with Eq. (229). This is to be contrasted with the anomalous case of even $n$, where $M = 2n$. In this case, Eq. (213) agrees with Eq. (232). A more detailed comparison, shows that $\zeta$ in Eq. (224) is $\zeta = (-1)^{n-1}$. The same result was observed in a deformation of the SU($n$) Heisenberg chain with the same low-energy theory [66].

## 6.2 Mixture of symmetries

The goal of this section is to study further the model of [32], which we discussed in section 4.2. In particular, we will study its emanant symmetry that arises from lattice translation and what happens for $L \neq 0 \mod 4$. We will do it by starting with the discussion of the XXZ model (8) in section 5.6.1. And then we will gauge an appropriate $\mathbb{Z}_2$ symmetry to find the answer for the model of [32].

As we reviewed around Eq. (8) and in section 5.6.1, the XXZ model flows to the $c = 1$ compact boson theory with radius (10). The global O(2) symmetry of the lattice model is enhanced in the continuum to $\left( U(1)_m \times U(1)_w \right) \rtimes \mathbb{Z}_2^{\mathcal{R}}$. We follow the notation in section 4.2.3 and denote the U(1) charges by $\tilde{Q}_m$ and $\tilde{Q}_w$. The U(1)$_m$ charge $\tilde{Q}_m$ is the U(1) charge of the lattice model and the U(1)$_w$ charge $\tilde{Q}_w$ is emergent. As in section 5.6.1, the emanant $\mathbb{Z}_2^{\tilde{C}}$ symmetry, which is generated by (11)

$$\tilde{C} = e^{i\pi(\tilde{Q}_m + \tilde{Q}_w)}, \tag{233}$$

arises from lattice translation. (We denote this symmetry generator by $\tilde{C}$ rather than by $C$ for the same reason that the charges in the right hand side of this equation are denoted by $\tilde{Q}_m$ and $\tilde{Q}_w$.)

As in the discussion in section 5.6.1, the system has a modulo four periodicity in $L$ as we take $L$ to infinity. It is interpreted as leading to the continuum theory with a twist by $\tilde{C}^N$ (with $N = L \mod 4$). The fact that a twist by $\tilde{C}$ to an even power is nontrivial, reflects the anomaly of the continuum $\mathbb{Z}_2^{\tilde{C}}$ symmetry.

Next, we follow sections 4.2.3 and gauge $\mathbb{Z}_2 \subset U(1)_m$ to find the Hamiltonian (133). The lattice system has an internal $U(1) \times \mathbb{Z}_2^X$ symmetry. This system flows to the $c = 1$ boson with half the radius with U(1) charges (148)

$$Q_m = \frac{1}{2}\tilde{Q}_m, \quad Q_w = 2\tilde{Q}_w, \quad Q_m, Q_w \in \mathbb{Z}. \tag{234}$$

Recall that as in section 4.2.3, the lattice charge $\mathcal{Q}$ was identified with the continuum charge $Q_m$.

More interesting is the role of the $\mathbb{Z}_2$ symmetry generated by $\tilde{C}$ of Eq. (233)

$$\tilde{C} = e^{i\pi(\tilde{Q}_m + \tilde{Q}_w)} = e^{i\pi(2Q_m + \frac{1}{2}Q_w)} = e^{\frac{i\pi}{2}Q_w}. \tag{235}$$

We see that after the gauging, it becomes a $\mathbb{Z}_4^{\tilde{C}} \subset \mathrm{U}(1)_w$ symmetry, satisfying

$$\tilde{C}^2 = X. \tag{236}$$

Furthermore, this $\mathbb{Z}_4^{\tilde{C}}$ symmetry is anomaly free, as it is a subgroup of $\mathrm{U}(1)_w$.

Let us study the continuum theory with a twist using the group element $g = \tilde{C}^N e^{i\sigma Q_m}$ corresponding to $(\eta_m, \eta_w) = \left(\sigma, \frac{\pi N}{2}\right)$. In order to identify how it acts in the twisted theory, we follow the discussion around Eq. (54) and write

$$\tilde{C}\left(\sigma, \frac{\pi N}{2}\right) = e^{\frac{i\pi}{2}Q_w(0)}. \tag{237}$$

With this choice, we have

$$
\begin{aligned}
U\left(\sigma, \frac{\pi N}{2}\right) &= e^{i\sigma Q_m\left(\frac{\pi N}{2}\right)} e^{\frac{i\pi N}{2}Q_w(0)}, \\
\tilde{C}\left(\sigma, \frac{\pi}{2}\right)^2 &= X\left(\sigma, \frac{\pi}{2}\right), \\
X\left(\sigma, \frac{\pi}{2}\right) &= e^{i\pi Q_w(0)}, \\
\tilde{C}\left(\sigma, \frac{\pi}{2}\right)^4 &= 1,
\end{aligned}
\tag{238}
$$

without anomalous phases. (Compare with the discussion around (65).)

This allows us to identify the continuum limit of the lattice model. We take $L$ to infinity with fixed $N = L \bmod 4$. Then, this discussion means that the resulting continuum theory is the $c = 1$ boson twisted by $e^{i\frac{\pi N}{2}Q_w}$. Therefore, we suggest the exact expression for lattice translation

$$T\left(\sigma, \frac{\pi N}{2}\right) = \tilde{C}\left(\sigma, \frac{\pi N}{2}\right)^{-1} e^{\frac{2\pi i}{L}P\left(\sigma, \frac{\pi N}{2}\right)} = e^{-\frac{i\pi}{2}Q_w(0)} e^{\frac{2\pi i}{L}P\left(\sigma, \frac{\pi N}{2}\right)}. \tag{239}$$

Note that unlike the discussion with anomalous $C$, here we do not have factors like $i^N$. Instead, the modulo four periodicity arises because the symmetry is $\mathbb{Z}_4$ and it is anomaly free. This is consistent with the general discussion in section 6.1 with $M = n = 4$. As another check, the ground state with $Q_w(0) = 0$ and $P = 0$ has $T = 1$.[41]

The internal emanant symmetry that arises from $T$ is $\mathbb{Z}_4^{\tilde{C}} \subset \mathrm{U}(1)_w$. Interestingly, its $\mathbb{Z}_2^X$ subgroup is realized microscopically as an internal symmetry $X = e^{i\pi Q_w}$. This is consistent with Eq. (239), which shows that for large $L$, $T^2$ acts as $X = e^{i\pi Q_w}$.

The identification of $N = L \bmod 4$ implies that a twist by $e^{i\frac{\pi L}{2}Q_w} = e^{i\frac{\pi N}{2}Q_w}$ quantizes $Q_m$ to $\mathbb{Z} + \frac{L}{4}$. This quantization can also be seen directly on the lattice from the following operator identity:

$$e^{2\pi i \mathcal{Q}} = i^L. \tag{240}$$

---

[41]The situation here is similar to the one in the $\mathbb{Z}_2 \times \mathbb{Z}_2$ model in section 5.7. There, the internal symmetry of the continuum theory that originates from the $\mathbb{Z}_L$ lattice translation also does not have an anomaly and hence the ground state momentum vanishes.

For $L = 2$ mod 4, $\mathcal{Q}$ is quantized to half integers. In this case, the $\mathbb{Z}_2^{\mathcal{R}}$ symmetry still exists, and the system transforms as a projective representation of the $O(2) = U(1) \rtimes \mathbb{Z}_2^{\mathcal{R}}$ symmetry. In the continuum, this corresponds to $\frac{\eta_w}{2\pi} = \frac{1}{2}$ mod 1, which we discussed in section 3.3.

For odd $L$, $\mathcal{Q}$ is quantized to $\mathbb{Z} + \frac{1}{4}$ or $\mathbb{Z} + \frac{3}{4}$. In this case, there cannot be any symmetry associated with $\mathcal{R}$. (See also the comment after Eq. (130).) This is in agreement with the continuum picture, where $L$ odd corresponds to $\frac{\eta_w}{2\pi} = \pm\frac{1}{4}$ mod 1.

The spectrum of the system depends on $N = L$ mod 4. Let us demonstrate the pattern for small sizes $L = 2, 3, 4$. We diagonalize the Hamiltonian (132) with $\lambda_z = 0$, and write the wavefunctions in the basis with diagonal $\sigma_j^z = \uparrow, \downarrow$. Then, we can use

$$X = \prod_j \sigma_j^x, \qquad \mathcal{Q} = \frac{1}{4}\sum_j \sigma_j^z \sigma_{j+1}^z, \tag{241}$$

and for even $L$ also

$$\mathcal{R} = \prod_{j=1}^{L/2} \sigma_{2j}^x, \tag{242}$$

to find the quantum numbers of the low-lying states.

For $L = 2$, the Hamiltonian vanishes. In fact, in this case, even the term with $\lambda_z$ vanishes. We find four degenerate states

$$
\begin{aligned}
\frac{1}{\sqrt{2}}\Big(|\uparrow\uparrow\rangle + |\downarrow\downarrow\rangle\Big), &\quad \text{with} \quad T = +1, \quad X = +1, \quad \mathcal{Q} = +\frac{1}{2}, \\
\frac{1}{\sqrt{2}}\Big(|\uparrow\downarrow\rangle + |\downarrow\uparrow\rangle\Big), &\quad \text{with} \quad T = +1, \quad X = +1, \quad \mathcal{Q} = -\frac{1}{2}, \\
\frac{1}{\sqrt{2}}\Big(|\uparrow\uparrow\rangle - |\downarrow\downarrow\rangle\Big), &\quad \text{with} \quad T = +1, \quad X = -1, \quad \mathcal{Q} = +\frac{1}{2}, \\
\frac{1}{\sqrt{2}}\Big(|\uparrow\downarrow\rangle - |\downarrow\uparrow\rangle\Big), &\quad \text{with} \quad T = -1, \quad X = -1, \quad \mathcal{Q} = -\frac{1}{2}.
\end{aligned}
\tag{243}
$$

Even though the system becomes highly degenerate for $L = 2$, the result is still consistent with the continuum theory. Here we should compare our lattice answers with the continuum theory twisted by $X = e^{i\pi Q_w}$. Its lowest energy states have $Q_m = \pm\frac{1}{2}$, $Q_w = 0$ and hence $T = X = 1$, as in the first two states in (243). The two other states can be identified with continuum states with $Q_m = \frac{1}{2}$, $Q_w = 1$ and hence $T = -X = 1$, and another state, related to it by $\mathcal{R}$, with $Q_m = -\frac{1}{2}$, $Q_w = -1$ and hence $T = X = -1$. The fact that these two excited states of the continuum theory are degenerate with the ground states and that other states of the continuum theory are absent are lattice artifact that are corrected at larger values of $L$.

We can also add an $X$ defect. Now the Hamiltonian is nontrivial. Using (136), it is

$$H(X) = 2(\sigma_1^x + \sigma_2^x). \tag{244}$$

And the symmetry operators are $\mathcal{Q}(X) = Q(1) - \frac{1}{2}\sigma_1^z\sigma_2^z = 0$ and $T(X) = -\sigma_1^x T$. As we mentioned above, there is phase freedom in the definition of $T(X)$. Here we chose a convenient phase, so the ground state has $T(X) = 1$.

The eigenstates of the Hamiltonian are

$$
\begin{aligned}
\frac{1}{2}\Big(|\uparrow\uparrow\rangle + |\downarrow\downarrow\rangle - |\uparrow\downarrow\rangle - |\downarrow\uparrow\rangle\Big), &\quad \text{with} \quad T(X) = +1, \quad X = +1, \quad E = -4, \\
\frac{1}{2}\Big(|\uparrow\uparrow\rangle - |\downarrow\downarrow\rangle + i|\uparrow\downarrow\rangle - i|\downarrow\uparrow\rangle\Big), &\quad \text{with} \quad T(X) = -i, \quad X = -1, \quad E = 0, \\
\frac{1}{2}\Big(|\uparrow\uparrow\rangle - |\downarrow\downarrow\rangle - i|\uparrow\downarrow\rangle + i|\downarrow\uparrow\rangle\Big), &\quad \text{with} \quad T(X) = +i, \quad X = -1, \quad E = 0, \\
\frac{1}{2}\Big(|\uparrow\uparrow\rangle + |\downarrow\downarrow\rangle + |\uparrow\downarrow\rangle + |\downarrow\uparrow\rangle\Big), &\quad \text{with} \quad T(X) = -1, \quad X = +1, \quad E = +4.
\end{aligned}
\tag{245}
$$

Since we identify states with $L = 2 \mod 4$ with the continuum theory twisted by $X$, these lattice models twisted by $X$ should correspond to the untwisted continuum theory. Indeed, we identify the first state in (245) with the ground state with $Q_m = Q_w = 0$. The second and third states correspond to $Q_m = 0$ with $Q_w = \pm 1$. And the fourth line corresponds to $Q_m = Q_w = 0$ with $P = 1$ or $P = -1$.

Let us move to $L = 3$. Here the ground state of the lattice model is

$$\frac{1}{\sqrt{6}}\Big(|\downarrow\uparrow\uparrow\rangle + |\uparrow\downarrow\uparrow\rangle + |\uparrow\uparrow\downarrow\rangle + |\downarrow\downarrow\uparrow\rangle + |\downarrow\uparrow\downarrow\rangle + |\uparrow\downarrow\downarrow\rangle\Big). \tag{246}$$

It has $X = 1$, $T = 1$, and $\mathcal{Q} = -\frac{1}{4}$. This agrees with the continuum theory, twisted by $e^{i\frac{3\pi}{2}Q_w}$, where the ground state has $Q_m = -\frac{1}{4}$ and $Q_w = 0$.

The states at the next level are two-fold degenerate:

$$\frac{1}{\sqrt{6}}\Big[\big(|\downarrow\uparrow\uparrow\rangle + \omega|\uparrow\downarrow\uparrow\rangle + \omega^2|\uparrow\uparrow\downarrow\rangle\big) - \big(|\uparrow\downarrow\downarrow\rangle + \omega|\downarrow\uparrow\downarrow\rangle + \omega^2|\downarrow\downarrow\uparrow\rangle\big)\Big], \qquad \omega = e^{\pm\frac{2\pi i}{3}}. \tag{247}$$

These states indeed have $X = -1$, $T$ eigenvalues $\omega^{-1}$, and $\mathcal{Q} = -\frac{1}{4}$. This matches with the continuum values $Q_m = -\frac{1}{4}, Q_w = \pm 1$, so $P = \mp\frac{1}{4}$. Eq. (239) then gives $T = e^{\mp i\frac{\pi}{2}}e^{\mp\frac{2\pi i}{3}\frac{1}{4}} = e^{\mp\frac{2\pi i}{3}}$.

For $L = 4$, the ground state has the following wavefunction:

$$\Big(|\downarrow\uparrow\uparrow\uparrow\rangle + |\uparrow\downarrow\uparrow\uparrow\rangle + |\uparrow\uparrow\downarrow\uparrow\rangle + i|\uparrow\uparrow\uparrow\downarrow\rangle + (\uparrow\longleftrightarrow\downarrow)\Big) + \sqrt{2}\Big(|\downarrow\downarrow\uparrow\uparrow\rangle + |\uparrow\downarrow\downarrow\uparrow\rangle + |\uparrow\uparrow\downarrow\downarrow\rangle + |\downarrow\uparrow\uparrow\downarrow\rangle\Big). \tag{248}$$

Clearly it has $T = 1$, $X = 1$, and $\mathcal{Q} = 0$. In the continuum theory, since $N = 0$, the theory is not twisted and the unique ground state has $Q_m = Q_w = 0$.

The states at the next level are two-fold degenerate. One of them is given by:

$$\Big(|\downarrow\uparrow\uparrow\uparrow\rangle + i|\uparrow\downarrow\uparrow\uparrow\rangle - |\uparrow\uparrow\downarrow\uparrow\rangle - i|\uparrow\uparrow\uparrow\downarrow\rangle - (\uparrow\longleftrightarrow\downarrow)\Big) + \sqrt{2}e^{\frac{i\pi}{4}}\Big(|\downarrow\downarrow\uparrow\uparrow\rangle + i|\uparrow\downarrow\downarrow\uparrow\rangle - |\uparrow\uparrow\downarrow\downarrow\rangle - i|\downarrow\uparrow\uparrow\downarrow\rangle\Big). \tag{249}$$

The other is its complex conjugate. They have $X = -1$, $T = \pm i$, and $\mathcal{Q} = 0$. In the continuum theory, we identify these states as states with $Q_m = 0$, $Q_w = \pm 1$. Eq. (239) then gives $T = \pm i$.

# 7 Luttinger's theorem and filling constraints

## 7.1 General considerations

Luttinger's theorem [2, 3] relates the electron density to the volume of the Fermi surface. Just like anomalies, it depends only on the existence of global symmetries (in this case U(1) of electric charge, and translation symmetry) and it is not sensitive to small changes in the Hamiltonian. And like 't Hooft anomaly matching, it relates a UV property (the particle density) to an IR property (the volume of the Fermi surface). Indeed, Oshikawa's derivation of Luttinger's theorem [7] uses a background U(1) gauge fields, a standard tool in the study of anomalies, and makes it clear that the Luttinger theorem should be understood as the consequence of the filling constraint. That is, for a translation-invariant system with U(1) symmetry, at a given filling (i.e., U(1) charge per unit volume), the UV theory coupled to background gauge field must obey an anomalous transformation rule very similar to those that define 't Hooft anomaly, and must be matched by the IR theory [34]. Our goal in this section is to relate the filling constraint to our discussion of anomalies and to explain in what sense it is like an anomaly.

Let us start by reviewing our general setup for a one-dimensional system with a global U(1) symmetry generated by a charge $Q$. We will assume that the U(1) symmetry is not anomalous. We will discuss in parallel a lattice system with $L$ sites and periodic boundary conditions and the continuum theory on a circle.

We twist space by the group element $g = e^{i\sigma Q}$. This can be viewed as twisted boundary conditions, or as a flat background U(1) gauge field along space, or as a topological defect. Then, we compute a trace over the Hilbert space. On the lattice, it is

$$\mathcal{Z}(\beta, \sigma, \xi, n) = \text{Tr}\left[e^{-\beta H(\sigma)}e^{i\xi Q}T(\sigma)^n\right], \tag{250}$$

and in the continuum, it is

$$\mathcal{Z}(\beta, \sigma, \xi, \vartheta) = \text{Tr}\left[e^{-\beta H(\sigma)}e^{i\xi Q}e^{i\vartheta P(\sigma)}\right]. \tag{251}$$

Here $T(\sigma)$ is the lattice translation operator in the twisted Hilbert space and $P(\sigma)$ is the continuum translation operator in the twisted theory, normalized with the length of space such that the eigenvalues of $P(0)$ are integers. Here we trace over states with all possible values of $Q$. It is also common to restrict the trace to a fixed $Q$ subspace. We will do it soon.

Since $Q$ is not rotated by $\sigma$ (no self-anomaly), these partition functions are $2\pi$-periodic in $\sigma$ and $\xi$

$$\mathcal{Z}(\beta, \sigma, \xi, n) = \mathcal{Z}(\beta, \sigma + 2\pi, \xi, n) = \mathcal{Z}(\beta, \sigma, \xi + 2\pi, n), \tag{252}$$

and

$$\mathcal{Z}(\beta, \sigma, \xi, \vartheta) = \mathcal{Z}(\beta, \sigma + 2\pi, \xi, \vartheta) = \mathcal{Z}(\beta, \sigma, \xi + 2\pi, \vartheta). \tag{253}$$

The key equation due to the twist by $\sigma$ is (19). It relates a complete translation of space to the twisted boundary conditions

$$U(\sigma) = T(\sigma)^L = e^{i\sigma Q}, \tag{254}$$

on the lattice and

$$U(\sigma) = e^{2\pi i P(\sigma)} = e^{i\sigma Q}, \tag{255}$$

in the continuum. Consequently, for $\sigma \notin 2\pi\mathbb{Z}$ the momentum eigenvalues are not integers and the eigenvalues of $T(\sigma)$ are not $L$'th roots of one.

If there is no anomaly, Eq. (254) or (255) lead to

$$\mathcal{Z}(\beta, \sigma, \xi, n + L) = \mathcal{Z}(\beta, \sigma, \xi + \sigma, n), \tag{256}$$

or

$$\mathcal{Z}(\beta, \sigma, \xi, \vartheta + 2\pi) = \mathcal{Z}(\beta, \sigma, \xi + \sigma, \vartheta). \tag{257}$$

In fact, since we do not assume that there are any other symmetries in the problem, even if there are phases in these relations, as in (31), i.e., even if there are additional phases in Eq. (254) or (255), we can absorb these phase in redefinitions of the generators (as we did above).

The parameter space includes a two-torus labeled by $(\sigma, \xi) \sim (\sigma + 2\pi, \xi) \sim (\sigma, \xi + 2\pi)$. And in addition, we have the parameters $n \sim n + L$ or $\vartheta \sim \vartheta + 2\pi$. But these two sets of parameters are combined nontrivially. On the lattice, $(\sigma, \xi, n + L) \sim (\sigma, \xi + \sigma, n)$. And in the continuum, $(\sigma, \xi, \vartheta + 2\pi) \sim (\sigma, \xi + \sigma, \vartheta)$. This means that the circle parameterized by $\xi$ is fibered nontrivially over the torus parameterized by $(\sigma, \vartheta)$. This three manifold is known as the Heisenberg manifold.

We emphasize that the appearance of this nontrivial parameter space, corresponding to the twisted periodicity in (256) or (257) is not an anomaly. Equivalently, the partition function $\mathcal{Z}$, is a function on that space, rather than a section of a line bundle. In particular, the large $\beta$ limit of the partition function, which is determined by the low-lying states can be a constant reflecting a trivial gapped state.

The conclusion so far seems trivial: a system with a global (anomaly free) U(1) symmetry can have a unique gapped ground state in which the global symmetry is unbroken. The only

somewhat less trivial fact is that the arguments of the partition function $\mathcal{Z}$ take value in a nontrivial space. This fact will be important below.

If the system does not have an anomaly, why does it look as if it does have an anomaly? The crucial fact is that Luttinger's theorem applies to a system with fixed U(1) charge $q$, rather than to a system with all possible values of the charge. Here $q$ is the value of the total charge operator $Q$. Then typically the low-energy physics depends on the charge $q$, or the charge density.[42]

Naively, the partition function with fixed charge $\mathcal{Z}_q$ is the Fourier transform of $\mathcal{Z}$

$$\mathcal{Z}_q = \int d\xi e^{-i\xi q} \mathcal{Z}. \tag{258}$$

However, this Fourier transform is not well defined. $\mathcal{Z}$ is subject to the identification (256) (or (257)), but the exponential function $e^{-i\xi q}$ is not. Therefore, the integral in Eq. (258) is more subtle than it looks. To define it, we need to relax some of the identifications above, i.e., work on the covering space, such that (258) is meaningful. But then, when we go back to the original parameter space, we clearly find

$$\mathcal{Z}_q(\beta, \sigma, n + L) = e^{i\sigma q} \mathcal{Z}_q(\beta, \sigma, n), \tag{259}$$

or

$$\mathcal{Z}_q(\beta, \sigma, \vartheta + 2\pi) = e^{i\sigma q} \mathcal{Z}_q(\beta, \sigma, \vartheta). \tag{260}$$

In order to see whether the phases in the relations (259) and (260) lead to an anomaly, we should try to redefine them away. Consider first the case of the lattice theory (259). We can try to redefine $T(\sigma) \to e^{i\sigma \frac{q}{L}} T(\sigma)$, such that $T(\sigma)^L = 1$ and the phase in (259) is absent. However, after this redefinition, the shift $\sigma \to \sigma + 2\pi$ multiplies $\mathcal{Z}_q$ by $e^{2\pi i \frac{qn}{L}}$ and the anomaly is still present. An obvious exception is the special case where the charge density $\nu = \frac{q}{L}$ is an integer. Then, this redefinition removes all the phases and hence there is no anomaly. Thus the anomaly depends only on $\nu$ mod $\mathbb{Z}$. This will be demonstrated clearly in the example in section 7.2.

No such redefinition is possible in the continuum expression (260). Indeed, in this case, there is no dimensionless density $\nu$ that can affect it. In that case, $\mathcal{Z}_q$ is not a function on the parameter space, but it is a section of a line bundle, whose first Chern class is the integer $q$. Clearly, for $q = 0$ there is no anomaly.

In both cases, the anomaly leads to relations between the UV and the IR theory. When there is such an anomaly, the partition function $\mathcal{Z}_q$ cannot be a constant independent of $\sigma$ and $n$, or $\sigma$ and $\vartheta$. In particular the system cannot have a unique gapped ground state. (If that had been the case, then the large $\beta$ limit of $\mathcal{Z}_q$ would have been a constant.) For lattice systems with fixed filling $\nu$, the conclusion holds for $\nu \notin \mathbb{Z}$. In the continuum it applies to any $q \neq 0$. As an example, it implies that in the continuum, there is no "Mott insulator" phase, i.e., a gapped phase with finite charge density.

It is instructive to compare this conclusion to our statement above about lack of anomaly in the system with unconstrained $Q$ (Eq. (256) and (257)). In that case, we can have a unique gapped ground state – a state with zero charge. The conclusion here is that if we consider the sector of the Hilbert space with a fixed nonzero $Q$ eigenvalue $q$ there cannot be such a state.

---

[42]In practice, one considers a lattice with large $L$ and large $q$ with fixed charge density $\nu = \frac{q}{L}$ and then allow small changes in the charge that are much smaller than $q$. This can be done by fixing an appropriate chemical potential. (Note that the parameter $\xi$ is like an imaginary chemical potential.) We will demonstrate it in section 7.2.

It is trivial to generalize this discussion to several dimensions labeled by $i = 1, 2, .., D$. Now we have $D$ twists $\sigma_i$ and $D$ shifts $n^i$ (or $\vartheta^i$). Then, the partition functions (250) and (251) become

$$\mathcal{Z}(\beta, \sigma_i, \xi, n^i) = \text{Tr}\left[e^{-\beta H(\sigma_i)} e^{i\xi Q} T_i(\sigma)^{n^i}\right], \tag{261}$$

and

$$\mathcal{Z}(\beta, \sigma_i, \xi, \vartheta^i) = \text{Tr}\left[e^{-\beta H(\sigma_i)} e^{i\xi Q} e^{i\vartheta^i P_i(\sigma_i)}\right]. \tag{262}$$

We define the fixed charge partition functions as above (i.e., we go to the covering space, use (258), and then restrict the to fundamental domain). Then, the relations (259) and (260) become

$$\mathcal{Z}_q(\beta, \sigma_i, n^i + k^i L_i) = e^{iq \sum_i \sigma_i k^i} \mathcal{Z}_q(\beta, \sigma_i, n^i), \tag{263}$$

or

$$\mathcal{Z}(\beta, \sigma_i, \vartheta^i + 2\pi k^i) = e^{iq \sum_i \sigma_i k^i} \mathcal{Z}(\beta, \sigma_i, \vartheta^i), \tag{264}$$

with any integers $k^i$. We see that the eigenvalues of $U_i = T_i(\sigma_i)^{L_i}$ (or $U_i = e^{2\pi i P_i(\sigma_i)}$) are

$$e^{iq\sigma_i}. \tag{265}$$

For large systems, we expect the exponent in the eigenvalue of $U_i$ to be extensive. Therefore, we should write this eigenvalue as

$$e^{i\nu\sigma_i \prod_i L_i}, \quad \nu = \frac{q}{\prod_i L_i}, \tag{266}$$

i.e., $\nu$ is the charge density, or the filling fraction.

## 7.2 Emanant symmetries and their anomalies in the XY rotor model at fixed filling

Let us demonstrate these considerations in the special case of the $c = 1$ compact boson of section 3, or equivalently, the Luttinger liquid (34). These considerations will relate our results to the classic work of [67].

As a UV theory, we can take the XY rotor model in Eq. (7). Alternatively, we will consider the Villain model (93), which describes essentially the same physics as the XY rotor model:

$$H_{\text{Villain}} = \sum_j \left(\frac{U_0}{2} p_j^2 + \frac{J_0}{2}(\phi_{j+1} - \phi_j - 2\pi n_{j,j+1})^2\right) - g^2 \sum_j (e^{iE_{j,j+1}} + \text{h.c.}),$$
$$G_j = e^{i(E_{j,j+1} - E_{j-1,j})} e^{-2\pi i p_j} = 1. \tag{267}$$

The continuum limit of this model for small $U_0/J_0$ is the $c = 1$ compact boson theory. The real variable $\phi_j$ becomes effectively compact and is identified in the continuum with the scalar field $\Phi$. The periodic variable $E_{j,j+1}$ becomes the dual scalar $\Theta$. The continuum theory describes low-energy states with finite total charge $Q_m = \sum_j p_j$. In particular, the ground state has $Q_m = 0$.

We would like to consider the problem with fixed total charge $Q_m = \sum_j p_j = q = \nu L$. Since $q \in \mathbb{Z}$, the filling fraction

$$\nu = \frac{q}{L} = \frac{\hat{M}}{M}, \qquad \gcd(\hat{M}, M) = 1 \tag{268}$$

is rational. We will be interested in large $L$ with fixed $\nu$ and therefore $L = KM$ with large $K$.[43] Then, we can allow small changes in the total charge $Q_m = \sum_j p_j$ around $q = \nu L$.

---

[43]We can also take the large $L$ limit and let $\nu$ change at the same time, such that it converges to an irrational number.

Instead of constraining the total charge, as in Eq. (258), we shift the Hamiltonian (267) by an appropriate chemical potential term $-U_0 \nu Q_m$. Since the shift is by a conserved charge, it does not affect the dynamics. Now the low-energy states are those with $Q_m \approx q$.

Then, it is natural to we change variables to

$$
\begin{aligned}
\tilde{p}_j &= p_j - \nu, \\
\tilde{\phi}_j &= \phi_j, \\
\tilde{E}_{j,j+1} &= E_{j,j+1} - 2\pi \nu j, \\
\tilde{n}_{j,j+1} &= n_{j,j+1}.
\end{aligned}
\tag{269}
$$

It is easy to see that this change of variables preserves the commutation relations. Note that since $E_{j,j+1}$ and therefore also $\tilde{E}_{j,j+1}$ are $2\pi$-periodic, this redefinition is periodic under $j \to j + L$. (In checking that, use $2\pi \nu L = 2\pi q \in 2\pi \mathbb{Z}$.)

Using the new variables, after the shift by the chemical potential term, the Villain Hamiltonian (up to an additive constant), Gauss's law, and the conserved charge become

$$
\begin{aligned}
&\sum_j \left( \frac{U_0}{2} \tilde{p}_j^2 + \frac{J_0}{2} (\tilde{\phi}_{j+1} - \tilde{\phi}_j - 2\pi \tilde{n}_{j,j+1})^2 \right) - 2g^2 \sum_j \cos(\tilde{E}_{j,j+1} + 2\pi \nu j), \\
&G_j = e^{i(\tilde{E}_{j,j+1} - \tilde{E}_{j-1,j})} e^{-2\pi i \tilde{p}_j} = 1, \\
&\tilde{Q}_m = Q_m - q = \sum_j \tilde{p}_j.
\end{aligned}
\tag{270}
$$

Notably, we find that under the lattice translation generator $T$

$$
T \tilde{E}_{j,j+1} T^{-1} = E_{j+1,j+2} - 2\pi \nu j = \tilde{E}_{j+1,j+2} + 2\pi \nu.
\tag{271}
$$

We see that $T$ acts on the new variables as a combination of naive translation $\tilde{T}$ and a shift of $\tilde{E}_{j,j+1}$. We will soon use the fact that since $\nu = \frac{\hat{M}}{M}$ and $\tilde{E}_{j,j+1}$ is $2\pi$-periodic, $T^M = \tilde{T}^M$.

For $g = 0$, the lattice model (270) is the same as the original lattice model (267) (after removing the tildes). This can be understood as follows. For $g = 0$ the system has a winding symmetry and shifting the original Hamiltonian by $-U_0 \nu Q_m$ can be thought of as adding some winding defects. Since these defects are topological, they can be shifted to one point and since the total charge is an integer the defect is trivial and we find the original Hamiltonian. Indeed, in this case the model has both the original translation symmetry generated by $T$ and the one differing from it by a winding transformation $\tilde{T}$. This is not the case for nonzero $g$. Indeed, the winding operator in the Hamiltonians (267) and (270) do not differ only by removing the tildes.

We are interested in large $L$ and focus on the low-lying states. They have

$$
\tilde{Q}_m = \sum_j \tilde{p}_j \ll q \sim L,
\tag{272}
$$

and hence, small $\tilde{p}_j$. Because of Gauss's law, the shifted electric field $\tilde{E}_{j,j+1}$ is then smooth. Therefore, the new variables (270) have smooth continuum limits.

Let us examine the continuum limit. Consider first the simple case of $g = 0$. Then for every value of the coupling constants, the continuum theory is again the $c = 1$ compact boson (i.e., a Luttinger liquid), where $\tilde{E}_{j,j+1}$ is identified with $\Theta$, and $\tilde{\phi}_j$ becomes $\Phi$. (Recall that $\tilde{E}_{j,j+1}$ is smooth in the limit.)

Next, we would like to map the local operators of the UV lattice theory to local operators of the continuum theory. This is easy because for $g = 0$, the model in the new variables is the

same as the modified Villain model in section 4.1. Following the discussion there, we have the map

$$e^{im\tilde{E}_{j,j+1}} \rightarrow e^{im\Theta}\,. \tag{273}$$

The lattice operators $e^{imE_{j,j+1}} = e^{im(\tilde{E}_{j,j+1}+2\pi\nu j)}$ differ from $e^{im\tilde{E}_{j,j+1}}$ by $j$-dependent phases, so their correlation functions are the same up to these phases. However, since we defined the variables such that $\tilde{E}_{j,j+1}$ becomes smooth, the operators $e^{im\tilde{E}_{j,j+1}}$ have smooth continuum limits, while this is not the case for $e^{imE_{j,j+1}} = e^{im(\tilde{E}_{j,j+1}+2\pi\nu j)}$.

Now, we can perturb this $g = 0$ theory by turning on generic translation invariant, $\tilde{E}$-dependent operators in the lattice model, e.g., the term with $g^2$ in (270).

The operator that could have triggered the standard BKT transition, $\int e^{i\Theta}$ arises from $\sum_j e^{i\tilde{E}_{j,j+1}}$. However, this operator violates the lattice translation symmetry (271) and therefore it is absent.

Other operators like $\sum_j e^{iE_{j,j+1}} = \sum_j e^{i(\tilde{E}_{j,j+1}+2\pi\nu j)}$ preserve the lattice translation symmetry (indeed, they are present in the Hamiltonian (270) before we set $g = 0$), but they do not have smooth continuum limits. The $j$-dependent phase means that from the continuum point of view, they look like $e^{i\Theta}$ but with large spatial momentum $P = q \sim L$. Such operators do not act in the low-energy Hilbert space. Consequently, they can be ignored.

The previous discussion shows that the only deformations of the continuum limit of the $g = 0$ theory that violate its winding symmetry are of the form

$$\sum_j e^{imE_{j,j+1}} = \sum_j e^{im\tilde{E}_{j,j+1}} \rightarrow \int e^{im\Theta}\,, \qquad m \in M\mathbb{Z}\,. \tag{274}$$

They respect an emanant $\mathbb{Z}_M^{\mathcal{W}} \subset \mathrm{U}(1)_w$ symmetry.

Other $\mathrm{U}(1)_m$-invariant lattice operators are either not translation invariant or oscillate rapidly on the lattice scale. The most relevant of these is the one with $m = M$. It triggers a BKT transition at $R = \frac{2}{M}$ [68, 69]. The gapped states have $\langle\tilde{E}_{j,j+1}\rangle = \frac{2\pi}{M}n$ where $n = 0, 1, \ldots, M-1$ and spontaneously breaks the lattice translation symmetry.

This understanding allows us to relate the translation symmetry generator of the lattice model $T$ to the naive translation generator $\tilde{T}$ of the theory (270). The action of $T$ in (271) means that

$$T = e^{2\pi i\nu\tilde{Q}_w}\tilde{T}\,. \tag{275}$$

As a check, $T^M = \tilde{T}^M$.

Now, we can repeat the discussion of the various models above, and use the continuum information to find an exact lattice result

$$\begin{aligned} T &= \mathcal{W}^{-1}e^{\frac{2\pi i}{L}P}\,, \\ \mathcal{W} &= e^{-2\pi i\nu\tilde{Q}_w}\,. \end{aligned} \tag{276}$$

The two factors here are meaningful in the continuum limit, but only their product is meaningful on the lattice. However, using the continuum information, we can deduce this exact result for the low-lying states.

To summarize, by constraining the total charge density to be $\nu = \frac{\hat{M}}{M}$ the low-energy theory has an emanant $\mathbb{Z}_M^{\mathcal{W}}$ symmetry. This symmetry is an emanant symmetry rather than an emergent symmetry. Lattice operators that violate it are not merely irrelevant operators in the low-energy theory. They have large (order $L$) momentum in the low-energy theory and hence they decouple. Our discussion of the continuum theory in section 3 and the modified Villain model in section 4.1 show that this emanant $\mathbb{Z}_M^{\mathcal{W}}$ winding symmetry is anomaly free, but has a mixed anomaly with the momentum $\mathrm{U}(1)_m$ symmetry.

Let us add defects to this system and start with a defect of the emanant $\mathbb{Z}_M^{\mathcal{W}}$ symmetry. As in section 6.1, we consider $N = L$ mod $M \neq 0$. We would like to keep the Hamiltonian density as for $N = 0$, so we shift the Hamiltonian (267) by the chemical potential term $-U_0 \nu Q_m$. Note that since $N \neq 0$, now $\nu L$ is not an integer and cannot be interpreted as the quantized charge of the system $Q_m$. Again, we change variables

$$
\begin{aligned}
\tilde{p}_1 &= p_1 - \nu + N\nu, \\
\tilde{p}_j &= p_j - \nu, & j &= 2, 3, \cdots, L, \\
\tilde{E}_{j,j+1} &= E_{j,j+1} - 2\pi\nu j, & j &= 1, 2, \cdots, L.
\end{aligned}
\tag{277}
$$

As for $N = 0$ in (269), Gauss's law retains its form in term of the new variables $\tilde{p}_j$ and $\tilde{E}_{j,j+1}$. However, unlike the case $N = 0$ in (269), this change of variables is not periodic under $j \to j + N$ and this is the reason we wrote $j = 1, 2, \cdots L$ in the equation.

For $g = 0$, the Hamiltonian becomes (up to an additive constant)

$$
\frac{U_0}{2}\left( (\tilde{p}_1 - N\nu)^2 + \sum_{j\neq 1} \tilde{p}_j^2 \right) + \frac{J_0}{2} \sum_j (\phi_{j+1} - \phi_j - 2\pi n_{j,j+1})^2.
\tag{278}
$$

We recognized this as the Hamiltonian (109) with a winding defect with $\eta_w = -2\pi N\nu$.

This can be interpreted as follows. For $N = 0$, the total charge of the system was $q = \nu L \in \mathbb{Z}$ and it was spread evenly between the sites. Now, the chemical potential term leads to charge $\nu L$, which is not an integer. We still spread this charge evenly throughout the lattice, but in order to keep the total charge quantized, we add a defect with $\eta_w = -2\pi N\nu$ at one site. Then, we use the fact that the system with $g = 0$ has a winding symmetry, think of the charges at the sites as due to winding defects and move all of them to the first site. This leads to the Hamiltonian (278). We see that for $N \neq 0$, the $g = 0$ system retains its form, but has a single winding defect with $\eta_w = -2\pi N\nu$.

Repeating the discussion for $N = 0$, we can add various perturbation, including the one with $g^2$, and break the $U(1)_w$ symmetry of this system. However, an emanant $\mathbb{Z}_M^{\mathcal{W}} \subset U(1)_w$ symmetry remains. Also, it is clear that the winding defect with $\eta_w = -2\pi N\nu$ is a defect of that emanant symmetry. The way this symmetry arises and the properties of its defects are exactly the same as in our general discussion in section 6.1. Comparing with that section we see that we have $M = n$ and hence this symmetry does not have an anomaly.

Next, it is easy to add momentum defects. As in section 3, they are added by shifting in the Hamiltonian $\phi_{J+1} - \phi_J - 2\pi n_{J,J+1} \to \phi_{J+1} - \phi_J - 2\pi n_{J,J+1} + 2\pi\sigma$ at some links $(J, J+1)$. Their analysis is identical to the discussion in section 3, so we will not repeat it here.

Finally, let us rephrase this discussion from a continuum perspective. The starting point is the $c = 1$ compact boson of section 3. It can be described by various dual Lagrangians (33), (34), (36)

$$
\begin{aligned}
&\frac{R^2}{4\pi} \partial_\mu \Phi \partial^\mu \Phi, \\
&\frac{1}{4\pi R^2} \partial_\mu \Theta \partial^\mu \Theta, \\
&\frac{1}{2\pi} \partial_t \Phi \partial_x \Theta - \frac{1}{4\pi}\left[ R^{-2}(\partial_x \Theta)^2 + R^2(\partial_x \Phi)^2 \right], \\
&\Theta \sim \Theta + 2\pi, \qquad \Phi \sim \Phi + 2\pi.
\end{aligned}
\tag{279}
$$

The $U(1)_m$ charge is

$$
Q_m = \frac{R^2}{2\pi} \int_0^\ell dx\, \partial_t \Phi = \frac{1}{2\pi} \int_0^\ell dx\, \partial_x \Theta.
\tag{280}
$$

We are interested in this theory with $U(1)_m$ charge $q$ and in the presence of a $U(1)_w$ violating deformation $\cos(\Theta)$. We can use either of the three dual formulations in (279). Let us use the second one. We take space to be a circle with circumference $\ell$, add the chemical potential and the $U(1)_w$ violating operator and find

$$\frac{1}{4\pi R^2}\left[(\partial_t \Theta)^2 - \left(\partial_x \Theta - \frac{2\pi q}{\ell}\right)^2\right] + \lambda \cos(\Theta), \tag{281}$$

with a small coefficient $\lambda$, which is similar to $g^2$ in the lattice Hamiltonian (267). Then, we proceed as in the lattice discussion above. We express the model using $\tilde{\Theta} = \Theta - \frac{2\pi q x}{\ell}$. (Note that for integer $q$, this change of variables preserves the periodicity.) The Lagrangian becomes

$$\frac{1}{4\pi R^2}\left[(\partial_t \tilde{\Theta})^2 - (\partial_x \tilde{\Theta})^2\right] + \lambda \cos\left(\tilde{\Theta} + \frac{2\pi q x}{\ell}\right). \tag{282}$$

For $\lambda = 0$, this is the same as the model with $q = 0$.

The main point is that the winding violating operator $\cos(\Theta) = \cos\left(\tilde{\Theta} + \frac{2\pi q x}{\ell}\right)$ becomes an operator with spatial momentum $P = q$. (Recall that we normalize the spatial momentum $P$ with the size of the system, such that its eigenvalues are quantized.) For large $q$ the winding violating operator has large momentum and it decouples from the low-energy theory. It does not decouple because it is irrelevant, but because of its large spatial momentum. Consequently, $U(1)_w$ is an emanant symmetry for any value of the radius $R$.

## Acknowledgements

We thank I. Affleck, T. Banks, D. Else, E. Fradkin, D. Freed, P. Gorantla, D. Harlow, A. Kapustin, H.T. Lam, J. McGreevy, M. Metlitski, A. Prem, S. Sachdev, T. Senthil, S.-H. Shao, R. Thorngren, H.-H. Tu, A. Vishwanath, C. Wang, and X.-G. Wen for helpful discussions. We also thank T. Banks, H.T. Lam, and S.-H. Shao for comments on the manuscript.

**Funding information** The work of MC was supported in part by NSF under award number DMR-1846109. The work of NS was supported in part by DOE grant DE−SC0009988 and by the Simons Collaboration on Ultra-Quantum Matter, which is a grant from the Simons Foundation (651440, NS). Opinions and conclusions expressed here are those of the authors and do not necessarily reflect the views of funding agencies.

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
