# Peer review of "Lieb-Schultz-Mattis, Luttinger, and 't Hooft -- anomaly matching in lattice systems"

_SciPost Physics, doi:SciPost Phys. 15, 051 (2023)_

## Round 2 · Referee Report · Anonymous · 2023-2-11

Strengths
1. The authors develop a very detailed theory of anomalies in lattice systems with translation symmetry. I certainly think that there is much value and conceptual insight to be gained in formulating these ideas precisely, and they discuss many examples in detail.
2. The results on finite-size systems, for example the relationship between the momentum of the ground state and the anomalies, are particularly welcome and I think totally new.
Weaknesses
1. Regarding section 5: From various perspectives on LSM,it is clear that for a general internal symmetry group G, there should be an "anomaly" whenever the translation unit cell transforms projectively under G. However, this seems to be far from manifest from the perspectives described here. It's clear enough if G is Abelian -- then you can just read off the projective representation of a unit cell from eq. (5.4). [Indeed it's a well-known result about the cohomology of finite Abelian groups with U(1) coefficients that if phi is a 2-cocycle, then phi(g,h) - phi(h,g) contains all the information about the cohomology class of phi]. But the situation for non-Abelian G seems much less clear, especially due to the restriction that h \in C_g.
2. Throughout the paper, the authors are careful to consider systems with a finite (but large) system size L. This allows them to consider interesting properties of the finite-size systems, e.g. the momenta of the low-lying states. However, in condensed matter physics we are often interested only in the thermodynamic limit. I wonder whether there is a clean way to take the L->infty limit in order to simplify the formalism described, or if worrying about the details of finite-L is in fact unavoidable?
Report
For the reasons described above, I would recommend acceptance, subject to the authors responding to the points in the "Weaknesses" and "Requested changes" sections of this report.
Requested changes
1. In Section I, regarding the sentence "The modern view of these anomalies involves coupling the system to classical background gauge fields for these symmetries, placing the system on a closed Euclidean spacetime, e.g., a torus, and studying the partition function"
Although this is *one* way to think about anomalies, it is certainly not the *only* way as the authors seem to be implying here. For example for local anomalies (which includes at least some of the anomalies discussed in the present work) one can talk about non-conservation of charge in response to background gauge fields, or non-commutation of the local density operators. Even for anomalies of discrete symmetries there are ways to formulate them in a Hilbert space / Hamiltonian language without having to talk about partition functions.
2. In Section 1: "This picture of anomalies assumes a continuous and Lorentz invariant space-time".
I do not believe that Lorentz invariance plays any essential role in discussing anomalies in continuum theories.
3. In Section 5.5, I can see why for sufficiently large L, there should be an exact agreement with the continuum theory regarding the momentum of the low-lying states. However for very small values of L, isn't the agreement that the authors found in this particular model merely a coincidence, or curiosity? For a generic Hamiltonian, with extremely small system sizes like L=2, there is no reason to think that the eigenstates of the Hamiltonian will have anything to do with the continuum theory.
4. The terminology and notation in section 2 was a bit confusing:
(a) In the general setup in section 2, are the authors already assuming that the internal symmetries are on-site? Otherwise it seems there may be some ambiguities in how to introduce the twist defects.
(b) In Section 2.3.2, the sentence: "Consider first the symmetry operators associated with the internal symmetry G and denote them by h \in G". Since the authors are talking about the symmetry operators in the twisted theory here, shouldn't they already be calling them "h(g)"?
(c) On the lattice, for internal on-site symmetries, is it not the case that h(g) = h(1) anyway, if h is in the centralizer of g?'
(d) In the paragraph below (2.10), the sentence "We will denote these operators as h(g)", it is unclear what exactly "these operators" is referring to.
Author: Meng Cheng on 2023-04-07 [id 3564]
(in reply to Report 1 on 2023-02-11)
Weaknesses
- The anomaly we describe indeed follows from the projective representation of the local Hilbert space. We phrased it in the way we did in order to frame it in the standard way anomalies are being studied in quantum field theory and in mathematics. We would like to point out the following.
- Our main examples are based on SO(3) and O(2), which are non-Abelian.
- As we said, we limited ourselves to flat background fields. This is the reason for the appearance of the centralizer. We do not know whether all anomalies can be probed using flat fields, but this is the case in all the examples we studied.
- We examined the anomaly at finite L. Then, we studied in detail the limit as L goes to infinity. Interestingly, that limit can be subtle and it can depend on L mod N for some fixed finite N. This dependence on L mod N corresponds to defects in the low-energy continuum theory and exposes subtle anomalies. They cannot be found by studying the infinite L theory. We think that these facts are of interest also to condensed matter physicists.
Requested changes:
- We added a footnote to clarify it.
- The treatment of anomalies in quantum field theory and in mathematics definitely depends on Lorentz invariance. In special cases, the anomaly can be derived without assuming Lorentz invariance. And this is the approach we took here.
- A clarifying comment was added.
- See below (a) There is no assumption of “on-site” at this stage. Therefore, there might be some ambiguity in constructing the defect and this issue is discussed in the later examples. The discussion at this stage is independent of it. (b) The notation is that h is an arbitrary group element. It can be realized in different ways depending on the twist. In particular, the realization might be projective. This is explained later and then we introduce and use the notation h(g). (c) At this stage it was not assumed that the symmetry acts “on-site.” The precise meaning of “on-site” is discussed later. (d) This was clarified.
Author: Meng Cheng on 2023-04-09 [id 3567]
(in reply to Report 3 on 2023-03-30)Weaknesses
It would be nice to apply these methods to new systems. Indeed, the lattice model in section 4.1 is new and it demonstrates nicely our general picture and methods.
Report
We agree with the referee that extending these ideas to higher dimensions and to fermionic systems would be very interesting. We know that a number of people are thinking about it right now.
As the referee said, the XXZ model has a transition at $\lambda_z=\pm 1$ and there is no transition at $\lambda_z=-\frac{1}{\sqrt{2}}$. This might appear surprising because the low energy theory is the same as that of the XY model, which exhibits a BKT transition at the point corresponding to $\lambda_z=-\frac{1}{\sqrt{2}}$ . The discussion of the emanant symmetry in section 1.3 explains why there is no transition at $\lambda_z=-\frac{1}{\sqrt{2}}$.
The same issue with emanant symmetry is used in section 7.2, where the system with a chemical potential exhibits a new emanant symmetry that prevents the standard BKT transition. This is discussed in the paragraph following equation 7.33 where the operator is absent in the low-energy effective Lagrangian. We hope that this explanation addresses the last comment of the referee.
Requested changes:

---

## Round 2 · Referee Report · Anonymous · 2023-3-30

Strengths
1-The manuscript is written in a pedagogical form and is very readable. With slight polish, the paper can be adapted to be an introductory review or textbook.
2-The work pays attention to the anomalies on lattice systems, which bridges the models on discrete finite lattice in UV limit to low-energy physics on continuum field theory in IR limit.
3-The framework can unify the Lieb-Schultz-Mattis theorem, Luttinger theorem and anomaly matching on the conceptual level and has the potential to offer new perspective in these topics.
Weaknesses
1-The examples of lattice models are all well studied by other methods before. A novel lattice model is required to manifest the power of present theory.
Report
The paper presents a delicate lattice theory of anomalies with anomalous symmetries including both internal and spatial symmetries. The framework can harbor Lieb-Schultz-Mattis theorem, Luttinger theorem, and so on.
I highly recommend the work towards publication. Nevertheless I still suggest the authors respond to the points in the “Weakness” and “Requested changes” to improve the manuscript.
The authors limit to the 1+1D bosonic lattice systems. A sketchy discussion about possible generalizations may benefit readers, like 2+1D systems and/or fermionic systems, other anomalies and so on.
The XXZ model has phase transitions at lambda_z=±1. The statement on page.9 “The standard BKT transition at R_XXZ=2 (corresponding to lambda_z=-1/sqrt(2))” is a bit confusing.
The statement on page. 76 “It does not decouple because it is irrelevant, but because of its large spatial momentum.” seems to be self-contradictory if the contexts are understood correctly. Also there is a concern about such decouple in regard to whether the order matters. Does “first decouple on finite size and then take the thermodynamic limit” equivalent to “first take the thermodynamic limit and then decouple”?
Requested changes
1-Typo at footnote of page.6, “rho(g(1))” should be “rho(h(1))”.
2-Add doi and publication information of References, e.g. Ref.[1] “https://doi.org/10.1016/0003-4916(61)90115-4”, Ref.[6] “DOI 10.4171/90-2/46 Published in Rupert L. Frank, Ari Laptev, Mathieu Lewin, and Robert Seiringer eds. "The Physics and Mathematics of Elliott Lieb'' vol.~2, pp.~405--446 (European Mathematical Society Press, 2022)”, Ref.[10] “https://doi.org/10.1007/s00220-019-03343-5”, and others.
3-Define the “M” below eq.(1.6) in “Phi→Phi+2pi/M” precisely.
4-For the completeness of the presentation, there should have a “Summary and discussion” part at the end of the manuscript.

---

## Round 3 · Referee Report · Anonymous (Referee 3) · 2023-4-12

Report

Now that the authors have responded to the points in the Report, I recommend publication in SciPost Physics.

---

## Round 3 · Referee Report · Anonymous (Referee 1) · 2023-4-17

Report

The authors have addressed my comments from the previous report and the paper is now ready for publication.

---

## Round 3 · List of Changes

We have added a few clarifications and comments to address questions from the referee reports and have corrected typos. An additional reference ([67]) was added.

---

## Editorial Decision

published